# Lasso Bandit with Compatibility Condition on Optimal Arm

**Harin Lee**[*]
Seoul National University
harinboy@snu.ac.kr

**Taehyun Hwang**[*]
Seoul National University
th.hwang@snu.ac.kr

**Min-hwan Oh**[†]
Seoul National University
minoh@snu.ac.kr

## Abstract

We consider a stochastic sparse linear bandit problem where only a sparse subset of context features affects the expected reward function, i.e., the unknown reward parameter has a sparse structure. In the existing Lasso bandit literature, the compatibility conditions, together with additional diversity conditions on the context features are imposed to achieve regret bounds that only depend logarithmically on the ambient dimension $d$. In this paper, we demonstrate that even without the additional diversity assumptions, the *compatibility condition on the optimal arm* is sufficient to derive a regret bound that depends logarithmically on $d$, and our assumption is strictly weaker than those used in the lasso bandit literature under the single-parameter setting. We propose an algorithm that adapts the forced-sampling technique and prove that the proposed algorithm achieves $\mathcal{O}(\text{poly} \log dT)$ regret under the margin condition. To our knowledge, the proposed algorithm requires the weakest assumptions among Lasso bandit algorithms under the single-parameter setting that achieve $\mathcal{O}(\text{poly} \log dT)$ regret. Through numerical experiments, we confirm the superior performance of our proposed algorithm.

## 1 Introduction

Linear contextual bandits (Abe & Long, 1999; Auer, 2002; Chu et al., 2011; Lattimore & Szepesvári, 2020) are a generalization of the classical Multi-Armed Bandit problem (Robbins, 1952; Lai & Robbins, 1985). In this sequential decision-making problem, the decision-making agent is provided with a context in the form of a feature vector for each arm in each round, and the expected reward of the arm is a linear function of the context vector for the arm and the unknown reward parameter. To be specific, in each round $t \in [T] := \{1, ..., T\}$, the agent observes feature vectors of the arms $\{\mathbf{x}_{t,k} \in \mathbb{R}^d : k \in [K]\}$. Then, the agent selects an arm $a_t \in [K]$ and observes a sample of a stochastic reward with mean $\mathbf{x}_{t,a_t}^\top \boldsymbol{\beta}^*$, where $\boldsymbol{\beta}^* \in \mathbb{R}^d$ is a fixed parameter that is unknown to the agent. Linear contextual bandits are applicable in various problem domains, including online advertising, recommender systems, and healthcare applications (Chu et al., 2011; Li et al., 2016; Zeng et al., 2016; Tewari & Murphy, 2017). In many applications, the feature space may exhibit high dimensionality ($d \gg 1$); however, only a small subset of features typically affects the expected reward, while the remainder of the features may not influence the reward at all. Specifically, the unknown parameter vector $\boldsymbol{\beta}^*$ is said to be *sparse* when only the elements corresponding to pertinent features possess non-zero values. The sparsity of $\boldsymbol{\beta}^*$ is represented by the sparsity index $s_0 = \|\boldsymbol{\beta}^*\|_0 < d$, where $\|\mathbf{x}\|_0$ denotes the number of non-zero entries in the vector $\mathbf{x}$. Such a problem setting is called the *sparse linear contextual bandit*.

There has been a large body of literature addressing the sparse linear contextual bandit problem (Abbasi-Yadkori et al., 2012; Gilton & Willett, 2017; Wang et al., 2018; Kim & Paik, 2019; Bastani & Bayati, 2020; Hao et al., 2020b; Li et al., 2021; Oh et al., 2021; Ariu et al., 2022; Chen et al., 2022; Li et al., 2022; Chakraborty et al., 2023). To efficiently take advantage of the sparse structure, the Lasso (Tibshirani, 1996) estimator is widely used to estimate the unknown parameter vector. Utilizing the $\ell_1$-error bound of the Lasso estimation, many Lasso-based linear bandit

---

[*]Equal contribution
[†]Corresponding author

algorithms achieve sharp regret bounds that only depend logarithmically on the ambient dimension $d$. Furthermore, a margin condition (see Assumption 2) is often utilized to derive an even poly-logarithmic regret in the time horizon, thereby achieving (poly-)logarithmic dependence on both $d$ and $T$ simultaneously (Bastani & Bayati, 2020; Wang et al., 2018; Li et al., 2021; Ariu et al., 2022; Li et al., 2022; Chakraborty et al., 2023).

While these algorithms attain sharper regret bounds, there is no free lunch. The analysis of the existing results achieving $\mathcal{O}(\text{poly} \log dT)$ regret heavily depends on various stochastic assumptions on the context vectors, whose relative strengths often remain unchecked. The regret analysis of the Lasso-based bandit algorithms necessitates satisfaction of the compatibility condition (Van De Geer & Bühlmann, 2009) for the empirical Gram matrix $\sum_t \mathbf{x}_{t,a_t} \mathbf{x}_{t,a_t}^\top$ constructed from previously selected arms. Ensuring this compatibility—or an alternative form of regularity, such as the restricted eigenvalue condition—for the empirical Gram matrices requires an underlying assumption about the compatibility of the theoretical Gram matrix, e.g., $\frac{1}{K}\mathbb{E}[\sum_k \mathbf{x}_{t,k}\mathbf{x}_{t,k}^\top]$. Moreover, to establish regret bounds, additional assumptions regarding the diversity of context vectors — e.g., anti-concentration, relaxed symmetry, and balanced covariance — are made (refer to Table 1 for a comprehensive comparison). Many of these assumptions are needed solely for technical purposes, and their complexity often obscures the relative strength of one assumption over another. Thus, the following research question arises:

**Question**: *Is it possible to construct weaker conditions than those in existing conditions to achieve $\mathcal{O}(\text{poly} \log dT)$ regret in the sparse linear contextual bandit (under the single-parameter setting)?*

In this paper, we provide an affirmative answer to the above question. We show that (i) the compatibility condition on the optimal arm is strictly weaker than the existing stochastic conditions imposed on context vectors for $\mathcal{O}(\text{poly} \log dT)$ regret in the sparse linear bandit literature (under the single-parameter setting). That is, the existing conditions in the relevant literature imply our proposed compatibility condition on the optimal arm, but the converse does not hold (refer to Figure 1). Additionally, (ii) we propose an algorithm that achieves $\mathcal{O}(\text{poly} \log dT)$ regret under the compatibility condition on the optimal arm combined with the margin condition. Therefore, to the best of our knowledge, the compatibility condition on the optimal arm that we study in this work — combined with the margin condition — is the mildest condition that allows $\mathcal{O}(\text{poly} \log dT)$ regret for sparse linear contextual bandits (Oh et al., 2021; Li et al., 2021; Ariu et al., 2022; Chakraborty et al., 2023).

Our contributions are summarized as follows:

- We propose a forced-sampling-based algorithm for sparse linear contextual bandits: `FS-WLasso`. The proposed algorithm utilizes the Lasso estimator for dependent data based on the compatibility condition on the optimal arm. `FS-WLasso` explores for a number of rounds by uniformly sampling context features and then exploits the Lasso estimated obtained by weighted mean squared error with $\ell_1$-penalty. We establish that the regret bound of our proposed algorithm is $\mathcal{O}(\text{poly} \log dT)$.

- One of the key challenges in the regret analysis for bandit algorithms using Lasso is ensuring that the empirical Gram matrix satisfies the compatibility condition. Most existing sparse bandit algorithms based on Lasso not only assume the compatibility condition on the expected Gram matrix, but also impose an additional diversity condition for context features (e.g., anti-concentration, relaxed symmetry, and balanced covariance), facilitating automatic feature space exploration. However, we show that the *compatibility condition on the optimal arm* is sufficient to achieve $\mathcal{O}(\text{poly} \log dT)$ regret under the margin condition, and demonstrate that our assumption on the context distribution is strictly weaker than those used in the existing sparse linear bandit literature that achieve $\mathcal{O}(\text{poly} \log dT)$ regret. We believe that the compatibility condition on the optimal arm studied in our work can be of interest in future Lasso bandit research.

- To establish the regret bounds in Theorems 2 and 3, we introduce a novel analysis technique based on high-probability analysis that utilizes mathematical induction, which captures the cyclic structure of optimal arm selection and the resulting small estimation errors. We believe that this new technique can be applied to the analysis of other bandit algorithms and therefore can be of independent interest (See discussions in Section 3.3).

- We evaluate our algorithms through numerical experiments and demonstrate its consistent superiority over existing methods. Specifically, even in cases where the context features of

Table 1: Comparisons with the existing high-dimensional linear bandits with a single parameter setting. For algorithms using the margin condition, we present regret bounds for the 1-margin (for simple exposition). We define $\mathbf{\Sigma} := \frac{1}{K}\mathbb{E}[\sum_{k=1}^{K} \mathbf{x}_{t,k}\mathbf{x}_{t,k}^\top]$, $\mathbf{\Sigma}_k := \mathbb{E}[\mathbf{x}_{t,k}\mathbf{x}_{t,k}^\top]$ for each $k \in [K]$, $\mathbf{\Sigma}_\Gamma^* := \mathbb{E}[\mathbf{x}_{t,a_t^*}\mathbf{x}_{t,a_t^*}^\top \mid \mathbf{x}_{t,a_t^*}^\top\boldsymbol{\beta}^* \geq \max_{k \neq a_t^*} \mathbf{x}_{t,k}^\top\boldsymbol{\beta}^* + \Delta_*]$, and $\mathbf{\Sigma}^* := \mathbb{E}[\mathbf{x}_{t,a_t^*}\mathbf{x}_{t,a_t^*}^\top]$.

| Paper | Compatibility or Eigenvalue | Margin | Additional Diversity | Regret |
|---|---|---|---|---|
| Kim & Paik (2019) | Compatibility on $\mathbf{\Sigma}$ | ✗ | ✗ | $\mathcal{O}(s_0\sqrt{T}\log(dT))$ |
| Hao et al. (2020b) | Minimum eigenvalue of $\mathbf{\Sigma}$ | ✗ | ✗ | $\mathcal{O}((s_0 T \log d)^{\frac{2}{3}})$ |
| Oh et al. (2021) | Compatibility on $\mathbf{\Sigma}$ | ✗ | Relaxed symmetry & balanced covariance | $\mathcal{O}(s_0\sqrt{T\log(dT)})$ |
| Li et al. (2021) | Bounded sparse eigenvalue of $\mathbf{\Sigma}_\Gamma^*$ | ✓ | Anti-concentration | $\mathcal{O}(s_0^2(\log(dT))\log T)$ |
| Ariu et al. (2022) | Compatibility on $\mathbf{\Sigma}$ | ✓ | Relaxed symmetry & Balanced covariance | $\mathcal{O}(s_0^2 \log dT)^\dagger$ |
| Chakraborty et al. (2023) | Maximum sparse eigenvalue of $\mathbf{\Sigma}_k$ | ✓ | Anti-concentration | $\mathcal{O}(s_0^2(\log(dT))\log T)$ |
| **This work** | Compatibility on $\mathbf{\Sigma}^*$ | ✓ | ✗ | $\mathcal{O}(s_0^2(\log(dT))\log T)$ |

$\dagger$ Ariu et al. (2022) show a regret bound of $\mathcal{O}(s_0^2 \log d + s_0(\log s_0)^{\frac{3}{2}}\log T)$, but they implicitly assume that the $\ell_2$ norm of feature is bounded by $s_A$ when applying the Cauchy-Schwarz inequality in their proof of Lemma 5.8. We display the regret bound when only the $\ell_\infty$ norms of features are bounded.

all arms except for the optimal arm are fixed (thus, assumptions such as anti-concentration are not valid), our proposed algorithms outperform the existing algorithms.

## 1.1 RELATED LITERATURE

Although significant research has been conducted on linear bandits (Abe & Long, 1999; Auer, 2002; Dani et al., 2008; Rusmevichientong & Tsitsiklis, 2010; Abbasi-Yadkori et al., 2011; Chu et al., 2011; Agrawal & Goyal, 2013; Abeille & Lazaric, 2017; Kveton et al., 2020a) and generalized linear bandits (Filippi et al., 2010; Li et al., 2017; Faury et al., 2020; Kveton et al., 2020b; Abeille et al., 2021; Faury et al., 2022), applying them to high-dimensional linear contextual bandits poses challenges in leveraging the sparse structure within the unknown reward parameter. Consequently, it might lead to a regret bound that scales with the ambient dimension $d$ rather than the sparse set of features with cardinality $s_0$. To overcome such challenges, high-dimensional linear contextual bandits have been investigated under the sparsity assumption and have attracted significant attention under different problem settings. Bastani & Bayati (2020) consider a *multiple-parameter* setting where each arm has its own underlying parameter and only one context vector is generated per round. Bastani & Bayati (2020) propose `Lasso Bandit` that uses the forced sampling technique (Goldenshluger & Zeevi, 2013) and the Lasso estimator (Tibshirani, 1996). They establish a regret bound of $\mathcal{O}(Ks_0^2(\log dT)^2)$ where $K$ is the number of arms. Under the same problem setting as Bastani & Bayati (2020), Wang et al. (2018) propose `MCP-Bandit` that uses uniform exploration for $\mathcal{O}(s_0^2 \log(dT))$ rounds and the minimax concave penalty (MCP) estimator (Zhang, 2010). They show the improved regret bound of $\mathcal{O}(s_0^2(\log d + s_0)\log T)$.

On the other hand, there has also been an amount of work in the *single-parameter* setting where $K$ different contexts are generated for each arm at each round and the rewards of all arms are determined by one shared parameter. Kim & Paik (2019) leverage a doubly-robust technique (Bang & Robins, 2005) from the missing data literature to develop `DR Lasso Bandit`, achieving a regret upper bound of $\mathcal{O}(s_0\sqrt{T}\log(dT))$. Oh et al. (2021) present `SA LASSO BANDIT`, which requires neither knowledge of the sparsity index nor an exploration phase, enjoying the regret upper bound of $\mathcal{O}(s_0\sqrt{T}\log(dT))$. Ariu et al. (2022) design `TH Lasso Bandit`, adapting the idea of Lasso with thresholding, originally proposed by Zhou (2010). This algorithm estimates the unknown reward parameter along with its support, achieving a regret bound of $\mathcal{O}(s_0^2 \log dT)$ under the 1-margin condition (Assumption 2). All the aforementioned algorithms rely on the compatibility condition of the expected Gram matrix for the averaged arm, denoted by $\mathbf{\Sigma} := \frac{1}{K}\mathbb{E}[\sum_{k\in[K]} \mathbf{x}_k\mathbf{x}_k^\top]$. Moreover, Oh et al. (2021); Ariu et al. (2022) impose strong conditions on the context distribution, such as relaxed symmetry and balanced covariance (Assumptions 7 and 8). There is another line of work that combines the Lasso estimator with exploration techniques in the linear bandit literature, such as the upper confidence bound (UCB) or Thompson sampling (TS). Li et al. (2021) introduce an algorithm

that constructs an $\ell_1$-confidence ball centered at the Lasso estimator, then selects an optimistic arm from the confidence set. Chakraborty et al. (2023) propose a Thompson sampling algorithm that utilizes the sparsity-inducing prior suggested by Castillo et al. (2015) for posterior sampling. Under assumptions such as the general margin condition, bounded sparse eigenvalues of the expected Gram matrix for each arm, and anti-concentration conditions on context features, both Li et al. (2021) and Chakraborty et al. (2023) achieve a $\mathcal{O}(\text{poly} \log dT)$ regret bound. Hao et al. (2020b) propose `ESTC`, an *explore-then-commit* paradigm algorithm that achieves a regret bound of $\mathcal{O}((s_0 T \log d)^{\frac{2}{3}})$ under the fixed arm set setting. Li et al. (2022) introduce a unified algorithm framework named *Explore-the-Structure-Then-Commit* for various high-dimensional stochastic bandit problems. Li et al. (2022) establish a regret bound of $\mathcal{O}(s_0^{\frac{1}{3}} T^{\frac{2}{3}} \sqrt{\log(dT)})$ for the Lasso bandit problem. Chen et al. (2022) propose `SPARSE-LINUCB` algorithm, which estimates the reward parameter using the best subset selection method based on generalized support recovery.

## 2 PRELIMINARIES

**Notations.** For a positive number $N$, we denote $[N]$ as a set containing positive integers up to $N$, i.e., $[N] := \{1, \ldots, N\}$. For a vector $\mathbf{v} \in \mathbb{R}^d$, we denote its $j$-th component by $v_j$ for $j \in [d]$, its transpose by $\mathbf{v}^\top$, its $\ell_0$-norm by $\|\mathbf{v}\|_0 = \sum_{j \in [d]} \mathbb{1}\{v_j \neq 0\}$, its $\ell_2$-norm by $\|\mathbf{v}\|_2 = \sqrt{\mathbf{v}^\top \mathbf{v}}$, and its $\ell_\infty$-norm by $\|\mathbf{v}\|_\infty = \max_{j \in [d]} |v_j|$. For each $I \subset [d]$ and $\mathbf{v} \in \mathbb{R}^d$, $\mathbf{v}_I = [v_{1,I}, \ldots, v_{d,I}]^\top$ where for all $j \in [d]$, $v_{j,I} = v_j \mathbb{1}\{j \in I\}$. Refer to Appendix A for a more detailed explanation of the notations.

**Problem Setting.** We consider a stochastic linear contextual bandit problem where $T$ is the number of rounds and $K(\geq 3)$ is the number of arms. In each round $t \in [T]$, the learning agent observes a set of context features for all arms $\{\mathbf{x}_{t,i} \in \mathcal{X} : i \in [K]\} \subset \mathbb{R}^d$ drawn i.i.d. from an unknown joint distribution, chooses an arm $a_t \in [K]$, and receives a reward $r_{t,a_t}$. We assume that $r_{t,a_t} = \mathbf{x}_{t,a_t}^\top \boldsymbol{\beta}^* + \eta_t$ where $\boldsymbol{\beta}^* \in \mathbb{R}^d$ is the unknown reward parameter and $\eta_t$ is an independent $\sigma$-sub-Gaussian random variable such that $\mathbb{E}[\eta_t | \mathcal{F}_{t-1}] = 0$ for the sigma-algebra $\mathcal{F}_t$ generated by $(\{\mathbf{x}_{\tau,i}\}_{\tau \in [t], i \in [K]}, \{a_\tau\}_{\tau \in [t]}, \{r_{\tau, a_\tau}\}_{\tau \in [t-1]})$, i.e., $\mathbb{E}[e^{s \eta_t} | \mathcal{F}_t] \leq e^{s^2 \sigma^2 / 2}$ for all $s \in \mathbb{R}$. We assume $\{\mathbf{x}_{t,1}, \ldots, \mathbf{x}_{t,K}\}_{t \geq 1}$ is a sequence of i.i.d. samples from some unknown distribution $\mathcal{D}_\mathcal{X}$ on the Lebesgue measurable sets. Note that dependency across arms in a given round is allowed. We also define the active set $S_0 = \{j : \boldsymbol{\beta}_j^* \neq 0\}$ as the set of indices $j$ for which $\boldsymbol{\beta}_j^*$ is non-zero. Let $s_0 := |S_0|$ denote the cardinality of the active set $S_0$, which satisfies $s_0 \ll d$.

Define $a_t^* := \text{argmax}_{k \in [K]} \mathbf{x}_{t,k}^\top \boldsymbol{\beta}^*$ as the optimal arm in round $t$. Then, the goal of the agent is to minimize the following cumulative regret:

$$R(T) = \sum_{t=1}^{T} \left( \mathbf{x}_{t,a_t^*}^\top \boldsymbol{\beta}^* - \mathbf{x}_{t,a_t}^\top \boldsymbol{\beta}^* \right) .$$

### 2.1 ASSUMPTIONS

We present a list of assumptions used for the regret analysis later in Section 3.2.

**Assumption 1** (Boundedness). *For absolute constants $x_{\max}, b > 0$, we assume $\|\mathbf{x}\|_\infty \leq x_{\max}$ for all $\mathbf{x} \in \mathcal{X}$, and $\|\boldsymbol{\beta}^*\|_1 \leq b$, where $b$ may be unknown.*

**Assumption 2** ($\alpha$-margin condition). *Let $\Delta_t = \mathbf{x}_{t,a_t^*}^\top \boldsymbol{\beta}^* - \max_{k \neq a_t^*} \mathbf{x}_{t,k}^\top \boldsymbol{\beta}^*$ be the instantaneous gap in round $t$. For $\alpha > 0$, there exists a constant $\Delta_* > 0$ such that for any $h > 0$ and for all $t \in [T]$, $\mathbb{P}\left(\Delta_t \leq h\right) \leq \left(\frac{h}{\Delta_*}\right)^\alpha$ .*

**Assumption 3** (Compatibility condition on the optimal arm). *For a matrix $\mathbf{M} \in \mathbb{R}^{d \times d}$ and a set $I \subseteq [d]$, the compatibility constant $\phi(\mathbf{M}, I)$ is defined as*

$$\phi^2(\mathbf{M}, I) := \min_{\boldsymbol{\beta}} \left\{ \frac{|I| \boldsymbol{\beta}^\top \mathbf{M} \boldsymbol{\beta}}{\|\boldsymbol{\beta}_I\|_1^2} : \|\boldsymbol{\beta}_{I^c}\|_1 \leq 3 \|\boldsymbol{\beta}_I\|_1 \neq 0 \right\} .$$

*Let us denote the context feature for the optimal arm in round $t$ by $\mathbf{x}_{t,a_t^*}$. Then, we assume that the expected Gram matrix of the optimal arm $\boldsymbol{\Sigma}^* := \mathbb{E}[\mathbf{x}_{t,a_t^*} \mathbf{x}_{t,a_t^*}^\top]$ satisfies the compatibility condition*

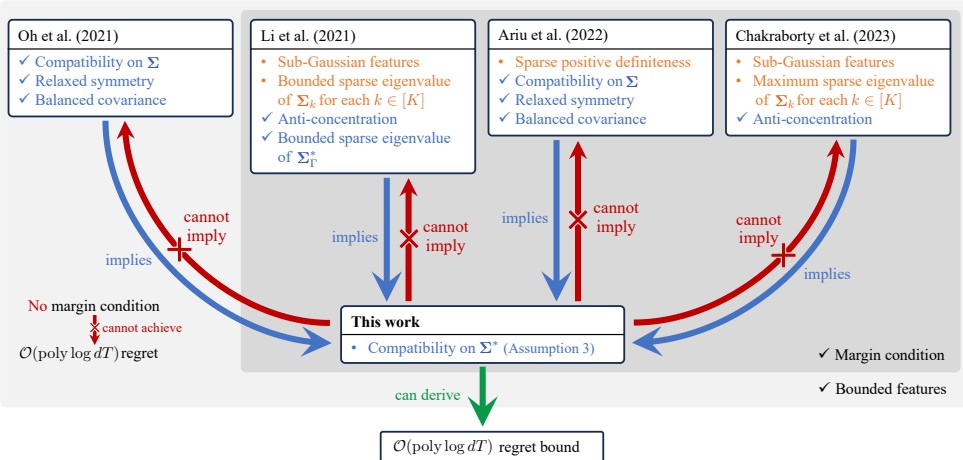

Figure 1: Illustration of relationships among distributional assumptions on context used in the sparse linear contextual bandit literature. The blue arrows represent *implication* relationships while the red arrows represent *infeasible implication* relationships. The conditions written in blue with the check bullet ✓ in the figure imply the compatibility on the optimal arm (Assumption 3), serving as sufficient conditions, while the conditions written in orange indicate additional assumptions necessary to achieve the existing methods' regret guarantees, but not needed in our analysis. The case where all sub-optimal arms are fixed serves as a counter-example for the *infeasible implication* relationships. We provide the proofs of the implication relationship in Appendix B which may be of independent interest.

with $\phi_* > 0$, i.e., $\phi^2(\Sigma^*, S_0) \geq \phi_*^2$. Note that $\Sigma^*$ is time-invariant since the set of features is drawn i.i.d. for each round.

**Discussion of assumptions.** Assumption 1 is a standard regularity assumption commonly used in the sparse linear bandit literature (Bastani & Bayati, 2020; Hao et al., 2020b; Ariu et al., 2022; Li et al., 2022; Chakraborty et al., 2023). It indicates that both the context features and the true parameter are bounded.

Assumption 2 restricts the probability that the expected reward of the optimal arm is close to those of the sub-optimal arms. To our best knowledge, the margin condition in the bandit setting was first introduced in Goldenshluger & Zeevi (2013) and is widely used in linear contextual bandit literature (Wang et al., 2018; Bastani & Bayati, 2020; Papini et al., 2021; Li et al., 2021; Bastani et al., 2021; Ariu et al., 2022; Chakraborty et al., 2023). Unlike the minimum gap condition (Abbasi-Yadkori et al., 2011; Papini et al., 2021), which prohibits the instantaneous gap from being smaller than a fixed constant, the margin condition allows a probability of a small gap. The case where $\alpha = 0$ imposes no additional constraints, while $\alpha = \infty$ is equivalent to the minimum gap condition. The margin condition with general $\alpha$ smoothly bridges the cases with and without the minimum gap.

Assumption 3 is related to the compatibility condition used to guarantee the convergence property of the Lasso estimator in the high-dimensional statistics literature (Bühlmann & Van De Geer, 2011). Since the compatibility condition ensures that the Lasso estimator approaches its true value as the number of samples grows large, many pieces of high-dimensional linear contextual bandit literature assume the condition (Wang et al., 2018; Kim & Paik, 2019; Bastani & Bayati, 2020; Oh et al., 2021; Ariu et al., 2022). Kim & Paik (2019); Oh et al. (2021); Ariu et al. (2022) assume the compatibility condition on $\Sigma := \frac{1}{K}\mathbb{E}[\sum_k \mathbf{x}_{t,k}\mathbf{x}_{t,k}^\top]$. Li et al. (2021) assume the minimum sparse eigenvalue of the expected Gram matrix of *the optimal arm* when the instantaneous gap is greater than a constant $\Delta_*$, whose definition slightly differs from ours. Unlike previous works, we assume the *compatibility condition on the optimal arm without any constraints*. Under this assumption, a theoretical guarantee about the convergence of the Lasso estimator can be derived only if sufficient selections of the optimal arms are guaranteed, which necessitates more technical analysis. On the other hand, most of the previous work in sparse linear bandit that achieves poly-logarithmic regret under the margin condition implicitly assumes Assumption 3, indicating that *our assumptions are strictly weaker than others*.

---

**Algorithm 1** FS-WLasso (*Forced-Sampling then Weighted Loss Lasso*)

---

1: **Input:** Number of exploration $M_0$, Weight $w$, Regularization parameters $\{\lambda_t\}_{t \geq 1}$
2: **for** $t = 1, 2, ..., T$ **do**
3:      Observe $\{\mathbf{x}_{t,k}\}_{k=1}^{K}$
4:      **if** $t \leq M_0$ **then**                                              ▷ *Forced sampling stage*
5:          Choose $a_t \sim \text{Unif}(\mathcal{A})$ and observe $r_{t,a_t}$
6:      **else**                                                           ▷ *Greedy selection stage*
7:          Compute $\hat{\boldsymbol{\beta}}_{t-1} = \underset{\boldsymbol{\beta}}{\text{argmin}}\, w \sum_{i=1}^{M_0} (\mathbf{x}_{i,a_i}^{\top}\boldsymbol{\beta} - r_{i,a_i})^2 + \sum_{i=M_0+1}^{t-1} (\mathbf{x}_{i,a_i}^{\top}\boldsymbol{\beta} - r_{i,a_i})^2 + \lambda_{t-1}\|\boldsymbol{\beta}\|_1$
8:          Select $a_t = \text{argmax}_{k \in [K]}\, \mathbf{x}_{t,k}^{\top}\hat{\boldsymbol{\beta}}_{t-1}$ and observe $r_{t,a_t}$
9:      **end if**
10: **end for**

---

**Theorem 1.** *The compatibility condition on the optimal arm (Assumption 3) is strictly weaker than the assumptions made in previous Lasso bandit works under the single-parameter setting (Oh et al., 2021; Li et al., 2021; Ariu et al., 2022; Chakraborty et al., 2023), as illustrated in Figure 1.*

**Discussion of Theorem 1**     Oh et al. (2021); Ariu et al. (2022) assume the relaxed symmetry and balanced covariance of the context features, while other works in the literature, such as Li et al. (2021); Chakraborty et al. (2023) assume an anti-concentration condition for the feature vectors. These conditions imply that estimation error is reduced when data is obtained by a greedy policy, or, in some cases, by any policy. Since choosing the optimal arm is also a greedy policy with respect to the true parameter, the assumptions in prior works imply ours. The case where the context feature vectors of sub-optimal arms are fixed and only the feature vector of the optimal arm has randomness indicates that the converse does not hold. For a detailed proof of Theorem 1, refer to Appendix B.

**Remark 1.** *Under the multiple-parameter setting, Bastani & Bayati (2020); Wang et al. (2018) assume the compatibility condition on the feature vectors whose instantaneous gaps are lower-bounded by $h$. On the other hand, we impose no such constraint in Assumption 3 for the single-parameter setting. Further direct comparisons of the compatibility conditions are not possible since compatibility conditions do not translate directly across the two different settings. However, we show that Assumption 3 is weaker than those of Bastani & Bayati (2020); Wang et al. (2018) when compared within the problem instances that are convertible to both settings through a certain conversion (Kim & Paik, 2019; Oh et al., 2021). Refer to Appendix C for more details. It is important to note that we mainly compare our results with the Lasso bandit results under the single-parameter setting (Oh et al., 2021; Li et al., 2021; Ariu et al., 2022; Chakraborty et al., 2023), which is the predominant setup in linear contextual bandits (Abbasi-Yadkori et al., 2011; Abeille & Lazaric, 2017; Chakraborty et al., 2023; Filippi et al., 2010; Hao et al., 2020b; Kim & Paik, 2019; Li et al., 2021; Oh et al., 2021).*

## 3   Forced Sampling then Weighted Loss Lasso

### 3.1   Algorithm: FS-WLasso

In this section, we present FS-WLasso (*Forced Sampling then Weighted Loss Lasso*) that adapts the forced-sampling technique (Goldenshluger & Zeevi, 2013; Bastani & Bayati, 2020). FS-WLasso consists of two stages: *Forced sampling stage* & *Greedy selection stage*. First, during the *Forced sampling stage* the agent chooses an arm uniformly at random for $M_0$ rounds. Then, for $t$ in the *Greedy selection stage*, the agent computes the Lasso estimator given by

$$\hat{\boldsymbol{\beta}}_{t-1} = \underset{\boldsymbol{\beta}}{\text{argmin}}\, wL_0(\boldsymbol{\beta}) + L_{t-1}(\boldsymbol{\beta}) + \lambda_{t-1}\|\boldsymbol{\beta}\|_1\,, \tag{1}$$

where $L_0(\boldsymbol{\beta}) := \sum_{i=1}^{M_0}(\mathbf{x}_{i,a_i}^{\top}\boldsymbol{\beta} - r_{i,a_i})^2$ is the sum of squared errors over the samples acquired through random sampling, $L_{t-1}(\boldsymbol{\beta}) := \sum_{i=M_0+1}^{t-1}(\mathbf{x}_{i,a_i}^{\top}\boldsymbol{\beta} - r_{i,a_i})^2$ is the sum of squared errors over the samples observed in the *Greedy selection stage*, $w$ is the weight between the two loss functions,

and $\lambda_{t-1} > 0$ is the regularization parameter. The agent chooses the arm that maximizes the inner product of the feature vector and the Lasso estimator. `FS-WLasso` is summarized in Algorithm 1.

**Remark 2.** *Both `FS-WLasso` and ESTC (Hao et al., 2020b) have exploration stages, where the agent randomly selects arms for some initial rounds. However, the commit stages are very different. ESTC estimates the reward parameter only using the samples obtained during the exploration stage and does not update the parameters during the commit stage, whereas `FS-WLasso` continues to update the parameter using the samples obtained during the greedy selection stage. Therefore, our algorithm demonstrates superior statistical performance, achieving lower regret (and thus higher reward) by fully utilizing all accessible data.*

**Remark 3.** *The minimization problem* (1) *takes the sum of squared errors, whereas the standard Lasso estimator takes the average. While $\lambda_t$ is typically chosen to be proportional to $\sqrt{1/t}$ in the existing literature (Bastani & Bayati, 2020; Oh et al., 2021; Ariu et al., 2022; Li et al., 2021), this slight difference leads to $\lambda_t$ being proportional to $\sqrt{t}$ in Theorems 2 and 3.*

### 3.2 REGRET BOUND OF `FS-WLasso`

**Definition 1** (Compatibility constant ratio). *Let $\boldsymbol{\Sigma} := \frac{1}{K}\mathbb{E}[\sum_{k\in[K]} \mathbf{x}_{t,k}\mathbf{x}_{t,k}^\top]$ be the expected Gram matrix of the averaged arm. We define the constant $\rho := \phi_*^2/\phi^2(\boldsymbol{\Sigma}, S_0)$ as the ratio of the compatibility constant for $\boldsymbol{\Sigma}^*$ to the compatibility constant for $\boldsymbol{\Sigma}$.*

By the definition of $\boldsymbol{\Sigma}$, it holds that $\boldsymbol{\Sigma} = \frac{1}{K}\mathbb{E}[\mathbf{x}_{t,a_t^*}, \mathbf{x}_{t,a_t^*}^\top] + \frac{1}{K}\mathbb{E}[\sum_{k\neq a_t^*} \mathbf{x}_{t,k}\mathbf{x}_{t,k}^\top] \succeq \frac{1}{K}\mathbb{E}[\mathbf{x}_{t,a_t^*}, \mathbf{x}_{t,a_t^*}^\top]$, which implies $\phi^2(\boldsymbol{\Sigma}, S_0) \geq \phi^2(\boldsymbol{\Sigma}^*, S_0)/K \geq \phi_*^2/K > 0$. Hence, $\rho$ is well-defined with $0 < \rho \leq K$.

**Remark 4.** *When comparing the compatibility conditions only, the compatibility condition on the optimal arm implies the compatibility condition on the averaged arm. However, that is not the essence of what we compare between our work and the existing works. Note that under the margin condition, the entire set of stochastic context assumptions (e.g., the compatibility condition along with additional diversity assumptions) in the previous literature implies the compatibility condition on the optimal arm, as illustrated in Figure 1 and demonstrated in Appendix B.*

We present the regret upper bound of Algorithm 1. A formal version of the theorem and its proof are deferred to Appendix E.2.

**Theorem 2** (Regret Bound of `FS-WLasso`). *Suppose Assumptions 1-3 hold. For $\delta \in (0,1]$, let $\tau$ be a constant that depends on $x_{\max}, s_0, \phi_*, \sigma, \alpha, \Delta_*, \log d, \log \delta$. If we set the input parameters of Algorithm 1 by*

$$M_0 = \bar{C}_1 \max\left\{ \rho^2 x_{\max}^4 s_0^2 \phi_*^{-4} \log(d/\delta)\, , \rho^2\sigma^2 x_{\max}^{4+\frac{4}{\alpha}} s_0^{2+\frac{2}{\alpha}} \Delta_*^{-2}\phi_*^{-4-\frac{4}{\alpha}} \left(\log\log\tau + \log(d/\delta)\right) \right\},$$

$$\lambda_t = \bar{C}_2\sigma x_{\max}\left( \sqrt{(t-M_0)\log\left(d(\log(t-M_0))^2/\delta\right)} + \sqrt{w^2 M_0 \log(d/\delta)}\right), w = \sqrt{\tau/M_0}\,,$$

*for some universal constants $\bar{C}_1, \bar{C}_2 > 0$, then with probability at least $1 - \delta$, Algorithm 1 achieves the following cumulative regret:*

$$R(T) \leq 2x_{\max}bM_0 + I_\tau + I_T\,,$$

*where $I_\tau = \mathcal{O}\left(\sigma^2\Delta_*^{-1}\left(x_{\max}^2 s_0/\phi_*^2\right)^{1+\frac{1}{\alpha}}\log(d/\delta)\right)$ and*

$$I_T = \begin{cases} \mathcal{O}\left(\frac{(\sigma x_{\max}^2 s_0/\phi_*^2)^{1+\alpha}}{\Delta_*^\alpha(1-\alpha)}T^{\frac{1-\alpha}{2}}\left(\log d + \log\frac{\log T}{\delta}\right)^{\frac{1+\alpha}{2}}\right) & \text{for } \alpha \in (0,1)\,, \\[2ex] \mathcal{O}\left(\frac{(\sigma x_{\max}^2 s_0/\phi_*^2)^2}{\Delta_*}\log T\left(\log d + \log\frac{\log T}{\delta}\right)\right) & \text{for } \alpha = 1\,, \\[2ex] \mathcal{O}\left(\frac{\alpha}{(\alpha-1)^2}\cdot\frac{\sigma^2\left(x_{\max}^2 s_0/\phi_*^2\right)^{1+\frac{1}{\alpha}}}{\Delta_*}\left(\log d + \log\frac{1}{\delta}\right)\right) & \text{for } 1 < \alpha \leq \infty\,. \end{cases}$$

**Discussion of Theorem 2** In terms of key problem instances ($s_0, d$, and $T$), Theorem 2 establishes the regret bounds that scale poly-logarithmically on $d$ and $T$, specifically, $\mathcal{O}(s_0^{\alpha+1}T^{\frac{1-\alpha}{2}}(\log d +$

$\log \log T)^{\frac{\alpha+1}{2}}$) for $\alpha \in (0, 1)$, $\mathcal{O}(s_0^2 \log T(\log d + \log \log T))$ for $\alpha = 1$, and $\mathcal{O}(s_0^{2+\frac{2}{\alpha}} \log d)$ for $\alpha > 1$. Li et al. (2021) construct a regret lower bound of $\mathcal{O}(T^{\frac{1-\alpha}{2}} (\log d)^{\frac{\alpha+1}{2}} + \log T)$ when $\alpha \in [0, 1]$, which our algorithm achieves up to a $\log T$ factor. The expected regret for Algorithm 1 can also be obtained by taking $\delta = 1/T$. For the $T$-agnostic setting, we derive `FS-Lasso`, which uses forced samples adaptively, and establish the same regret bound as in Theorem 2 (Appendix F). Existing Lasso bandit literature that achieves $\mathcal{O}(\text{poly} \log dT)$ regret under the single parameter setting necessitates stronger assumptions on the context distribution (e.g., relaxed symmetry & balanced covariance or anti-concentration), which are non-verifiable in practical scenarios. In addition, when context distributions do not satisfy the strong assumptions employed in the previous literature, the existing algorithms can critically undermine regret performance, with no recourse for adjustment nor guarantees provided. That is, there is nothing one can do when such strong context assumptions are not satisfied in the existing literature. However, we show that the compatibility condition on the optimal arm is sufficient to achieve poly-logarithmic regret under the margin condition, and demonstrate that our assumption is strictly weaker than those used in other Lasso bandit literature under the single-parameter setting.

Our result also improves the known regret bound for the low-dimensional setting, where $s_0$ may be replaced with $d$. In this case, Assumption 3 becomes equivalent to the HLS condition (Hao et al., 2020a; Papini et al., 2021). Under the HLS condition and the minimum gap condition, Papini et al. (2021) show that `OFUL` (Abbasi-Yadkori et al., 2011) achieves a constant regret bound independent of $T$ with high probability (Lemma 2 in (Papini et al., 2021)). However, when the margin condition (Assumption 2) is assumed, the result of Papini et al. (2021) guarantees $O(\log T)$ regret bound only when $\alpha > 2$. Our algorithm achieves a constant regret bound with high probability when $\alpha > 1$, expanding the range of $\alpha$ that the constant regret is attainable.

**Remark 5.** *Theorem 2 requires the value of $s_0$ when determining $M_0$, the length of the forced sampling stage. On the other hand, there are sparsity-agnostic Lasso bandit algorithms (Oh et al., 2021; Ariu et al., 2022; Chakraborty et al., 2023). However, these sparsity-agnostic algorithms require stronger diversity assumptions on the context distribution that are not verifiable in practice. Even when the sparsity is known, other works in the literature still either incorporate extra stochastic conditions (Li et al., 2021) or apply specific optimality criteria for context distributions (Bastani & Bayati, 2020; Wang et al., 2018). As discussed earlier, these additional assumptions may pose obstacles in practical applications. Regardless of sparsity-awareness, our work focuses on alleviating these stringent stochastic assumptions on context distributions, providing the weakest conditions known to achieve poly-logarithmic regret. Furthermore, by tuning $M_0$ as a whole, not knowing the sparsity does not worsen the complexity nor the performance of the algorithm in practice. $M_0$ does not solely depend on $s_0$, but also on other problem-dependent factors that may be unknown to the algorithm in practice and hence $M_0$ should be regarded as a tunable parameter. Note that all Lasso bandit including the sparsity agnostic ones and parametric bandit algorithms also have parameters that must be tuned in practice, such as the ones that depend on the sub-Gaussian parameter of the noise $\sigma$. Refer to Appendix D for more details.*

In most regret analyses of sparse linear bandit algorithms under the single-parameter setting (Kim & Paik, 2019; Li et al., 2021; Oh et al., 2021; Ariu et al., 2022; Chakraborty et al., 2023), the maximum regret is incurred during the *burn-in* phase, where the compatibility condition of the empirical Gram matrix is not guaranteed. The compatibility condition after the burn-in phase is ensured by additional diversity assumptions on context features (e.g., anti-concentration (Li et al., 2021; Chakraborty et al., 2023), relaxed symmetry & balanced covariance (Oh et al., 2021; Ariu et al., 2022)), rather than by explicit exploration within the algorithms. Therefore, the Lasso estimator calculation (Oh et al., 2021; Ariu et al., 2022) or explicit exploration (UCB in Li et al. (2021) or TS in Chakraborty et al. (2023)) during their burn-in phases does not contribute to the regret bound.

On the other hand, our forced sampling stage does not compute parameters but acquires diverse samples without requiring diversity assumptions on context features beyond the compatibility condition on the optimal arm, making it more efficient during the burn-in phases. If additional diversity assumptions (Li et al., 2021; Oh et al., 2021; Ariu et al., 2022; Chakraborty et al., 2023) are also applied to our algorithm, we show that $\mathcal{O}(\text{poly} \log T)$ regret is achieved without the forced sampling stage in Algorithm 1.

**Theorem 3.** *Suppose that Assumptions 1-3 hold, and further assume either the anti-concentration (Assumption 4) or relaxed symmetry & balanced covariance (Assumption 6-8) assumptions. Let $\phi_G$ be an appropriate constant that is determined by the employed assumptions, and $\tau$ be a constant*

*that depends on $\sigma$, $x_{\max}$, $s_0$, $\Delta_*$, $\phi_*$, $\phi_G$, $\alpha$, $\log d$, and $\log \delta$. If we set the input parameters of Algorithm 1 by $M_0 = 0$, i.e. no forced-sampling stage, and $\lambda_t = \bar{C}_2 \sigma x_{\max} \sqrt{t \log (d(\log t)^2/\delta)}$, where $\bar{C}_2$ is the same universal constant as in Theorem 2, then with probability at least $1 - \delta$, Algorithm 1 achieves the following cumulative regret with probability at least $1 - \delta$:*

$$R(T) \leq \begin{cases} I_b + I_2(T) & T \leq \tau \\ I_b + I_2(\tau) + I_T & T > \tau, \end{cases}$$

*where*

$$I_b = \mathcal{O}\left( x_{\max}^5 b s_0^2 \phi_G^{-4} \left( \log(x_{\max} s_0 \phi_G^{-1}) + \log d - \log \delta \right) \right),$$

$$I_2(T) = \begin{cases} \mathcal{O}\left( \frac{(\sigma x_{\max}^2 s_0/\phi_G^2)^{1+\alpha}}{\Delta_*^{\alpha}(1-\alpha)} T^{\frac{1-\alpha}{2}} \left( \log d + \log \frac{\log T}{\delta} \right)^{\frac{1+\alpha}{2}} \right) & \text{for } \alpha \in [0,1), \\ \mathcal{O}\left( \frac{(\sigma x_{\max}^2 s_0/\phi_G^2)^2}{\Delta_*} \log T \left( \log d + \log \frac{\log T}{\delta} \right) \right) & \text{for } \alpha = 1, \\ \mathcal{O}\left( \frac{\alpha^2}{(\alpha-1)^2} \cdot \frac{(\sigma x_{\max}^2 s_0/\phi_G^2)^2}{\Delta_*} \left( \log d + \log \frac{1}{\delta} \right) \right) & \text{for } 1 < \alpha \leq \infty, \end{cases}$$

*and $I_T$ takes the same value as in Theorem 2.*

**Discussion of Theorem 3** Theorem 3 offers that random exploration of Algorithm 1 may not be necessary if the additional diversity assumptions on context features are given. This result indicates that the number of exploration may be tuned according to the specific problem instance. The assumptions of the Theorem 3 are still weaker than, or equally strong as Oh et al. (2021); Li et al. (2021); Chakraborty et al. (2023), while the regret bounds are no greater than theirs. We slightly improve the regret bound of Li et al. (2021) when $1 < \alpha \leq \infty$. Specifically, a term proportional to $s_0^2/(\Delta_* \phi_*^4)$ in Li et al. (2021) is sharpened to $s_0^{1+\frac{1}{\alpha}}/(\Delta_* \phi_*^{2+\frac{2}{\alpha}})$ in our result. We also achieve a tighter regret bound than Chakraborty et al. (2023), which is proportional to $K^4$. Our result is proportional to at most $K^2$ since $\phi_*^2 \geq \Omega(\frac{1}{K})$ holds under their assumptions, as shown in Lemma 1.

### 3.3 TECHNICAL CHALLENGES AND SKETCH OF PROOFS

Under Assumption 3, a small estimation error of $\hat{\beta}_t$ is ensured when the optimal arms have been chosen a sufficient number of times. Specifically, if the optimal arms have been selected sufficiently many times up to round $t$, it ensures the compatibility constant of the empirical Gram matrix is $\Omega(\phi_*^2 t)$. Then, the Lasso estimation error can be controlled via the oracle inequality for the weighted squared Lasso estimator (Lemma 19). A well-estimated estimator, in turn, leads to the selection of the optimal arm in the next round. This observation highlights the cyclic relationship between estimation error and the selection of optimal arms. However, this is not a case of circular reasoning; rather, it is a domino-like phenomenon that propagates forward in time.

On the other hand, such cyclic structure has not been observed in previous Lasso bandit literature (Bastani & Bayati, 2020; Oh et al., 2021; Li et al., 2021; Ariu et al., 2022; Chakraborty et al., 2023). This is because existing methods rely on diversity assumptions on the context distribution, which ensure that samples obtained by the agent's policy automatically explore the feature space, resulting in a positive compatibility constant for the empirical Gram matrix regardless of the previously selected arms. However, since such convenience is no longer available in our setting, we meticulously analyze the cyclic structure between the estimation error and the selection of optimal arms by deriving a novel mathematical induction argument.

There are three main difficulties that lie in the way of constructing the induction argument. First, the initial condition of the induction must be satisfied, in other words, the cycle must begin. We guarantee the initial condition through random exploration (Theorem 2) or additional diversity assumptions (Theorem 3). We show that after the initial stages, the algorithm attains a sufficiently accurate estimator, which starts the cycle. Second, the algorithm must be able to propagate such favorable events to the next round. A small estimation error does not always guarantee the selection of the optimal arm. Instead, we show that it leads to a bounded ratio of sub-optimal selections over time. The compatibility condition on the optimal arm implies that if the optimal arms constitute a large

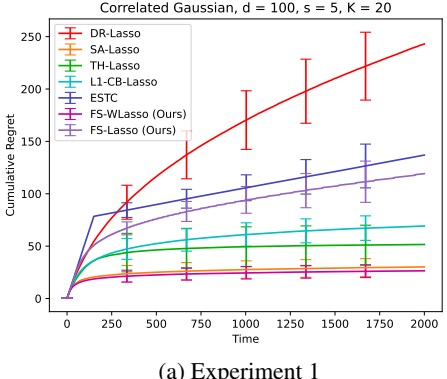 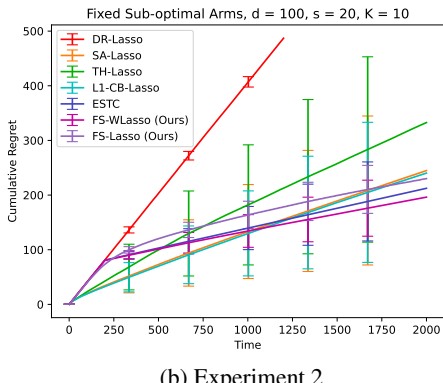

(a) Experiment 1            (b) Experiment 2

Figure 2: The evaluations of Lasso bandit algorithms are presented. Figure 2a shows results where all context feature vectors are sampled from a correlated Gaussian distribution. Figure 2b shows results where the context feature vectors of sub-optimal arms are fixed throughout time, and only the feature vector of the optimal arm has randomness. We plot the mean and standard deviation of cumulative regret across 100 runs for each algorithm.

portion of the observed data, the algorithm attains a small estimation error. We build an induction argument upon these relationships. Lastly, due to the stochastic nature of the problem, the algorithm suffers a small probability of failing to propagate the good events in every round. Without careful analysis, the sum of such probabilities easily exceeds 1, invalidating the whole proof. We bound the sum to be small by carefully constructing high-probability events that occur independently of the induction argument, then prove that the induction argument always holds under the events. The complete proof is illustrated in Appendix E.

## 4 NUMERICAL EXPERIMENTS

We perform numerical evaluations on synthetic datasets. We compare our algorithms, `FS-WLasso` and `FS-Lasso`, with sparse linear bandit algorithms including `DR Lasso Bandit` (Kim & Paik, 2019), `SA Lasso BANDIT` (Oh et al., 2021), `TH Lasso Bandit` (Ariu et al., 2022), $\ell_1$-Confidence Ball Based Algorithm (`L1-CB-Lasso`) (Li et al., 2021), and `ESTC` (Hao et al., 2020b). We plot the mean and standard deviation of cumulative regret across 100 runs for each algorithm.

The results clearly demonstrate that our proposed algorithms outperform the existing sparse linear bandit methods we evaluated. In particular, even in cases where the context features of all arms, except for the optimal arm, are fixed (rendering assumptions such as anti-concentration invalid), our proposed algorithms surpass the performance of existing ones. More details are presented in Appendix I.

## 5 CONCLUSION

In this work, we study the stochastic context conditions under which the Lasso bandit algorithm can achieve a poly-logarithmic regret. We present rigorous comparisons on the relative strengths of the conditions utilized in the sparse linear bandit literature, which provide insights that can be of independent interest. Our regret analysis shows that the proposed algorithms establish a poly-logarithmic dependency on the feature dimension and time horizon.

## 6 REPRODUCIBILITY STATEMENT

For each theoretical result, we provide the full set of assumptions in the main paper (Section 2.1), and the complete proofs of the main results are provided in Appendix E and F. We have also included the data and code, along with instructions to reproduce the main experimental results, in the supplementary material.

## ACKNOWLEDGEMENTS

This work was supported by the National Research Foundation of Korea (NRF) grant funded by the Korea government (MSIT) (No. RS-2022-NR071853 and RS-2023-00222663) and by AI-Bio Research Grant through Seoul National University.

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

# Appendix

## Table of Contents

## A  NOTATIONS AND DEFINITIONS

We introduce notations that are necessary for the analysis.

**Linear Bandit**

- $\boldsymbol{\beta}^* \in \mathbb{R}^d$: True reward parameter
- $\mathbf{x}_{t,k} \in \mathbb{R}^d$: Context feature vector in round $t$, arm $k$
- $\mathcal{X}$: Set of all possible context feature vectors
- $\mathcal{D}_\mathcal{X}$: Distribution of context vectors tuple $\{\mathbf{x}_{t,k}\}_{k=1}^K$
- $a_t$: Chosen arm in round $t$
- $a_t^*$: Optimal arm in round $t$
- $\eta_t$: Zero-mean sub-Gaussian noise in round $t$
- $\sigma$: Variance proxy of $\eta_t$
- $r_{t,a_t} = \mathbf{x}_{t,a_t}^\top \boldsymbol{\beta}^* + \eta_t$: Observed reward in round $t$

- $\text{reg}_t = \mathbf{x}_{t,a_t^*}^\top \boldsymbol{\beta}^* - \mathbf{x}_{t,a_t}^\top \boldsymbol{\beta}^*$: Instantaneous regret in round $t$
- $d$: Dimension of feature and true parameter vectors
- $K$: Number of arms
- $T$: Time horizon

## High-Dimensional Statistics

- $S_0 := \left\{ j \in [d] : (\boldsymbol{\beta}^*)_j \neq 0 \right\}$: Active set
- $s_0 := |S_0|$ Sparsity index
- $v_{j,S_0} := v_j \mathbb{1}\left\{ j \in S_0 \right\}$
- $\mathbf{v}_{S_0} := [v_{1,S_0}, \ldots, v_{d,S_0}]^\top$
- $\mathbf{v}_{S_0^c} = \mathbf{v}_{[d]\setminus S_0}$
- $\mathbb{C}(S_0) = \left\{ \mathbf{v} \in \mathbb{R}^d : \|\mathbf{v}_{S_0^c}\|_1 \leq 3\|\mathbf{v}_{S_0}\|_1 \right\}$
- $\phi^2 (\mathbf{M}, S_0)$: Compatibility constant of matrix $\mathbf{M}$ over set $S_0$

Note that the definition of compatibility constant in Assumption 3 can be rewritten as $\phi^2(\mathbf{M}, I) = \inf_{\mathbf{v} \in \mathbb{C}(I)\setminus\{\mathbf{0}_d\}} \frac{s_0 \mathbf{v}^\top \mathbf{M} \mathbf{v}}{\|\mathbf{v}_I\|_1^2}$.

## Assumptions

- $x_{\max}$: $\ell_\infty$-norm upper bound of $\mathbf{x} \in \mathcal{X}$
- $b$: $\ell_1$-norm upper bound of $\boldsymbol{\beta}^*$
- $\Delta_t := \max_{a \neq a_t^*} \mathbf{x}_{t,a_t^*}^\top \boldsymbol{\beta}^* - \mathbf{x}_{t,a}^\top \boldsymbol{\beta}^*$: Instantaneous gap
- $\Delta_*$: Margin constant, or relaxed minimum gap
- $\alpha$: Margin condition parameter
- $\mathbf{x}_*$: Optimal arm feature as random vector
- $\boldsymbol{\Sigma}^* := \mathbb{E}\left[\mathbf{x}_* \mathbf{x}_*^\top\right]$: Expected Gram matrix of optimal arm
- $\phi_*$: Lower bound of $\phi^2 (\boldsymbol{\Sigma}^*, S_0)$

## Algorithm

- $M_0$: Number of random exploration rounds
- $w$: Weight between square errors of random samples and greedy samples
- $\lambda_t$: Lasso regularization parameter
- $\hat{\boldsymbol{\beta}}_t$: Lasso estimate of $\boldsymbol{\beta}^*$

## Analysis

- $\delta$: Probability of failure
- $\boldsymbol{\Sigma} := \frac{1}{K}\mathbb{E}\left[\sum_{k=1}^K \mathbf{x}_{t,k}\mathbf{x}_{t,k}^\top\right]$: Theoretical Gram matrix of all arms
- $\boldsymbol{\Sigma}_\Gamma^* := \mathbb{E}\left[\mathbf{x}_* \mathbf{x}_* \mid \Delta_t > \Delta_*\right]$: Theoretical Gram matrix of optimal arm with large gap
- $\boldsymbol{\Sigma}_k := \mathbb{E}\left[\mathbf{x}_{t,k}\mathbf{x}_{t,k}^\top\right]$: Theoretical Gram matrix of arm $k$
- $\rho$: Compatibility constant ratio
- $\hat{\mathbf{V}}_{M_0+\tau} := \sum_{t=1}^{M_0} w\mathbf{x}_{t,a_t}\mathbf{x}_{t,a_t}^\top + \sum_{t=M_0+1}^{M_0+\tau} \mathbf{x}_{t,a_t}\mathbf{x}_{t,a_t}^\top$: (Weighted) Empirical Gram matrix
- $N_{\tau_1}(t')$: Number of sub-optimal selections during $t = M_0 + \tau_1 + 1$ to $M_0 + \tau_1 + t'$
- $\overline{\Delta}_t$: Upper bound of $2x_{\max}\|\boldsymbol{\beta}^* - \hat{\boldsymbol{\beta}}_t\|_1$
- $\mathcal{F}_t$: $\sigma$-algebra generated by $\{\mathbf{x}_{\tau,i}\}_{\tau \in [t], i \in [K]}, \{a_\tau\}_{\tau \in [t]}, \{r_{\tau,a_\tau}\}_{\tau \in [t-1]}$
- $\mathcal{F}_t^+$: $\sigma$-algebra generated by $\{\mathbf{x}_{\tau,i}\}_{\tau \in [t], i \in [K]}, \{a_\tau\}_{\tau \in [t]}, \{r_{\tau,a_\tau}\}_{\tau \in [t]}$

**Generic notations**

- $\mathbb{N}_0 = \mathbb{N} \cup \{0\}$
- $[N] := \{1, 2, \ldots, N\}$: Set of natural numbers up to $N$
- $\mathbb{R}_{\geq 0}$: Set of non-negative real numbers
- $\mathbb{1}$: Indicator function
- $\|\cdot\|_0$: $\ell_0$-norm of a vector, i.e. number of non-zero elements
- $\|\cdot\|_2$: $\ell_2$-norm of a vector
- $\|\cdot\|_\infty$: $\ell_\infty$-norm of a vector or a matrix, i.e., maximum of absolute values of elements
- $(\cdot)_j$: $j$-th element of a vector
- $(\cdot)_{ij}$: $ij$-th element of a matrix
- $\mathbf{0}_d$: Zero vector in $\mathbb{R}^d$
- $\mathbf{I}_d$: Identity matrix in $\mathbb{R}^{d \times d}$
- $(\Omega, \mathcal{F}, \mathbb{P})$: Probability space

# B    DISCUSSION ON ASSUMPTION 3 AND PROOF OF THEOREM 1

We introduce some of the assumptions made in related works about sparse linear bandit. We show that these assumptions imply Assumption 3, proving that our assumptions are strictly weaker than others.

**Assumption 4** (Anti-concentration (Li et al., 2021; Chakraborty et al., 2023))**.** *There exists a positive constant $\xi$ such that for each $k \in [K]$, $t \in [T]$, $\mathbf{v} \in \left\{\mathbf{u} \in \mathbb{R}^d \mid \|\mathbf{u}\|_0 \leq C_d\right\}$, and $h > 0$, $\mathbb{P}((\mathbf{x}_{t,k}^\top \mathbf{v})^2 \leq h\|\mathbf{v}\|_2^2) \leq \xi h$. $C_d$ equals $d$ in Li et al. (2021) and is a big enough constant that depends on $\xi$, $K$, $s_0$, and more in Chakraborty et al. (2023).*

**Assumption 5** (Sparse eigenvalue of the optimal arm (Li et al., 2021))**.** *Let $\Gamma = \left\{\omega \in \Omega : \Delta_t \geq 2^{-\frac{1}{\alpha}}\Delta_*\right\}$ be the event that the instantaneous gap is large enough, and $\boldsymbol{\Sigma}_\Gamma^* = \mathbb{E}\left[\mathbf{x}_t^* \mathbf{x}_t^{*\top} \mid \Gamma\right]$ be the expected Gram matrix of the optimal arm conditioned on the event $\Gamma$. Then, there exists a constant $\phi_1 > 0$ such that*

$$\inf_{\substack{\mathbf{v} \in \mathbb{R}^d \setminus \{\mathbf{0}_d\} \\ \|\mathbf{v}\|_0 \leq C^* s_0 + 1}} \frac{\mathbf{v}^\top \boldsymbol{\Sigma}_\Gamma^* \mathbf{v}}{\|\mathbf{v}\|_2^2} \geq \phi_1^2,$$

*where $C^*$ is a big enough constant that depends on $\xi$ (in Assumption 4), $K$, and more.*

**Assumption 6** (Compatibility condition on the averaged arm (Oh et al., 2021; Ariu et al., 2022))**.** *Let $\boldsymbol{\Sigma} = \mathbb{E}_{\{\mathbf{x}_{t,k}\}_{k=1}^K \sim \mathcal{D}_{\mathcal{X}}}\left[\frac{1}{K}\sum_{k=1}^K \mathbf{x}_{t,k}\mathbf{x}_{t,k}^\top\right]$ be the expected Gram matrix of the averaged arm. Then, there exists a constant $\phi_2 > 0$ such that $\phi^2(\boldsymbol{\Sigma}, S_0) \geq \phi_2$.*

**Assumption 7** (Relaxed symmetry (Oh et al., 2021; Ariu et al., 2022))**.** *For the context distribution $\mathcal{P}_{\mathcal{X}}$, there exists a constant $1 \leq \nu < \infty$ such that $0 < \frac{\mathcal{P}_{\mathcal{X}}(-\mathbf{x})}{\mathcal{P}_{\mathcal{X}}(\mathbf{x})} \leq \nu$ for any $\mathbf{x} \in \mathcal{X}$ with $\mathcal{P}_{\mathcal{X}}(\mathbf{x}) \neq 0$.*

**Assumption 8** (Balanced covariance (Oh et al., 2021; Ariu et al., 2022))**.** *There exists $0 < C_{\mathcal{X}} < \infty$ such that for any permutation $(i_1, \ldots, i_K)$ of $(1, \ldots, K)$, any $k \in \{2, \ldots, K-1\}$, and any fixed $\boldsymbol{\beta} \in \mathbb{R}^d$, it holds that*

$$\mathbb{E}\left[\mathbf{x}_{i_k}\mathbf{x}_{i_k}^\top \mathbb{1}\{\mathbf{x}_{i_1}^\top\boldsymbol{\beta} < \ldots < \mathbf{x}_{i_K}^\top\boldsymbol{\beta}\}\right] \preceq C_{\mathcal{X}}\mathbb{E}\left[(\mathbf{x}_{i_1}\mathbf{x}_{i_1}^\top + \mathbf{x}_{i_K}\mathbf{x}_{i_K}^\top)\mathbb{1}\{\mathbf{x}_{i_1}^\top\boldsymbol{\beta} < \ldots < \mathbf{x}_{i_K}^\top\boldsymbol{\beta}\}\right].$$

We show that some of the assumptions imply the following property, which we name *the greedy diversity*.

**Definition 2** (Greedy diversity)**.** *For any fixed $\boldsymbol{\beta} \in \mathbb{R}^d$, define the greedy policy with respect to an estimator $\boldsymbol{\beta}$ as $\pi_{\boldsymbol{\beta}}\left(\{\mathbf{x}_k\}_{k=1}^K\right) = \text{argmax}_{k \in [K]} \mathbf{x}_k^\top\boldsymbol{\beta}$. Denote the chosen feature vector with*

respect to the greedy policy by $\mathbf{x}_{\boldsymbol{\beta}} = \mathbf{x}_{\pi_{\boldsymbol{\beta}}\left(\{\mathbf{x}_k\}_{k=1}^K\right)}$. *The context distribution* $\mathcal{D}_{\mathcal{X}}$ *satisfies the greedy diversity if there exists a constant* $\phi_G > 0$ *such that for any* $\boldsymbol{\beta} \in \mathbb{R}^d$,

$$\phi^2 \left( \mathbb{E}_{\{\mathbf{x}_k\}_{k=1}^K \sim \mathcal{D}_{\mathcal{X}}} \left[ \mathbf{x}_{\boldsymbol{\beta}} \mathbf{x}_{\boldsymbol{\beta}}^\top \right], S_0 \right) \geq \phi_G^2.$$

**Remark 6.** *Note that* $\mathbf{x}_{\boldsymbol{\beta}^*} = \mathbf{x}_*$. *Under the greedy diversity, Assumption 3 holds with* $\phi_* = \phi_G$ *by plugging in* $\boldsymbol{\beta} = \boldsymbol{\beta}^*$. *Therefore, the greedy diversity implies the compatibility condition on the optimal arm.*

**Anti-concentration to ours:**

The following lemma shows that anti-concentration implies the greedy diversity, hence it implies Assumption 3. While Li et al. (2021); Chakraborty et al. (2023) use $\epsilon$-net argument to ensure the compatibility condition of the empirical Gram matrix, we follow a slightly different approach to ensure the compatibility condition of the expected Gram matrix. Another point to note is that Li et al. (2021); Chakraborty et al. (2023) employ additional assumptions, such as sub-Gaussianity of feature vectors and maximum sparse eigenvalue condition, to upper bound the diagonal elements of the empirical Gram matrix. To make the analysis simpler, we replace the upper bound by $x_{\max}^2$.

**Lemma 1.** *If Assumption 4 holds with* $C_d \geq 64 x_{\max}^2 \xi K s_0 + 1$, *then the greedy diversity is satisfied with* $\phi_G^2 \geq \frac{1}{4\xi K}$.

*Proof of Lemma 1.* We first show that $\mathbb{E}\left[\mathbf{x}_{\boldsymbol{\beta}}\mathbf{x}_{\boldsymbol{\beta}}^\top\right]$ has a a positive minimum sparse eigenvalue, then use the Transfer principle (Lemma 31) adopted in Li et al. (2021); Chakraborty et al. (2023). Let $\mathbf{v} \in \mathbb{R}^d$ be a vector with $\|\mathbf{v}\|_2 = 1$ and $\|\mathbf{v}\|_0 \leq C_d$. For a fixed value of $h \geq 0$, $\left(\mathbf{x}_{\boldsymbol{\beta}}^\top \mathbf{v}\right)^2 \leq h$ implies that there exists at least one $k \in [K]$ such that $(\mathbf{x}_k^\top \mathbf{v})^2 \leq h$ holds. Then, we infer that

$$\mathbb{P}\left(\left(\mathbf{x}_{\boldsymbol{\beta}}^\top \mathbf{v}\right)^2 \leq h\right) \leq \mathbb{P}\left(\exists k \in [K] : (\mathbf{x}_k^\top \mathbf{v})^2 \leq h\right)$$
$$\leq \sum_{k=1}^K \mathbb{P}\left((\mathbf{x}_k^\top \mathbf{v})^2 \leq h\right)$$
$$\leq \xi K h,$$

where the second inequality is the union bound, and the last inequality is from Assumption 4. Then, using that $\left(\mathbf{x}_{\boldsymbol{\beta}}^\top \mathbf{v}\right)^2 = \mathbf{v}^\top \left(\mathbf{x}_{\boldsymbol{\beta}} \mathbf{x}_{\boldsymbol{\beta}}^\top\right) \mathbf{v}$, we bound the minimum sparse eigenvalue of the expected Gram matrix.

$$\mathbb{E}\left[\mathbf{v}^\top \left(\mathbf{x}_{\boldsymbol{\beta}}\mathbf{x}_{\boldsymbol{\beta}}^\top\right)\mathbf{v}\right] = \int_0^\infty \mathbb{P}\left(\mathbf{v}^\top \left(\mathbf{x}_{\boldsymbol{\beta}}\mathbf{x}_{\boldsymbol{\beta}}^\top\right)\mathbf{v} \geq x\right) dx$$
$$\geq \int_0^{\frac{1}{\xi K}} \mathbb{P}\left(\mathbf{v}^\top \left(\mathbf{x}_{\boldsymbol{\beta}}\mathbf{x}_{\boldsymbol{\beta}}^\top\right)\mathbf{v} \geq x\right) dx$$
$$\geq \int_0^{\frac{1}{\xi K}} (1 - \xi K x)\, dx$$
$$= \frac{1}{2\xi K}. \tag{2}$$

Now, we use the Transfer principle. Let $\hat{\boldsymbol{\Sigma}} = \mathbb{E}\left[\mathbf{x}_{\boldsymbol{\beta}}\mathbf{x}_{\boldsymbol{\beta}}^\top\right]$ and $\bar{\boldsymbol{\Sigma}} = \frac{1}{\xi K}\mathbf{I}_d$. Inequality (2) shows that for $\|\mathbf{v}\|_0 \leq C_d$, it holds that

$$\mathbf{v}^\top \hat{\boldsymbol{\Sigma}} \mathbf{v} \geq \frac{1}{2}\mathbf{v}^\top \bar{\boldsymbol{\Sigma}} \mathbf{v}.$$

For any $j \in [d]$, we have $\hat{\boldsymbol{\Sigma}}_{jj} = \mathbb{E}\left[(\mathbf{x}_{\boldsymbol{\beta}})_j^2\right] \leq x_{\max}^2$. Then, the conditions of Lemma 31 hold with $\eta = \frac{1}{2}$, $\mathbf{D} = x_{\max}^2 \mathbf{I}_d$, and $m = C_d$. Suppose $\mathbf{u} \in \mathbb{C}(S_0)$. By Lemma 31, we have

$$\mathbf{u}^\top \mathbb{E}\left[\mathbf{x}_{\boldsymbol{\beta}}\mathbf{x}_{\boldsymbol{\beta}}^\top\right]\mathbf{u} \geq \frac{1}{2\xi K}\|\mathbf{u}\|_2^2 - \frac{\left\|\mathbf{D}^{\frac{1}{2}}\mathbf{u}\right\|_1^2}{C_d - 1}. \tag{3}$$

The first term is lower bounded as the following:

$$\frac{1}{2\xi K} \|\mathbf{u}\|_2^2 \geq \frac{1}{2\xi K} \|\mathbf{u}_{S_0}\|_2^2$$
$$\geq \frac{1}{2\xi K s_0} \|\mathbf{u}_{S_0}\|_1^2 , \tag{4}$$

where the second inequality is the Cauchy-Schwarz inequality. The second term is upper bounded as the following:

$$\frac{\left\|\mathbf{D}^{\frac{1}{2}}\mathbf{u}\right\|_1^2}{C_d - 1} = \frac{\|x_{\max}\mathbf{u}\|_1^2}{64x_{\max}^2\xi K s_0} = \frac{\|\mathbf{u}\|_1^2}{64\xi K s_0} \leq \frac{\|\mathbf{u}_{S_0}\|_1^2}{4\xi K s_0} , \tag{5}$$

where the inequality holds by $\|\mathbf{u}\|_1 = \|\mathbf{u}_{S_0}\|_1 + \|\mathbf{u}_{S_0^c}\|_1 \leq 4\|\mathbf{u}_{S_0}\|_1$ when $\mathbf{u} \in \mathbb{C}(S_0)$. Putting inequalities (3), (4), and (5) together, we obtain

$$\mathbf{u}^\top \mathbb{E}\left[\mathbf{x}_\beta \mathbf{x}_\beta^\top\right]\mathbf{u} \geq \frac{\|\mathbf{u}_{S_0}\|_1^2}{4\xi K s_0} ,$$

which implies $\phi^2(\mathbb{E}\left[\mathbf{x}_\beta \mathbf{x}_\beta^\top\right], S_0) \geq \frac{1}{4\xi K}$.  $\qquad\square$

**Sparse eigenvalue to ours :**

Assumption 5 does not imply the greedy diversity, but still implies the compatibility condition on the optimal arm. As in the previous subsection, we replace the upper bound of the diagonal entries of the Gram matrix obtained in Li et al. (2021) with $x_{\max}^2$ for simpler analysis.

**Lemma 2.** *Suppose Assumptions 2, 4, and 5 hold with $C^* = 64x_{\max}^2\xi K$. Then, Assumption 3 holds with $\phi_*^2 \geq \frac{\phi_1^2}{3}$.*

*Proof of Lemma 2.* Lemma 1 shows that Assumption 4 implies compatibility condition on the optimal arm with $\phi_*^2 \geq \frac{1}{4\xi K}$. If $\frac{\phi_1^2}{3} \leq \frac{1}{4\xi K}$, then the proof is complete. Suppose $\frac{\phi_1^2}{3} \geq \frac{1}{4\xi K}$.

By the margin condition, the probability of the event $\Gamma$ is at least $\mathbb{P}(\Gamma) = 1 - \mathbb{P}\left(\Delta_t < 2^{-\frac{1}{\alpha}}\Delta_*\right) \geq 1 - \left(2^{-\frac{1}{\alpha}}\right)^\alpha = \frac{1}{2}$. Then, we have

$$\phi^2\left(\mathbf{\Sigma}^*, S_0\right) = \phi^2\left(\mathbb{E}\left[\mathbf{x}_*\mathbf{x}_*^\top \mathbb{1}\left\{\Gamma\right\}\right] + \mathbb{E}\left[\mathbf{x}_*\mathbf{x}_*^\top \mathbb{1}\left\{\Gamma^c\right\}\right], S_0\right)$$
$$\geq \phi^2\left(\mathbb{E}\left[\mathbf{x}_*\mathbf{x}_*^\top \mathbb{1}\left\{\Gamma\right\}\right], S_0\right)$$
$$= \phi^2\left(\mathbb{E}\left[\mathbf{x}_*\mathbf{x}_*^\top \mid \Gamma\right]\mathbb{P}\left(\Gamma\right), S_0\right)$$
$$\geq \frac{1}{2}\phi^2\left(\mathbf{\Sigma}_\Gamma^*, S_0\right) , \tag{6}$$

where the first inequality holds by concavity of the compatibility constant (Lemma 20) and $\phi^2\left(\mathbb{E}\left[\mathbf{x}_*\mathbf{x}_*^\top \mathbb{1}\left\{\Gamma^c\right\}\right], S_0\right) \geq 0$ (Lemma 21). By Assumption 5, for all $\mathbf{v} \in \mathbb{R}^d$ with $\|\mathbf{v}\|_0 \leq C^*s_0 + 1$, it holds that

$$\mathbf{v}^\top \mathbf{\Sigma}_\Gamma^* \mathbf{v} \geq \mathbf{v}^\top\left(\phi_1^2\mathbf{I}_d\right)\mathbf{v} .$$

By invoking Lemma 31 with $\hat{\mathbf{\Sigma}} = \mathbf{\Sigma}_\Gamma^*$, $(1-\eta)\bar{\mathbf{\Sigma}} = \phi_1^2\mathbf{I}_d$, $\mathbf{D} = x_{\max}^2\mathbf{I}_d$, and $m = C^*s_0 + 1$, we obtain

$$\forall \mathbf{u} \in \mathbb{C}\left(S_0\right), \mathbf{u}^\top \mathbf{\Sigma}_\Gamma^* \mathbf{u} \geq \phi_1^2\|\mathbf{u}\|_2^2 - \frac{\left\|\mathbf{D}^{\frac{1}{2}}\mathbf{u}\right\|_1^2}{C^*s_0} .$$

Following the proof of Lemma 1, especially inequalities (4) and (5), we derive that for all $\mathbf{u} \in \mathbb{C}\left(S_0\right)$,

$$\mathbf{u}^\top \mathbf{\Sigma}_\Gamma^* \mathbf{u} \geq \frac{\phi_1^2}{s_0}\|\mathbf{u}_{S_0}\|_1^2 - \frac{1}{4\xi K s_0}\|\mathbf{u}_{S_0}\|_1^2 .$$

Since we supposed that $\frac{1}{4\xi K} \leq \frac{\phi_1^2}{3}$, we deduce that

$$\frac{s_0 \mathbf{u}^\top \boldsymbol{\Sigma}_\Gamma^* \mathbf{u}}{\|\mathbf{u}_{S_0}\|_1^2} \geq \phi_1^2 - \frac{1}{4\xi K}$$

$$\geq \frac{2\phi_1^2}{3},$$

which proves $\phi^2\left(\boldsymbol{\Sigma}_\Gamma^*, S_0\right) \geq \frac{2\phi_1^2}{3}$. Together with inequality (6), we obtain $\phi^2\left(\boldsymbol{\Sigma}^*, S_0\right) \geq \frac{\phi_1^2}{3}$. $\quad\square$

**Relaxed symmetry & Balanced covariance to ours:**

The following lemma shows that assumptions from Oh et al. (2021); Ariu et al. (2022) imply the greedy diversity, hence they imply Assumption 3.

**Lemma 3.** *If Assumption 6-8 hold, then the greedy diversity holds with* $\phi_G^2 = \frac{\phi_2^2}{2\nu C_\mathcal{X}}$.

*Proof of Lemma 3.* See Lemma 10 of Oh et al. (2021) and the paragraph followed by its statement. $\quad\square$

## C   COMPARISONS WITH MULTIPLE-PARAMETER SETTING

In this section, we compare the assumptions for the multiple-parameter setting (Bastani & Bayati, 2020; Wang et al., 2018) with our Assumption 3.

In the multiple-parameter setting, there are $K$ true parameter vectors, one for each arm, denoted by $\boldsymbol{\beta}_1, \boldsymbol{\beta}_2, \ldots, \boldsymbol{\beta}_K \in \mathbb{R}^d$. The active sets of the parameters may differ, and they are denoted by $S_1, S_2, \ldots, S_K \subset [d]$. In each round $t \in [T]$, a single context vector $\mathbf{x}_t \in \mathcal{X}$ is sampled from a fixed distribution and revealed to the agent, and the mean reward of arm $i \in [K]$ is given by $\mathbf{x}_t^\top \boldsymbol{\beta}_i$.

We first note that direct comparisons of the assumptions defined for these two different problem settings are not possible. For instance, there is no "feature vector for optimal arm" in the multiple-parameter setting to start with, as every feature vector is optimal for some arm. Therefore, the algorithms and the assumptions defined for one specific setting must be converted into the other setting; only then it would be possible to make comparisons.

A method that converts a single-parameter bandit instance into a multiple-parameter one was introduced by Kim & Paik (2019); Oh et al. (2021); Ariu et al. (2022), although it has only been used for experimental comparisons, and theoretical comparisons between the two settings have never been made. We explain the procedure for the conversion for completeness. Suppose one has a single-parameter bandit instance and an algorithm that operates in the multiple-parameter setting. The conversion concatenates $K$ feature vectors of the single-parameter setting, $\mathbf{x}_{t,1}, \mathbf{x}_{t,2}, \ldots, \mathbf{x}_{t,K} \in \mathbb{R}^d$, into one $Kd$-dimensional vector, $\mathbf{x}_t := \begin{pmatrix} \mathbf{x}_{t,1}^\top & \mathbf{x}_{t,2}^\top & \cdots & \mathbf{x}_{t,K}^\top \end{pmatrix}^\top \in \mathbb{R}^{Kd}$ and provide it to the algorithm as the context vector. If the true parameter is $\boldsymbol{\beta}$, then the hidden parameters the arms that the algorithm must learn are $\boldsymbol{\beta}_i = \begin{pmatrix} \mathbb{1}\{i=1\}\boldsymbol{\beta}^\top & \mathbb{1}\{i=2\}\boldsymbol{\beta}^\top & \cdots & \mathbb{1}\{i=K\}\boldsymbol{\beta}^\top \end{pmatrix}^\top$ for $i = 1, 2, \ldots, K$. Formally, we introduce a conversion that maps a single-parameter bandit instance to a multiple-parameter bandit instance.

**Definition 3** (Conversion mapping single-parameter to multiple-parameter)**.** *Let* $(\boldsymbol{\beta}, \{\mathbf{x}_{t,1}, \ldots, \mathbf{x}_{t,K}\})$ *be a single-parameter bandit instance where* $\boldsymbol{\beta} \in \mathbb{R}^d$ *and* $\mathbf{x}_{t,k} \in \mathbb{R}^d$ *for* $k \in [K]$. *Then, a conversion mapping* $\mathcal{C}$ *from a single-parameter bandit instance to a multiple-parameter bandit instance is defined as follows:*

$$\mathcal{C}\left(\boldsymbol{\beta}, \{\mathbf{x}_{t,1}, \ldots, \mathbf{x}_{t,K}\}\right) = \left(\boldsymbol{\beta}_1, \ldots, \boldsymbol{\beta}_K, \mathbf{x}_t\right),$$

*where*

$$\boldsymbol{\beta}_i = \left(\mathbb{1}\{i=1\}\boldsymbol{\beta}^\top \; \cdots \; \mathbb{1}\{i=K\}\boldsymbol{\beta}^\top\right)^\top \in \mathbb{R}^{Kd} \text{ for } i \in [K], \quad \mathbf{x} = \left(\mathbf{x}_{t,1}^\top \cdots \mathbf{x}_{t,K}^\top\right)^\top \in \mathbb{R}^{Kd}.$$

*We will show that under this conversion, the converted assumptions of Bastani & Bayati (2020) and Wang et al. (2018) are stronger than ours.* We first recall the assumptions for the multiple-parameter setting.

**Assumption 9** (Arm optimality, Assumptions 3 in Bastani & Bayati (2020))**.** *There exist constants* $h, p_* > 0$ *such that* $\mathbb{P}(\mathbf{x} \in U_i) \geq p_*$ *for all* $i \in [K]$, *where*

$$U_i := \left\{ \mathbf{x} \in \mathcal{X} \mid \mathbf{x}^\top \boldsymbol{\beta}_i > \max_{j \neq i} \mathbf{x}^\top \boldsymbol{\beta}_j + h \right\}.$$

**Assumption 10** (Compatibility condition on the constrained optimal arm, Assumption 4 in Bastani & Bayati (2020))**.** *There exists a constant* $\phi_0 > 0$ *such that for all* $i \in [K]$, $\phi(\boldsymbol{\Sigma}_i, S_i) \geq \phi_0$, *where* $\boldsymbol{\Sigma}_i := \mathbb{E}[\mathbf{x}\mathbf{x}^\top | \mathbf{x} \in U_i]$.

Assumption 10 imposes the compatibility condition on the set of features that are optimal for the $i$-th arm with large gaps. Although the original statements impose the conditions on a subset of arms $\mathcal{K}_{\text{opt}} \subset [K]$, following the proof of Proposition 2 in Bastani & Bayati (2020) reveals that $\mathcal{K}_{\text{opt}} = [K]$ must hold, and we replace it with $[K]$ for simpler comparisons. We define another set for comparison, which is constructed in the same way as $U_i$ but without the instantaneous gap condition as follows:

$$W_i := \left\{ \mathbf{x} \in \mathcal{X} \mid \mathbf{x}^\top \boldsymbol{\beta}_i > \max_{j \neq i} \mathbf{x}^\top \boldsymbol{\beta}_j \right\}.$$

Clearly, $U_i \subset W_i$. Intuitively, Assumption 3 is translated into the converted multiple-parameter instances as imposing the compatibility condition on a principal sub-matrix of the Gram matrix generated by the features in $W_i$, which is weaker than Assumption 10 demonstrated in two steps: *the compatibility condition is imposed on a sub-matrix corresponding to the $i$-th arm*, and *the Gram matrix of interest is generated on a larger set*. We rigorously demonstrate the relationship between the assumptions under the conversion by the following lemma.

**Lemma 4.** *Suppose that Assumption 9 and 10 hold for a multiple-parameter bandit instance that is converted from a single-parameter instance by a conversion mapping $\mathcal{C}$ (Definition 3). Then, for those multiple-parameter instances, Assumption 3 holds with $\phi_*^2 \geq K p_* \phi_0^2$ in the original single-parameter setting.*

*Proof of Lemma 4.* Let $(\boldsymbol{\beta}, \{\mathbf{x}_1, \ldots, \mathbf{x}_K\})$ be a single-parameter bandit instance and $\mathcal{C}(\boldsymbol{\beta}, \{\mathbf{x}_1, \ldots, \mathbf{x}_K\}) = (\boldsymbol{\beta}_1, \ldots, \boldsymbol{\beta}_K, \mathbf{x})$ be the converted multiple-parameter bandit instance. Note that for fixed $i \in [K]$, if $\mathbf{x} \in U_i$, then $\mathbf{x}_i$ is the optimal arm in the original setting. Let $\mathbf{x}_* \in \mathbb{R}^d$ be the optimal feature vector in the single-parameter setting, i.e., $\mathbf{x}_* = \operatorname{argmax}_{i \in [K]} \mathbf{x}_i^\top \boldsymbol{\beta}$. Then, we have

$$\mathbb{E}[\mathbf{x}_* \mathbf{x}_*^\top] = \sum_{i=1}^K \mathbb{E}[\mathbf{x}_i \mathbf{x}_i^\top, \mathbf{x} \in W_i]$$

$$\succeq \sum_{i=1}^K \mathbb{E}[\mathbf{x}_i \mathbf{x}_i^\top, \mathbf{x} \in U_i]$$

$$\succeq \sum_{i=1}^K p_* \mathbb{E}[\mathbf{x}_i \mathbf{x}_i^\top \mid \mathbf{x} \in U_i],$$

where the first equality holds by the definition of $W_i$, the first inequality is due to that $U_i \subset W_i$, and the last inequality holds by Assumption 9. Note that $\mathbb{E}[\mathbf{x}_i \mathbf{x}_i^\top \mid \mathbf{x} \in U_i]$ is a $d \times d$ principal submatrix of $\boldsymbol{\Sigma}_i \in \mathbb{R}^{Kd \times Kd}$. Since $\boldsymbol{\Sigma}_i$ satisfies the compatibility condition with constant $\phi_0$ by Assumption 10, the compatibility constant for $\mathbb{E}[\mathbf{x}_i \mathbf{x}_i^\top \mid \mathbf{x} \in U_i]$ must be at least $\phi_0$. Formally, we

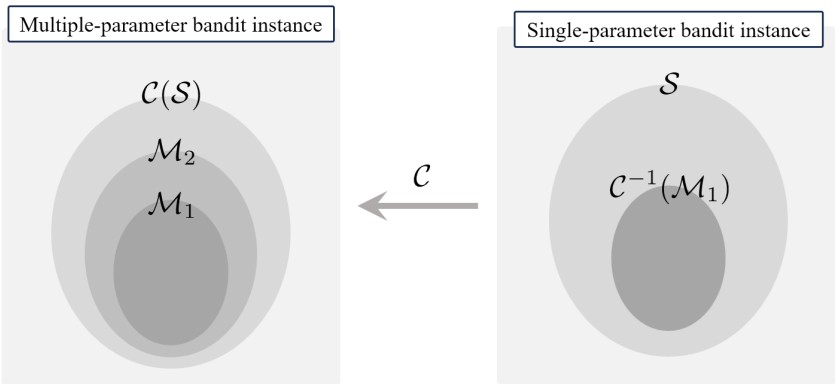

Figure 3: Illustration of the results of Lemma 4 and Lemma 5. Let $\mathcal{C}$ be a conversion mapping that converts a single-parameter bandit instance into a multiple-parameter one by $Kd$-dimensional context vector construction, $\mathcal{M}_1$ a set of multiple-parameter instances converted by $\mathcal{C}$ satisfying Assumption 9 and 10, $\mathcal{M}_2$ a set of multiple-parameter instances converted by $\mathcal{C}$ satisfying Assumption 11, and $\mathcal{S}$ a set of single-parameter instances satisfying Assumption 3. By the definition $\mathcal{C}(\mathcal{S})$ denotes the image of $\mathcal{S}$ under $\mathcal{C}$ which is the set of multiple-parameter instances converted from $\mathcal{S}$ by $\mathcal{C}$. Similarly, $\mathcal{C}^{-1}(\mathcal{M}_1)$ is the inverse image of $\mathcal{M}_1$ under $\mathcal{C}$ which is the set of single-parameter instances that map to a member of $\mathcal{M}_1$. By Lemma 4, we ensure that $\mathcal{C}^{-1}(\mathcal{M}_1) \subset \mathcal{S}$, which means that our compatibility condition on the optimal arm (Assumption 3) is weaker than those of Bastani & Bayati (2020); Wang et al. (2018) through the conversion mapping $\mathcal{C}$. On the other hand, Lemma 5 ensures that $\mathcal{M}_1 \subset \mathcal{M}_2$.

let $\boldsymbol{\Sigma}_i^{d \times d} = \mathbb{E}[\mathbf{x}_i \mathbf{x}_i^\top | \mathbf{x} \in U_i]$ and prove the claim by the following argument:

$$\phi_0^2 \leq \phi^2(\boldsymbol{\Sigma}_i, S_i)$$

$$= \inf_{\boldsymbol{\beta} \in \mathbb{C}(S_i) \setminus \{\mathbf{0}_{Kd}\}} \frac{s_0 \boldsymbol{\beta}^\top \boldsymbol{\Sigma}_i \boldsymbol{\beta}}{\|\boldsymbol{\beta}_{S_i}\|_1^2}$$

$$\leq \inf_{\substack{\boldsymbol{\beta} \in \mathbb{C}(S_i) \setminus \{\mathbf{0}_{Kd}\} \\ \boldsymbol{\beta}_{S_i^c} = \mathbf{0}_{Kd}}} \frac{s_0 \boldsymbol{\beta}^\top \boldsymbol{\Sigma}_i \boldsymbol{\beta}}{\|\boldsymbol{\beta}_{S_i}\|_1^2}$$

$$= \inf_{\boldsymbol{\beta} \in \mathbb{C}(S_0) \setminus \{\mathbf{0}_d\}} \frac{s_0 \boldsymbol{\beta}^\top \boldsymbol{\Sigma}_i^{d \times d} \boldsymbol{\beta}}{\|\boldsymbol{\beta}_{S_0}\|_1^2}$$

$$= \phi^2(\boldsymbol{\Sigma}_i^{d \times d}, S_0).$$

Therefore, by the concavity of compatibility constants (Lemma 20), we conclude that Assumption 3 holds with $\phi_*^2 \geq Kp_*\phi_0^2$. $\qquad\square$

Lemma 4 should be interpreted with particular care. We note that the comparison is possible only between bandit instances that are converted using the conversion mapping $\mathcal{C}$, and Lemma 4 states that for those instances, our assumption (Assumption 3) is weaker than those of Bastani & Bayati (2020); Wang et al. (2018) (Assumptions 9 and 10). However, it does not imply that our analysis holds for any multiple-parameter instances that satisfy Assumptions 9 and 10. This is trivial given that our algorithm and assumptions are presented only under the single-parameter setting and there are multiple-parameter bandit instances that satisfy Assumptions 9 and 10 but do not have corresponding single-parameter instances. If the whole algorithm and analysis were to be transferred to the multiple-parameter setting, we would require a multiple-parameter counterpart of Assumption 3 that validates the analysis. We introduce such an assumption and compare it with the assumptions in Bastani & Bayati (2020); Wang et al. (2018).

**Assumption 11** (Compatibility condition for the optimal feature). *There exists $p_* > 0$ such that $\mathbb{P}(\mathbf{x} \in W_i) \geq p_*$ for all $i \in [K]$. There exists $\phi_* > 0$ such that for all $i \in [K]$, $\phi(\Sigma_i', S_i) \geq \phi_*$, where $\Sigma_i' := \mathbb{E}[\mathbf{x}\mathbf{x}^\top | \mathbf{x} \in W_i]$.*

Assumption 11 does not impose the constraints regarding the gap $h$ in Assumptions 9 and 10, and hence it is strictly weaker than those. Formally, we introduce the following lemma.

**Lemma 5.** *In the multiple-parameter setting, Assumption 9 and 10 imply Assumption 11.*

*Proof of Lemma 5.* Since $U_i \subset W_i$ for all $i \in [K]$, we have that $\mathbb{P}(\mathbf{x} \in W_i) \geq \mathbb{P}(\mathbf{x} \in U_i)$. Therefore $\mathbb{P}(\mathbf{x} \in U_i) \geq p_*$ implies $\mathbb{P}(\mathbf{x} \in W_i) \geq p_*$. We also have that $\mathbb{E}[\mathbf{xx}^\top, \mathbf{x} \in W_i] \succeq \mathbb{E}[\mathbf{xx}^\top, \mathbf{x} \in U_i]$. Then, we derive that

$$\mathbb{E}[\mathbf{xx}^\top | \mathbf{x} \in W_i] \succeq \mathbb{E}[\mathbf{xx}^\top, \mathbf{x} \in W_i]$$
$$\succeq \mathbb{E}[\mathbf{xx}^\top, \mathbf{x} \in U_i]$$
$$\succeq p_* \mathbb{E}[\mathbf{xx}^\top | \mathbf{x} \in U_i],$$

which implies that $\phi^2(\mathbf{\Sigma}_i', S_i) \geq p_* \phi_0^2$. $\qquad\square$

## D  ADDITIONAL DISCUSSION ON SPARSITY-AWARENESS

In this section, we provide additional discussion on the sparsity-awareness of our algorithm in comparison to sparsity-agnostic algorithms (Oh et al., 2021; Ariu et al., 2022).

Although $M_0$ theoretically depends on $s_0$, $s_0$ does not need to be precisely specified (as long as it is within a constant factor of $s_0$). For instance, if an upper bound on $s_0$ smaller than the trivial ambient dimension $d$ exists, this information can be leveraged. Besides, it is essential to note that $M_0$ does not solely depend on $s_0$ but is a tunable parameter that depends on $s_0$ combined with other problem-dependent factors that are unknown to the algorithm. Tuning parameters that depend on unknown factors *applies not only to ours, but also to almost all Lasso bandit and parametric bandit algorithms* — such as parameters that depend on the sub-Gaussian parameter of the noise $\sigma$ and the upper bound of the norm of the context vector $x_{\max}$. We do not need to specify each of those problem parameters separately in practice. Rather, $M_0$ is tuned as a whole. We observe that our algorithm is not sensitive to the choice of $M_0$ in numerical experiments. Figure 4 shows the cumulative regret of `FS-WLasso` under the setting of Experiment 2 with different values of $M_0$ and shows the robust performances under different values of $M_0$.

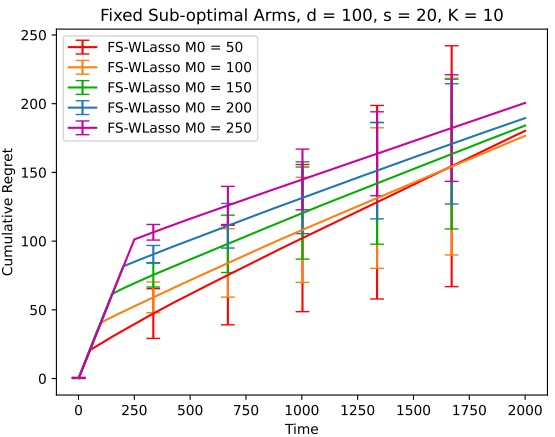

Figure 4: Evaluations of `FS-WLasso` with various lengths of forced-sampling stage under the setting of Experiment 2

There is a clear distinction between knowing sparsity and assuming stronger context distributions. Sparsity is about the *unknown parameter* $\boldsymbol{\beta}^*$, not about context distribution. Whether or not $s_0$ is known in practice, one still needs to tune hyper-parameters — and even if the algorithm is sparsity-agnostic, there are still hyper-parameters to be tuned anyway with other unknown problem-dependent factors such as the sub-Gaussian parameter of the noise. Hence, not knowing $s_0$ does not lead to any increased complexity in practice.

On the other hand, stronger context distributions employed in the existing literature (e.g., relaxed symmetry & balanced covariance) are about $\mathbf{x}_t$. When context distributions do not satisfy the strong assumptions in previous literature, algorithms can critically undermine regret performance, with no recourse for adjustment or guarantees. What is even worse is that such stochastic assumptions on the context distributions are NOT verifiable in practice, particularly in high dimensions. Our work primarily focuses on alleviating these stochastic assumptions on context, providing the weakest conditions on context distribution known to achieve the poly-logarithmic regret.

Besides, there are other works that still incorporate extra stochastic conditions despite knowing sparsity (Li et al., 2021) or specific optimality criteria for context distributions also under sparsity-awareness (Bastani & Bayati, 2020; Wang et al., 2018). Even with sparsity-awareness, *our work is the first Lasso bandit result that achieves the poly-logarithmic regret bound without additional context distributional assumptions after compatibility condition.* In this regard, we still provide a new insight that the previous literature did not know.

To conclude this section, we introduce the fundamental challenges of making our algorithms sparsity-agnostic. Regret analysis in Lasso bandits necessitates satisfying the compatibility condition of the empirical Gram matrix constructed from previously selected arms. Ensuring this requires (i) *an underlying assumption about the compatibility of the expected Gram matrix* and (ii) *a sufficient number of samples to guarantee that the empirical Gram matrix concentrates around the expected Gram matrix.* We note that the number of required samples depends on $s_0$ because the matrix concentration inequality we use (Lemma 30) depends on $s_0$. Therefore, in most sparse linear bandit algorithms (Kim & Paik, 2019; Li et al., 2021; Oh et al., 2021; Ariu et al., 2022; Chakraborty et al., 2023), including ours, the maximum regret is incurred during the burn-in phase, where the compatibility condition of the empirical Gram matrix is not guaranteed, and the length of the burn-in phase depends on $s_0$ if one would like the tightest bound.

As we demonstrate in Appendix B, the diversity assumptions employed by the previous works (Assumptions 4, 7,and 8) are designed to ensure that samples obtained by greedy selections (or even any other policies) automatically explore the feature space, allowing the algorithm to employ a single exploitative policy, regardless of which phase it is in. For example, in Oh et al. (2021), the length of the burn-in phase, denoted by $T_0$, clearly depends on $s_0$ as $T_0 = O(s_0^2)$, but their algorithm only makes greedy selection without explicitly specifying $T_0$. In contrast, in our case (Theorem 2), we only assume the compatibility condition on the optimal arm. *Without the diversity assumptions on the context distribution, a greedy policy or other exploitative policies no longer ensure the compatibility condition on the empirical Gram matrix.* Therefore, our algorithm runs in two phases: the *Forced sampling stage* and the *Greedy selection stage*. At the end of the forced sampling stage, we expect the compatibility condition on the empirical Gram matrix to be ensured. As a result, the length of the forced sampling stage depends on $s_0$. We again note that a possible range of $s_0$ is sufficient to set $M_0$ theoretically, instead of its precise value.

## E    REGRET BOUND OF `FS-WLasso`

In this section, we provide proofs for Theorems 2 and 3. We briefly mention some trivial implications of Assumptions 1 and 2. Under Assumption 1, we have $\text{reg}_t = \mathbf{x}_{t,a_t^*}^\top \boldsymbol{\beta}^* - \mathbf{x}_{t,a_t}^\top \boldsymbol{\beta}^* \leq \|\mathbf{x}_{t,a_t^*} - \mathbf{x}_{t,a_t}\|_\infty \|\boldsymbol{\beta}^*\|_1 \leq 2x_{\max}b$, where the Cauchy-Schwarz inequality and the triangle inequality are used. The fact that the instantaneous regret is at most $2x_{\max}b$ implies that $\Delta_* \leq 2x_{\max}b$, since otherwise $\mathbb{P}(\Delta_t > 2x_{\max}b) \geq 1 - (2x_{\max}b/\Delta_*)^\alpha > 0$ by Assumption 2, which contradicts $\Delta_t \leq 2x_{\max}b$.

### E.1    PROPOSITION 1

We introduce a proposition that establishes the core parts of the proofs for Theorem 2 and 3.

**Proposition 1.** *Suppose Assumptions 1-3 hold. Let $\delta \in (0,1]$ and $\tau_1 \in \mathbb{N}_0$ be given. Let $\tau_2$ be a constant that satisfies*

$$\tau_2 \geq \max \left\{ C_2 \log \frac{7d}{\delta} + 2C_2 \log \log \frac{28dC_2^2}{\delta}, \tau_1 + \frac{2048 x_{\max}^4 s_0^2}{\phi_*^4} \left( \log \frac{d^2}{\delta} + 2 \log \frac{64 x_{\max}^2 s_0}{\phi_*^2} \right), 2\tau_1, w^2 M_0 \right\},$$

*where $C_2 = \max\left\{2, \left(\frac{400\sigma x_{\max}^2 s_0}{\Delta_* \phi_*^2}\right)^2 \left(\frac{80x_{\max}^2 s_0}{\phi_*^2}\right)^{\frac{2}{\alpha}}\right\}$. Suppose the agent runs Algorithm 1 with $\lambda_t$ as follows:*

$$\lambda_t = 4\sigma x_{\max}\left(\sqrt{2w^2 M_0 \log\frac{2d}{\delta}} + 2^{\frac{3}{4}}\sqrt{(t - M_0)\log\frac{7d(\log 2(t - M_0))^2}{\delta}}\right).$$

*Define the (weighted) empirical Gram matrix as $\hat{\mathbf{V}}_{M_0+n} = \sum_{t=1}^{M_0} w\mathbf{x}_{t,a_t}\mathbf{x}_{t,a_t}^\top + \sum_{t=M_0+1}^{M_0+n} \mathbf{x}_{t,a_t}\mathbf{x}_{t,a_t}^\top$. If the compatibility constant of $\hat{\mathbf{V}}_{M_0+\tau_1}$ satisfies*

$$\phi^2\left(\hat{\mathbf{V}}_{M_0+\tau_1}, S_0\right) \geq \max\left\{\frac{4x_{\max}s_0}{\Delta_*}\left(\frac{80x_{\max}^2 s_0}{\phi_*^2}\right)^{\frac{1}{\alpha}}\lambda_{M_0+\tau_2}, 64x_{\max}^2 s_0 \log\frac{1}{\delta}\right\},$$

*then with probability $1 - 4\delta$, the estimation error of $\hat{\boldsymbol{\beta}}_t$ satisfies the following for all $t \geq M_0 + \tau_2 + 1$:*

$$\left\|\boldsymbol{\beta}^* - \hat{\boldsymbol{\beta}}_t\right\|_1 \leq \frac{200\sigma x_{\max}s_0}{\phi_*^2}\sqrt{\frac{2\log\log 2(t - M_0) + \log\frac{7d}{\delta}}{t - M_0}}.$$

*Furthermore, under the same event, the cumulative regret from $t = M_0 + \tau_1 + 1$ to $T$ with $T \geq M_0 + \tau_2$ is bounded as the following:*

$$\sum_{t=M_0+\tau_1+1}^{T} reg_t \leq I_{\tau_2} + I_T$$

*where*

$$I_{\tau_2} = \frac{5\Delta_*}{4}\left(\frac{80x_{\max}^2 s_0}{\phi_*^2}\right)^{-1-\frac{1}{\alpha}}(\tau_2 - \tau_1 + 1) + 4\Delta_* \log\frac{1}{\delta},$$

$$I_T = \begin{cases} \mathcal{O}\left(\frac{1}{\Delta_*^\alpha(1-\alpha)}\left(\frac{\sigma x_{\max}^2 s_0}{\phi_*^2}\right)^{1+\alpha} T^{\frac{1-\alpha}{2}}\left(\log d + \log\frac{\log T}{\delta}\right)^{\frac{1+\alpha}{2}}\right) & \alpha \in (0, 1), \\ \mathcal{O}\left(\frac{1}{\Delta_*}\left(\frac{\sigma x_{\max}^2 s_0}{\phi_*^2}\right)^2(\log T)\left(\log d + \log\frac{\log T}{\delta}\right)\right) & \alpha = 1, \\ \mathcal{O}\left(\frac{\alpha}{(\alpha-1)^2}\cdot\frac{\sigma^2}{\Delta_*}\left(\frac{x_{\max}^2 s_0}{\phi_*^2}\right)^{1+\frac{1}{\alpha}}\left(\log d + \log\frac{1}{\delta}\right)\right) & \alpha > 1. \end{cases}$$

*Proof of Proposition 1.* Let $N_{\tau_1}(t') = \sum_{i=M_0+\tau_1+1}^{M_0+\tau_1+t'} \mathbb{1}\{a_i \neq a_i^*\}$ be the number of sub-optimal arm selections during the first $t'$ greedy selections, starting from $t = M_0 + \tau_1 + 1$. Define the following events :

$$\mathcal{E}_e = \left\{\omega \in \Omega : \max_{j \in [d]}\left|\sum_{i=1}^{M_0} \eta_i(\mathbf{x}_{i,a_i})_j\right| \leq \sigma x_{\max}\sqrt{2M_0 \log\frac{d}{\delta}}\right\},$$

$$\mathcal{E}_g = \left\{\omega \in \Omega : \forall n \geq 1, \max_{j \in [d]}\left|\sum_{i=M_0+1}^{M_0+n} \eta_i(\mathbf{x}_{i,a_i})_j\right| \leq 2^{\frac{3}{4}}\sigma x_{\max}\sqrt{n \log\frac{7d(\log 2n)^2}{\delta}}\right\},$$

$$\mathcal{E}_N(\tau_1) = \left\{\omega \in \Omega : \forall t' \geq 0, N_{\tau_1}(t') \leq \frac{5}{4}\sum_{i=M_0+\tau_1+1}^{M_0+\tau_1+t'} \min\left\{1, \left(\frac{2x_{\max}}{\Delta_*}\|\boldsymbol{\beta}^* - \boldsymbol{\beta}_{i-1}\|_1\right)^\alpha\right\} + 4\log\frac{1}{\delta}\right\},$$

$$\mathcal{E}^*(\tau_1, \tau_2) = \left\{\omega \in \Omega : \forall t' \geq \tau_2 - \tau_1 + 1, \phi^2\left(\sum_{t=M_0+\tau_1+1}^{M_0+\tau_1+t'} \mathbf{x}_{t,a_t^*}\mathbf{x}_{t,a_t^*}^\top\right) \geq \frac{\phi_*^2 t'}{2}\right\}.$$

The first two events are concentration inequalities of the noise, which are necessary to guarantee the error bound of the Lasso estimator. The third event is upper boundedness of the number of sub-optimal arm selections conditioned on the estimation errors, and the event occurs with high

probability by the margin condition. The last event is that the compatibility constant of the empirical Gram matrix of the optimal feature vectors from round $t = M_0 + \tau_1 + 1$ being bounded below, which holds with high probability by concentration inequality of matrices and Assumption 3. In Appendix E.4.1, we show that each event happens with probability at least $1 - \delta$. By the union bound, all the events happens with probability at least $1 - 4\delta$, and we assume that these events are valid for the rest of the proof.

We first present a lemma that bounds the estimation errors in rounds $t = M_0 + \tau_1 + 1 \ldots M_0 + \tau_2$.

**Lemma 6.** *For each $t' = 0, \ldots \tau_2 - \tau_1$, the estimation error of $\hat{\boldsymbol{\beta}}_{M_0+\tau_1+t'}$ is bounded as the following:*

$$\left\| \boldsymbol{\beta}^* - \hat{\boldsymbol{\beta}}_{M_0+\tau_1+t'} \right\|_1 \leq \frac{\Delta_*}{2x_{\max}} \left( \frac{\phi_*^2}{80x_{\max}^2 s_0} \right)^{\frac{1}{\alpha}} .$$

Define $\overline{N}(t') = \sum_{t=M_0+\tau_1+1}^{M_0+\tau_1+t'} \left( \frac{2x_{\max}}{\Delta_*} \left\| \boldsymbol{\beta}^* - \hat{\boldsymbol{\beta}}_{t-1} \right\|_1 \right)^{\alpha}$. $\overline{N}(t')$ is determined by the errors of the estimators from round $M_0 + \tau_1 + 1$ to round $M_0 + \tau_1 + t'$. The following lemma shows that small $\overline{N}(t')$ implies small estimation error in round $M_0 + \tau_1 + t' + 1$ when $t' \geq \tau_2 - \tau_1 + 1$.

**Lemma 7.** *Suppose $t' \geq \tau_2 - \tau_1 + 1$ and $\overline{N}(t') \leq \frac{\phi_*^2}{80x_{\max}^2 s_0} t'$. Then, the following holds:*

$$\left\| \boldsymbol{\beta}^* - \hat{\boldsymbol{\beta}}_{M_0+\tau_1+t'} \right\|_1 \leq \frac{200\sigma x_{\max} s_0}{\phi_*^2} \sqrt{\frac{2\log\log 2(\tau_1 + t') + \log\frac{7d}{\delta}}{\tau_1 + t'}} .$$

Combining the two lemmas and using mathematical induction leads to the following lemma :

**Lemma 8.** $\overline{N}(t') \leq \frac{\phi_*^2}{80x_{\max}^2 s_0} t'$ *holds for all $t' \geq 0$.*

Combining Lemma 7 and Lemma 8, and by setting $t = M_0 + \tau_1 + t'$, we obtain that for all $t \geq M_0 + \tau_2 + 1$, it holds that

$$\left\| \boldsymbol{\beta}^* - \hat{\boldsymbol{\beta}}_t \right\|_1 \leq \frac{200\sigma x_{\max} s_0}{\phi_*^2} \sqrt{\frac{2\log\log 2(t - M_0) + \log\frac{7d}{\delta}}{t - M_0}} ,$$

which proves the first part of the proposition.

To prove the second part of the proposition, define $\overline{\Delta}_t$ as the following:

$$\overline{\Delta}_t = \begin{cases} \Delta_* \left( \frac{\phi_*^2}{80x_{\max}^2 s_0} \right)^{\frac{1}{\alpha}} & t \leq M_0 + \tau_2 \\ \frac{400\sigma x_{\max}^2 s_0}{\phi_*^2} \sqrt{\frac{2\log\log 2(t-M_0)+\log\frac{7d}{\delta}}{t-M_0}} & t \geq M_0 + \tau_2 + 1 \end{cases} .$$

Note that by Lemmas 6, 7 and 8, for all $t \geq M_0 + \tau_1$, it holds that $2x_{\max} \left\| \boldsymbol{\beta}^* - \hat{\boldsymbol{\beta}}_t \right\|_1 \leq \overline{\Delta}_t$. We utilize the following lemma.

**Lemma 9.** *Let $\tau \in \mathbb{N}_0$ be given. Suppose $\{\overline{\Delta}_t\}_{t=0}^{\infty}$ is a non-increasing sequence of real numbers that satisfies $2x_{\max} \left\| \boldsymbol{\beta}^* - \hat{\boldsymbol{\beta}}_t \right\|_1 \leq \overline{\Delta}_t$ for all $t \geq \tau$. Then, under the event $\mathcal{E}_N(\tau)$, the cumulative regret from $t = \tau + 1$ to $T$ is bounded as follows:*

$$\sum_{t=\tau+1}^{T} reg_t \leq 4\overline{\Delta}_\tau \log\frac{1}{\delta} + \frac{5}{4} \sum_{t=\tau}^{T-1} \overline{\Delta}_t \min \left\{ 1, \left( \frac{\overline{\Delta}_t}{\Delta_*} \right)^{\alpha} \right\} .$$

By Lemma 9 with $\tau = M_0 + \tau_1$, we have

$$\sum_{t=M_0+\tau_1+1}^{T} reg_t \leq 4\overline{\Delta}_{M_0+\tau_1} \log\frac{1}{\delta} + \frac{5}{4} \sum_{t=M_0+\tau_1}^{T-1} \frac{\overline{\Delta}_t^{1+\alpha}}{\Delta_*^{\alpha}} . \tag{7}$$

We are left to bound $\sum_{t=M_0+\tau_1}^{T-1} \overline{\Delta}_t^{1+\alpha}$. We separately bound the summation for cases where $t \le M_0 + \tau_2$ and $t \ge M_0 + \tau_2 + 1$. For $M_0 + \tau_1 \le t \le M_0 + \tau_2$, we have

$$\sum_{t=M_0+\tau_1}^{M_0+\tau_2} \overline{\Delta}_t^{1+\alpha} = \sum_{t=M_0+\tau_1}^{M_0+\tau_2} \Delta_*^{1+\alpha} \left( \frac{\phi_*^2}{80x_{\max}^2 s_0} \right)^{\frac{1+\alpha}{\alpha}}$$

$$= \Delta_*^{1+\alpha} \left( \frac{\phi_*^2}{80x_{\max}^2 s_0} \right)^{\frac{1+\alpha}{\alpha}} (\tau_2 - \tau_1 + 1).$$

Note that $\overline{\Delta}_{M_0+\tau_1} = \Delta_* \left( \frac{\phi_*^2}{80x_{\max}^2 s_0} \right)^{\frac{1}{\alpha}} \le \Delta_*$ by Lemma 21. If we set $I_{\tau_2} = 4\Delta_* \log \frac{1}{\delta} + \frac{5\Delta_*}{4} \left( \frac{80x_{\max}^2 s_0}{\phi_*^2} \right)^{-1-\frac{1}{\alpha}} (\tau_2 - \tau_1 + 1)$, then we have

$$4\overline{\Delta}_{M_0+\tau_1} \log \frac{1}{\delta} + \frac{5}{4} \sum_{t=M_0+\tau_1}^{M_0+\tau_2} \frac{\overline{\Delta}_t^{1+\alpha}}{\Delta_*^\alpha} \le I_{\tau_2} . \tag{8}$$

For $t = M_0 + \tau_2 + 1, \ldots, T - 1$, we have

$$\sum_{t=M_0+\tau_2+1}^{T-1} \overline{\Delta}_t^{1+\alpha} = \sum_{t=M_0+\tau_2+1}^{T-1} \left( \frac{400\sigma x_{\max}^2 s_0}{\phi_*^2} \right)^{1+\alpha} \left( \frac{2\log\log 2(t-M_0) + \log \frac{7d}{\delta}}{t - M_0} \right)^{\frac{1+\alpha}{2}}$$

$$= \left( \frac{400\sigma x_{\max}^2 s_0}{\phi_*^2} \right)^{1+\alpha} \sum_{n=\tau_2+1}^{T-M_0-1} \left( \frac{2\log\log 2n + \log \frac{7d}{\delta}}{n} \right)^{\frac{1+\alpha}{2}} . \tag{9}$$

By Lemma 26, we have

$$\sum_{n=\tau_2+1}^{T-M_0-1} \left( \frac{2\log\log 2n + \log \frac{7d}{\delta}}{n} \right)^{\frac{1+\alpha}{2}} \le \begin{cases} \frac{2}{1-\alpha} T^{\frac{1-\alpha}{2}} \left( 2\log\log 2T + \log \frac{7d}{\delta} \right)^{\frac{1+\alpha}{2}} & \alpha \in (0,1) \\ (\log T)(2\log\log 2T + \log \frac{7d}{\delta}) & \alpha = 1 \\ \frac{4\alpha}{(\alpha-1)^2} \cdot \frac{\left( 2\log\log 2\tau_2 + \log \frac{7d}{\delta} \right)^{\frac{\alpha+1}{2}}}{\tau_2^{\frac{\alpha-1}{2}}} & \alpha > 1 . \end{cases} \tag{10}$$

Lemma 26 requires $\tau_2 \ge 8$, and it is guaranteed by $\tau_2 \ge \frac{2048 x_{\max}^4 s_0}{\phi_*^2} \left( \log \frac{d}{\delta} + 2\log \frac{64 x_{\max}^2 s_0}{\phi_*^2} \right) \ge 8 \times \left( \log \frac{d}{\delta} + 2\log 4 \right)$, where the first inequality holds by the choice of $\tau_2 \ge \tau_1 + \frac{2048 x_{\max}^4 s_0}{\phi_*^2} \left( \log \frac{d}{\delta} + 2\log \frac{64 x_{\max}^2 s_0}{\phi_*^2} \right)$, and the second inequality holds by Lemma 21. We need to check another property of $\tau_2$ to simplify the regret when $\alpha > 1$. Recall that $\tau_2 \ge C_2 \log \frac{7d}{\delta} + 2C_2 \log\log \frac{28dC_2^2}{\delta}$, where $C_2 = \max \left\{ 2, \left( \frac{400\sigma x_{\max}^2 s_0}{\Delta_* \phi_*^2} \right)^2 \left( \frac{80x_{\max}^2 s_0}{\phi_*^2} \right)^{\frac{2}{\alpha}} \right\}$. Then, by Lemma 25 with $C = C_2$ and $b = \log \frac{7d}{\delta}$, it holds that

$$\forall n \ge \tau_2, \frac{2\log\log 2n + \log \frac{7d}{\delta}}{n} \le \left( \frac{400\sigma x_{\max}^2 s_0}{\Delta_* \phi_*^2} \right)^{-2} \left( \frac{80x_{\max}^2 s_0}{\phi_*^2} \right)^{-\frac{2}{\alpha}} . \tag{11}$$

Therefore, for $\alpha > 1$, it holds that

$$\frac{\left( 2\log\log 2\tau_2 + \log \frac{7d}{\delta} \right)^{\frac{\alpha+1}{2}}}{\tau_2^{\frac{\alpha-1}{2}}} = \left( \frac{2\log\log 2\tau_2 + \log \frac{7d}{\delta}}{\tau_2} \right)^{\frac{\alpha-1}{2}} \left( 2\log\log 2\tau_2 + \frac{7d}{\delta} \right)$$

$$\le \left( \frac{400\sigma x_{\max}^2 s_0}{\Delta_* \phi_*^2} \right)^{1-\alpha} \left( \frac{80x_{\max}^2 s_0}{\phi_*^2} \right)^{\frac{1-\alpha}{\alpha}} \left( 2\log\log 2\tau_2 + \frac{7d}{\delta} \right) . \tag{12}$$

Putting Eq. (9), (10), and (12) together, we obtain

$$\sum_{t=M_0+\tau_2+1}^{T-1} \overline{\Delta}_t^{1+\alpha} \le \begin{cases} \frac{2}{1-\alpha} \left( \frac{400\sigma x_{\max}^2 s_0}{\phi_*^2} \right)^{1+\alpha} T^{\frac{1-\alpha}{2}} \left( 2\log\log 2T + \log \frac{7d}{\delta} \right)^{\frac{1+\alpha}{2}} & \alpha \in (0,1) \\ \left( \frac{400\sigma x_{\max}^2 s_0}{\phi_*^2} \right)^2 (\log T) \left( 2\log\log 2T + \log \frac{7d}{\delta} \right) & \alpha = 1 \\ \frac{4\alpha \Delta_*^{\alpha-1}}{(\alpha-1)^2} \left( \frac{400\sigma x_{\max}^2 s_0}{\phi_*^2} \right)^2 \left( \frac{80x_{\max}^2 s_0}{\phi_*^2} \right)^{\frac{1}{\alpha}-1} \left( 2\log\log 2\tau_2 + \log \frac{7d}{\delta} \right) & \alpha > 1 . \end{cases}$$

Then, we conclude that

$$\frac{5}{4} \sum_{t=M_0+\tau_2+1}^{T-1} \frac{\overline{\Delta}_t^{1+\alpha}}{\Delta_*^\alpha} \leq I_T \,, \tag{13}$$

where

$$
I_T = \begin{cases}
\mathcal{O}\left( \frac{1}{(1-\alpha)\Delta_*^\alpha} \left(\frac{\sigma x_{\max}^2 s_0}{\phi_*^2}\right)^{1+\alpha} T^{\frac{1-\alpha}{2}} \left(\log d + \log \frac{\log T}{\delta}\right)^{\frac{1+\alpha}{2}} \right) & \alpha \in (0,1) \\[2mm]
\mathcal{O}\left( \frac{1}{\Delta_*} \left(\frac{\sigma x_{\max}^2 s_0}{\phi_*^2}\right)^2 (\log T) \left(\log d + \log \frac{\log T}{\delta}\right) \right) & \alpha = 1 \\[2mm]
\mathcal{O}\left( \frac{\alpha}{(\alpha-1)^2} \cdot \frac{\sigma^2}{\Delta_*} \left(\frac{x_{\max}^2 s_0}{\phi_*^2}\right)^{1+\frac{1}{\alpha}} \left(\log d + \log \frac{1}{\delta}\right) \right) & \alpha > 1 \,.
\end{cases}
$$

The proof is complete by combining inequalities (7), (8), and (13).

$$\sum_{t=M_0+\tau_1+1}^{T} \mathrm{reg}_t \leq 4\overline{\Delta}_{M_0+\tau_1} \log \frac{1}{\delta} + \frac{5}{4} \sum_{t=M_0+\tau_1}^{M_0+\tau_2} \frac{\overline{\Delta}_t^{1+\alpha}}{\Delta_*^\alpha} + \frac{5}{4} \sum_{t=M_0+\tau_2+1}^{T-1} \frac{\overline{\Delta}_t^{1+\alpha}}{\Delta_*^\alpha}$$

$$\leq I_{\tau_2} + I_T \,.$$

$\square$

## E.2 PROOF OF THEOREM 2

**Theorem** (Formal version of Theorem 2) *Suppose Assumptions 1-3 hold. For $\delta \in (0,1]$, let $\tau$ be a constant given by*

$$\tau = \max\left\{ C_2 \log \frac{7d}{\delta} + 2C_2 \log\log \frac{28dC_2^2}{\delta}, \frac{2048 x_{\max}^4 s_0^2}{\phi_*^4} \left(\log \frac{d^2}{\delta} + 2\log \frac{64 x_{\max}^2 s_0}{\phi_*^2}\right) \right\} \,,$$

*where $C_2 = \max\left\{ 2, \left(\frac{400\sigma x_{\max}^2 s_0}{\Delta_* \phi_*^2}\right)^2 \left(\frac{80 x_{\max}^2 s_0}{\phi_*^2}\right)^{\frac{2}{\alpha}} \right\}$. If we set the input parameters of Algorithm 1 by*

$$M_0 = \max\left\{ \rho^2 \left(\frac{100\sigma x_{\max}^2 s_0}{\Delta_* \phi_*^2}\right)^2 \left(\frac{80 x_{\max}^2 s_0}{\phi_*^2}\right)^{\frac{2}{\alpha}} \left(2\log\log 2\tau + \log \frac{7d}{\delta}\right), \frac{2048\rho^2 x_{\max}^4 s_0^2}{\phi_*^4} \log \frac{2d^2}{\delta} \right\} \,,$$

$$\lambda_t = 4\sigma x_{\max} \left( \sqrt{2w^2 M_0 \log \frac{2d}{\delta}} + 2^{\frac{3}{4}} \sqrt{(t-M_0)\log \frac{7d(\log 2(t-M_0))^2}{\delta}} \right) \,,$$

$w = \sqrt{\tau/M_0}$,

*then with probability at least $1 - 5\delta$, Algorithm 1 achieves the following total regret,*

$$\sum_{t=1}^{T} reg_t \leq 2x_{\max} b M_0 + I_\tau + I_T \,,$$

*where*

$$I_\tau = \mathcal{O}\left( \frac{\sigma^2}{\Delta_*} \left(\frac{x_{\max}^2 s_0}{\phi_*^2}\right)^{1+\frac{1}{\alpha}} \left(\log d + \log \frac{1}{\delta}\right) \right) \,,$$

$$
I_T = \begin{cases}
\mathcal{O}\left( \frac{1}{(1-\alpha)\Delta_*^\alpha} \left(\frac{\sigma x_{\max}^2 s_0}{\phi_*^2}\right)^{1+\alpha} T^{\frac{1-\alpha}{2}} \left(\log d + \log \frac{\log T}{\delta}\right)^{\frac{1+\alpha}{2}} \right) & \alpha \in (0,1) \,, \\[2mm]
\mathcal{O}\left( \frac{1}{\Delta_*} \left(\frac{\sigma x_{\max}^2 s_0}{\phi_*^2}\right)^2 (\log T) \left(\log d + \log \frac{\log T}{\delta}\right) \right) & \alpha = 1 \,, \\[2mm]
\mathcal{O}\left( \frac{\alpha}{(\alpha-1)^2} \cdot \frac{\sigma^2}{\Delta_*} \left(\frac{x_{\max}^2 s_0}{\phi_*^2}\right)^{1+\frac{1}{\alpha}} \left(\log d + \log \frac{1}{\delta}\right) \right) & \alpha > 1 \,.
\end{cases}
$$

*Proof of Theorem 1.* We prove Theorem 2 by invoking Proposition 1 with $\tau_1 = 0$ and $\tau_2 = \tau$. Observe that $\tau$ satisfies the lower bound condition of $\tau_2$ in Proposition 1 since $\tau_1 = 0$ and $w^2 M_0 = \tau$. We must show that the compatibility constant of $\hat{\mathbf{V}}_{M_0} = \sum_{i=1}^{M_0} w \mathbf{x}_{i,a_i} \mathbf{x}_{i,a_i}^\top$ satisfies the lower bound constraint of the proposition. We first show that $\phi^2(\hat{\mathbf{V}}_{M_0}) \geq \frac{4 x_{\max} s_0}{\Delta_*} \left( \frac{80 x_{\max}^2 s_0}{\phi_*^2} \right)^{\frac{1}{\alpha}} \lambda_{M_0 + \tau}$.
Let $\hat{\boldsymbol{\Sigma}}_e = \frac{1}{M_0} \sum_{t=1}^{M_0} \mathbf{x}_{t,a_t} \mathbf{x}_{t,a_t}^\top$. Since $a_t \sim \text{Unif}([K])$ for $t \leq M_0$, the expected value of $\hat{\boldsymbol{\Sigma}}_e$ is

$$\mathbb{E}\left[ \hat{\boldsymbol{\Sigma}}_e \right] = \mathbb{E}_{\substack{\{\mathbf{x}_k\}_{k=1}^K \sim \mathcal{D}_{\mathcal{X}} \\ a \sim \text{Unif}([K])}} \left[ \mathbf{x}_a \mathbf{x}_a^\top \right] .$$

By the definition of $\rho$, we have $\phi^2 \left( \mathbb{E}_{\substack{\{\mathbf{x}_k\}_{k=1}^K \sim \mathcal{D}_{\mathcal{X}} \\ a \sim \text{Unif}([K])}} \left[ \mathbf{x}_a \mathbf{x}_a^\top \right] \right) \geq \frac{\phi_*^2}{\rho}$. By Lemma 22, with probability at least $1 - 2d^2 \exp \left( \frac{\phi_*^2 M_0}{2048 \rho^2 x_{\max}^4 s_0^2} \right)$, it holds that

$$\phi^2 \left( \hat{\boldsymbol{\Sigma}}_e \right) \geq \frac{\phi_*^2}{2\rho} . \tag{14}$$

Since $M_0 \geq \frac{2048 \rho^2 x_{\max}^4 s_0^2}{\phi_*^2} \log \frac{2d^2}{\delta}$, inequality (14) holds with probability at least $1 - \delta$. Note that $\hat{\mathbf{V}}_{M_0} = \sum_{i=1}^{M_0} w \mathbf{x}_{i,a_i} \mathbf{x}_{i,a_i}^\top = w M_0 \hat{\boldsymbol{\Sigma}}_e$. Therefore, with probability at least $1 - \delta$, the compatibility constant of $\hat{\mathbf{V}}_{M_0}$ is lower bounded as the following:

$$\phi^2 \left( \hat{\mathbf{V}}_{M_0} \right) \geq \frac{\phi_*^2}{2\rho} w M_0 . \tag{15}$$

By the choice of $\tau$ and $w$, we obtain an upper bound of $\lambda_{M_0 + \tau}$.

$$\lambda_{M_0 + \tau} = 4 \sigma x_{\max} \left( \sqrt{2 w^2 M_0 \log \frac{d}{\delta}} + 2^{\frac{3}{4}} \sqrt{\tau \left( 2 \log \log 2\tau + \log \frac{7d}{\delta} \right)} \right)$$

$$\leq 4 \sigma x_{\max} \left( \sqrt{2 w^2 M_0 \left( 2 \log \log 2\tau + \log \frac{7d}{\delta} \right)} + 2^{\frac{3}{4}} \sqrt{w^2 M_0 \left( 2 \log \log 2\tau + \log \frac{7d}{\delta} \right)} \right)$$

$$\leq \frac{25 \sigma x_{\max} w}{2} \sqrt{M_0 \left( 2 \log \log 2\tau + \log \frac{7d}{\delta} \right)} , \tag{16}$$

where the first inequality is due to $\log \frac{d}{\delta} \leq 2 \log \log 2\tau + \log \frac{7d}{\delta}$ and $\tau = w^2 M_0$, and the last inequality is $4 \times \left( \sqrt{2} + 2^{\frac{3}{4}} \right) \leq \frac{25}{2}$. Then, it holds that

$$\frac{4 x_{\max} s_0}{\Delta_*} \left( \frac{80 x_{\max}^2 s_0}{\phi_*^2} \right)^{\frac{1}{\alpha}} \lambda_{M_0 + \tau} \leq \frac{50 \sigma x_{\max}^2 s_0 w}{\Delta_*} \left( \frac{80 x_{\max}^2 s_0}{\phi_*^2} \right)^{\frac{1}{\alpha}} \sqrt{M_0 \left( 2 \log \log 2\tau + \log \frac{7d}{\delta} \right)} \tag{17}$$

$$\leq \frac{\phi_*^2}{2\rho} w M_0$$

$$\leq \phi^2 \left( \hat{\mathbf{V}}_{M_0} \right) , \tag{18}$$

where the first inequality comes from inequality (16), the second inequality holds by the choice of $M_0 \geq \rho^2 \left( \frac{100 \sigma x_{\max}^2 s_0}{\Delta_* \phi_*^2} \right)^2 \left( \frac{80 x_{\max}^2 s_0}{\phi_*^2} \right)^{\frac{2}{\alpha}} \left( 2 \log \log 2\tau + \log \frac{7d}{\delta} \right)$, and the last inequality follows by inequality (15).

On the other hand, by the choice of $w = \sqrt{\frac{\tau}{M_0}}$, $\tau \geq \frac{2048 x_{\max}^4 s_0^2}{\phi_*^4} \log \frac{2d^2}{\delta}$, and $M_0 \geq \frac{2048 \rho^2 x_{\max}^4 s_0^2}{\phi_*^4} \log \frac{2d^2}{\delta}$, it holds that

$$
\begin{aligned}
w M_0 &= \sqrt{\tau M_0} \\
&\geq \sqrt{\left( \frac{2048 x_{\max}^4 s_0^2}{\phi_*^4} \log \frac{2d^2}{\delta} \right) \left( \frac{2048 \rho^2 x_{\max}^4 s_0^2}{\phi_*^4} \log \frac{2d^2}{\delta} \right)} \\
&= \frac{2048 \rho x_{\max}^4 s_0}{\phi_*^4} \log \frac{2d^2}{\delta} \, .
\end{aligned}
$$

Then, we have

$$
\begin{aligned}
\phi^2 \left( \hat{\mathbf{V}}_{M_0} \right) &\geq \frac{\phi_*^2}{2\rho} w M_0 & (19) \\
&\geq \frac{1024 x_{\max}^4 s_0^2}{\phi_*^2} \log \frac{2d^2}{\delta} \\
&\geq 64 x_{\max}^2 s_0 \log \frac{2d^2}{\delta} \\
&\geq 64 x_{\max}^2 s_0 \log \frac{1}{\delta} \, , & (20)
\end{aligned}
$$

where the third inequality holds by Lemma 21. Putting bounds (17)-(18) and (19)-(20) together, we obtain

$$
\phi^2 \left( \hat{\mathbf{V}}_{M_0} \right) \geq \max \left\{ \frac{4 x_{\max} s_0}{\Delta_*} \left( \frac{80 x_{\max}^2 s_0}{\phi_*^2} \right)^{\frac{1}{\alpha}} \lambda_{M_0+\tau}, 64 x_{\max}^2 s_0 \log \frac{1}{\delta} \right\} \, .
$$

Then, the conditions of Proposition 1 are met with $\tau_1 = 0$ and $\tau_2 = \tau$. Take the union bound over the event that $\phi^2 \left( \hat{\mathbf{V}}_{M_0} \right) \geq \frac{\phi_*^2}{2\rho} w M_0$ holds and the event of Proposition 1, which happen with probability at least $1-\delta$ and $1-4\delta$ respectively. Then, with probability at least $1-5\delta$, the cumulative regret from $t = M_0 + 1$ to $T$ is bounded by $I_{\tau_2} + I_T$ in Proposition 1. Since we know the value of $\tau_2 - \tau_1 + 1 = \tau + 1 = \mathcal{O} \left( \frac{\sigma^2}{\Delta_*^2} \left( \frac{x_{\max}^2 s_0}{\phi_*^2} \right)^{2+\frac{2}{\alpha}} \left( \log d + \log \frac{1}{\delta} \right) \right)$, we further bound $I_{\tau_2}$ as follows:

$$
\begin{aligned}
I_{\tau_2} &= 2\Delta_* \left( \frac{80 x_{\max}^2 s_0}{\phi_*^2} \right)^{-1-\frac{1}{\alpha}} (\tau_2 - \tau_1 + 1) + \log \frac{1}{\delta} \\
&= \mathcal{O} \left( \frac{\sigma^2}{\Delta_*} \left( \frac{x_{\max}^2 s_0}{\phi_*^2} \right)^{1+\frac{1}{\alpha}} \left( \log d + \log \frac{1}{\delta} \right) \right) \, .
\end{aligned}
$$

The cumulative regret of the first $M_0$ rounds is bounded by $2 x_{\max} b M_0$, which is the maximum regret possible. The proof is complete by renaming $I_{\tau_2}$ to $I_\tau$. $\qquad \square$

### E.3 PROOF OF THEOREM 3

**Theorem** (Formal version of Theorem 3) *Suppose Assumptions 1-3 hold. Further assume that either Assumption 4 or Assumptions 6-8 hold. Let $\phi_G > 0$ be a constant that depends on the employed assumptions, specifically,*

$$
\phi_G^2 = \begin{cases} \frac{1}{4\xi K} & \text{Under Assumption 4,} \\ \frac{\phi_2^2}{2\nu C_x} & \text{Under Assumptions 6-8.} \end{cases}
$$

*For $\delta \in (0, 1]$, let $\tau$ be the least even integer that satisfies*

$$
\tau \geq \max \left\{ C_3 \log \frac{7d}{\delta} + 2 C_3 \log \log \frac{28 d C_3^2}{\delta}, \frac{4096 x_{\max}^4 s_0^2}{\phi_G^4} \left( \log \frac{d^2}{\delta} + 2 \log \frac{64 x_{\max}^2 s_0}{\phi_G^2} \right) + 2 \right\} \, ,
$$

where $C_3 = \max\left\{2, \left(\frac{108\sigma x_{\max}^2 s_0}{\Delta_* \phi_G^2}\right)^2 \left(\frac{80 x_{\max}^2 s_0}{\phi_*^2}\right)^{\frac{2}{\alpha}}\right\}$. *If we set the input parameters of Algorithm 1*

*by $M_0 = 0$ and $\lambda_t = 2^{\frac{11}{4}} \sigma x_{\max} \sqrt{t \log \frac{7d(\log 2t)^2}{\delta}}$, then with probability at least $1 - 5\delta$, Algorithm 1 achieves the following total regret.*

$$\sum_{t=1}^T reg_t \leq \begin{cases} I_b + I_2(T) & T \leq \tau + 1 \\ I_b + I_2(\tau + 1) + I_T & T > \tau + 1, \end{cases}$$

*where*

$$I_b = 2x_{\max} b \left(\frac{2048 x_{\max}^4 s_0^2}{\phi_G^2} \left(\log \frac{d^2}{\delta} + 2\log \frac{64 x_{\max}^2 s_0}{\phi_G^2}\right) + 4\log\frac{1}{\delta}\right),$$

$$I_2(T) = \begin{cases} \mathcal{O}\left(\frac{1}{(1-\alpha)\Delta_*^\alpha}\left(\frac{\sigma x_{\max}^2 s_0}{\phi_G^2}\right)^{1+\alpha} T^{\frac{1-\alpha}{2}} \left(\log d + \log\frac{1}{\delta}\right)^{\frac{1+\alpha}{2}}\right) & \alpha \in [0, 1), \\ \mathcal{O}\left(\frac{\sigma^2}{\Delta_*}\left(\frac{x_{\max}^2 s_0}{\phi_G^2}\right)^2 (\log T)\left(\log d + \log\frac{\log T}{\delta}\right)\right) & \alpha = 1, \\ \mathcal{O}\left(\frac{\alpha^2}{(\alpha-1)^2} \cdot \frac{\sigma^2}{\Delta_*}\left(\frac{x_{\max}^2 s_0}{\phi_G^2}\right)^2 \left(\log d + \log\frac{1}{\delta}\right)\right) & \alpha > 1, \end{cases}$$

$$I_T = \begin{cases} \mathcal{O}\left(\frac{1}{(1-\alpha)\Delta_*^\alpha}\left(\frac{\sigma x_{\max}^2 s_0}{\phi_*^2}\right)^{1+\alpha} T^{\frac{1-\alpha}{2}} \left(\log d + \log\frac{\log T}{\delta}\right)^{\frac{1+\alpha}{2}}\right) & \alpha \in (0, 1), \\ \mathcal{O}\left(\frac{1}{\Delta_*}\left(\frac{\sigma x_{\max}^2 s_0}{\phi_*^2}\right)^2 (\log T)\left(\log d + \log\frac{\log T}{\delta}\right)\right) & \alpha = 1, \\ \mathcal{O}\left(\frac{\alpha}{(\alpha-1)^2} \cdot \frac{\sigma^2}{\Delta_*}\left(\frac{x_{\max}^2 s_0}{\phi_*^2}\right)^{1+\frac{1}{\alpha}} \left(\log d + \log\frac{1}{\delta}\right)\right) & \alpha > 1. \end{cases}$$

*Proof of Theorem 3.* From Lemma 1 and Lemma 3, we know that the greedy diversity, defined in Definition 2, holds with compatibility constant $\phi_G$. Let $\tau_0 = \frac{2048 x_{\max}^4 s_0^2}{\phi_G^4}\left(\log\frac{d^2}{\delta} + 2\log\frac{64 x_{\max}^2 s_0}{\phi_G^2}\right)$. We present a lemma about the greedy diversity.

**Lemma 10.** *Under the greedy diversity (Definition 2), suppose Algorithm 1 runs with $M_0 = 0$. Define the empirical Gram matrix as $\hat{\mathbf{V}}_t = \sum_{i=1}^t \mathbf{x}_{i,a_i} \mathbf{x}_{i,a_i}^\top$. For $\delta \in (0, 1]$, let $\mathcal{E}_{GD}$ be the event that the compatibility constant of the empirical Gram matrix being lower bounded for big enough $t$. Specifically,*

$$\mathcal{E}_{GD} = \left\{\omega \in \Omega : \forall t \geq \tau_0 + 1, \phi^2\left(\hat{\mathbf{V}}_t, S_0\right) \geq \frac{\phi_G^2 t}{2}\right\}.$$

*Then, we have $\mathbb{P}\left(\mathcal{E}_{GD}\right) \geq 1 - \delta$.*

We prove the theorem under the events $\mathcal{E}_{GD}$, $\mathcal{E}_g$, $\mathcal{E}_N(\tau_0)$, $\mathcal{E}_N(\tau)$, and $\mathcal{E}^*(\frac{1}{2}\tau, \tau)$. By Lemma 10 and Lemma 13-15, each of the events holds with probability at least $1 - \delta$, and by the union bound, all the events happen with probability at least $1 - 5\delta$. Next lemma states a regret bound of Algorithm 1 that is independent of Assumption 3.

**Lemma 11.** *Suppose Assumptions 1, 2 hold and $\mathcal{D}_{\mathcal{X}}$ satisfies the greedy diversity (Definition 2). Suppose Algorithm 1 runs as in Theorem 3. Then, under the events $\mathcal{E}_{GD}$, $\mathcal{E}_g$, and $\mathcal{E}_N(\tau_0)$, the cumulative regret is bounded as the following:*

$$\sum_{t=1}^T reg_t \leq I_b + I_2(T),$$

*where*

$$I_b = 2x_{\max} b \left( \frac{2048 x_{\max}^4 s_0^2}{\phi_G^2} \left( \log \frac{d^2}{\delta} + 2 \log \frac{64 x_{\max}^2 s_0}{\phi_G^2} \right) + 4 \log \frac{1}{\delta} \right),$$

$$I_2(T) = \begin{cases} \mathcal{O}\left( \frac{1}{(1-\alpha)\Delta_*^\alpha} \left( \frac{\sigma x_{\max}^2 s_0}{\phi_G^2} \right)^{1+\alpha} T^{\frac{1-\alpha}{2}} \left( \log d + \log \frac{1}{\delta} \right)^{\frac{1+\alpha}{2}} \right) & \alpha \in [0,1), \\ \mathcal{O}\left( \frac{1}{\Delta_*} \left( \frac{\sigma x_{\max}^2 s_0}{\phi_G^2} \right)^2 (\log T) \left( \log d + \log \frac{\log T}{\delta} \right) \right) & \alpha = 1, \\ \mathcal{O}\left( \frac{\alpha^2}{(\alpha-1)^2 \Delta_*} \left( \frac{\sigma x_{\max}^2 s_0}{\phi_G^2} \right)^2 \left( \log d + \log \frac{1}{\delta} \right) \right) & \alpha > 1. \end{cases}$$

We can assume that $\phi_*^2 \geq \phi_G^2$ by the Remark 6. If $\phi_* \approx \phi_G$, or specifically $\phi_*^2 \leq 8\phi_G^2$, then Theorem 3 reduces to Lemma 11 by replacing $\phi_*$ with $\phi_G$ and adjusting the constant factors appropriately. Lemma 11 is also sufficient to prove the theorem when $T \leq \tau + 1$. We suppose $\phi_*^2 \geq 8\phi_G^2$ and $T > \tau + 1$ from now on.

We invoke Proposition 1 with $\tau_1 = \frac{1}{2}\tau$ and $\tau_2 = \tau$. We must first show that $\tau$ satisfies the lower bound condition of $\tau_2$ in Proposition 1. Since we suppose $\phi_*^2 \geq 8\phi_G^2$, $C_3$ in the statement of Theorem 3 is greater than $C_2$ in the statement of Proposition 1. Hence, we have $\tau \geq C_2 \log \frac{7d}{\delta} + 2C_2 \log \log \frac{28dC_2^2}{\delta}$. $\tau$ trivially satisfies the rest of the lower bound conditions of $\tau_2$ when $\tau_1 = \frac{1}{2}\tau$ and $M_0 = 0$. Now, we must show that $\phi^2 \left( \hat{\mathbf{V}}_{\frac{1}{2}\tau}, S_0 \right)$ satisfies the lower bound constraint in Proposition 1. As we have chosen $\tau$ to satisfy $\tau \geq \frac{4096 x_{\max}^4 s_0^2}{\phi_G^4} \left( \log \frac{d^2}{\delta} + 2 \log \frac{64 x_{\max}^2 s_0}{\phi_G^2} \right) + 2$, we have $\frac{1}{2}\tau \geq \frac{2048 x_{\max}^4 s_0^2}{\phi_G^4} \left( \log \frac{d^2}{\delta} + 2 \log \frac{64 x_{\max}^2 s_0}{\phi_G^2} \right) + 1 = \tau_0 + 1$. Then, under the event $\mathcal{E}_{\text{GD}}$, $\phi^2 \left( \hat{\mathbf{V}}_{\frac{1}{2}\tau} \right) \geq \frac{\phi_G^2 \tau}{4}$ holds. By the choice of $\tau$ and Lemma 25, we have

$$\frac{2 \log \log 2\tau + \log \frac{7d}{\delta}}{\tau} \leq \left( \frac{\Delta_* \phi_G^2}{108 \sigma x_{\max}^2 s_0} \right)^2 \left( \frac{\phi_*^2}{80 x_{\max}^2 s_0} \right)^{\frac{2}{\alpha}}.$$

Then, we have

$$\lambda_\tau = 2^{\frac{11}{4}} \sigma x_{\max} \sqrt{\tau \log \frac{7d(\log 2\tau)^2}{\delta}}$$

$$= 2^{\frac{11}{4}} \sigma x_{\max} \tau \sqrt{\frac{2 \log \log 2\tau + \log \frac{7d}{\delta}}{\tau}}$$

$$\leq 2^{\frac{11}{4}} \sigma x_{\max} \tau \left( \frac{\Delta_* \phi_G^2}{108 \sigma x_{\max}^2 s_0} \right) \left( \frac{\phi_*^2}{80 x_{\max}^2 s_0} \right)^{\frac{1}{\alpha}}$$

$$= \frac{\Delta_* \phi_G^2 \tau}{16 x_{\max} s_0} \left( \frac{\phi_*^2}{80 x_{\max}^2 s_0} \right)^{\frac{1}{\alpha}}.$$

Therefore, it holds that

$$\frac{4 x_{\max} s_0}{\Delta_*} \left( \frac{80 x_{\max}^2 s_0}{\phi_*^2} \right)^{\frac{1}{\alpha}} \lambda_\tau \leq \frac{\phi_G^2 \tau}{4} \tag{21}$$

$$\leq \phi^2 \left( \hat{\mathbf{V}}_{\frac{1}{2}\tau} \right). \tag{22}$$

On the other hand, by $\tau \geq \frac{4096 x_{\max}^4 s_0}{\phi_G^4} \left( \log \frac{d^2}{\delta} + 2 \log \frac{64 x_{\max}^2 s_0}{\phi_G^2} \right)$, we have

$$\phi^2 \left( \hat{\mathbf{V}}_{\frac{1}{2}\tau} \right) \geq \frac{\phi_G^2 \tau}{4} \tag{23}$$

$$\geq \frac{1024 x_{\max}^4 s_0^2}{\phi_G^2} \left( \log \frac{d^2}{\delta} + 2 \log \frac{64 x_{\max}^2 s_0}{\phi_G^2} \right)$$

$$\geq 64 x_{\max}^2 s_0 \log \frac{1}{\delta}, \tag{24}$$

where the last inequality holds by Lemma 21. Putting inequalities (21)-(22) and (23)-(24) together, we obtain

$$\phi^2 \left( \hat{\mathbf{V}}_{\frac{1}{2}\tau} \right) \geq \max \left\{ \frac{4x_{\max}s_0}{\Delta_*} \left( \frac{80x_{\max}^2 s_0}{\phi_*^2} \right)^{\frac{1}{\alpha}} \lambda_\tau, 64x_{\max}^2 s_0 \log \frac{1}{\delta} \right\} .$$

Then, the conditions of Proposition 1 hold with $\tau_1 = \frac{1}{2}\tau$ and $\tau_2 = \tau$. By the first part of Proposition 1, we obtain

$$\left\| \boldsymbol{\beta}^* - \hat{\boldsymbol{\beta}}_t \right\|_1 \leq \frac{200\sigma x_{\max}s_0}{\phi_*^2} \sqrt{\frac{2 \log \log t + \frac{7d}{\delta}}{t}}$$

for $t > \tau$. On the other hand, by inequality (33) from the proof of Lemma 11, we obtain

$$\left\| \boldsymbol{\beta}^* - \hat{\boldsymbol{\beta}}_t \right\|_1 \leq \frac{27\sigma x_{\max}s_0}{\phi_G^2} \sqrt{\frac{2 \log \log 2t + \log \frac{7d}{\delta}}{t}}$$

for $t \geq \tau_0 + 1$. Define $\overline{\Delta}_t$ as follows:

$$\overline{\Delta}_t = \begin{cases} \frac{54\sigma x_{\max}^2 s_0}{\phi_G^2} \sqrt{\frac{2 \log \log 2t + \log \frac{7d}{\delta}}{t}} & t \leq \tau \\ \frac{400\sigma x_{\max}^2 s_0}{\phi_*^2} \sqrt{\frac{2 \log \log t + \frac{7d}{\delta}}{t}} & t > \tau . \end{cases}$$

Then, $2x_{\max} \left\| \boldsymbol{\beta}^* - \hat{\boldsymbol{\beta}}_t \right\|_1 \leq \overline{\Delta}_t$ holds for all $t \geq \tau_0 + 1$, and $\overline{\Delta}_t$ is decreasing in $t$ since we assumed that $\phi_*^2 \geq 8\phi_G^2$. By Lemma 9, it holds that

$$\sum_{t=\tau_0+1}^{T} \operatorname{reg}_t \leq 4\overline{\Delta}_{\tau_0} \log \frac{1}{\delta} + \frac{5}{4} \sum_{t=\tau_0}^{T-1} \overline{\Delta}_t \min \left\{ 1, \left( \frac{\overline{\Delta}_t}{\Delta_*} \right)^\alpha \right\} . \tag{25}$$

Following the proof of Proposition 1, especially inequality (13), we obtain that

$$\frac{5}{4} \sum_{t=\tau+1}^{T-1} \overline{\Delta}_t \left( \frac{\overline{\Delta}_t}{\Delta_*} \right)^\alpha \leq I_T .$$

Following the proof of Lemma 11, we observe that

$$\sum_{t=1}^{\tau} \operatorname{reg}_t \leq \sum_{t=1}^{\tau_0} \operatorname{reg}_t + 4\overline{\Delta}_{\tau_0} \log \frac{1}{\delta} + \frac{5}{4} \sum_{t=\tau_0}^{\tau} \overline{\Delta}_t \min \left\{ 1, \left( \frac{\overline{\Delta}_t}{\Delta_*} \right)^\alpha \right\}$$

$$\leq 2x_{\max}b \left( \tau_0 + 4 \log \frac{1}{\delta} \right) + I_2(\tau+1) . \tag{26}$$

Combining Eq. (25) and (26), we conclude that

$$\sum_{t=1}^{T} \operatorname{reg}_t \leq 2x_{\max}b \left( \tau_0 + 4 \log \frac{1}{\delta} \right) + I_2(\tau+1) + I_T .$$

$\square$

### E.4 Proof of Technical Lemmas in Appendix E.1-E.3

#### E.4.1 High Probability Events

We prove that the events assumed in the proof of Proposition 1 hold with high probability. Recall the definitions of the events.

$$\mathcal{E}_e = \left\{ \omega \in \Omega : \max_{j \in [d]} \left| \sum_{i=1}^{M_0} \eta_i \left( \mathbf{x}_{i,a_i} \right)_j \right| \leq \sigma x_{\max} \sqrt{2 M_0 \log \frac{d}{\delta}} \right\},$$

$$\mathcal{E}_g = \left\{ \omega \in \Omega : \forall n \geq 1, \max_{j \in [d]} \left| \sum_{i=M_0+1}^{M_0+n} \eta_i \left( \mathbf{x}_{i,a_i} \right)_j \right| \leq 2^{\frac{3}{4}} \sigma x_{\max} \sqrt{n \log \frac{7d \left( \log 2n \right)^2}{\delta}} \right\},$$

$$\mathcal{E}_N(n) = \left\{ \omega \in \Omega : \forall t' \geq 0, N_n(t') \leq \frac{5}{4} \sum_{i=M_0+n+1}^{M_0+n+t'} \min \left\{ 1, \left( \frac{2 x_{\max}}{\Delta_*} \left\| \boldsymbol{\beta}^* - \boldsymbol{\beta}_{i-1} \right\|_1 \right)^\alpha \right\} + 4 \log \frac{1}{\delta} \right\},$$

$$\mathcal{E}^*(\tau_1, \tau_2) = \left\{ \omega \in \Omega : \forall t' \geq \tau_2 - \tau_1 + 1, \phi^2 \left( \sum_{t=M_0+\tau_1+1}^{M_0+\tau_1+t'} \mathbf{x}_{t,a_t^*} \mathbf{x}_{t,a_t^*}^\top \right) \geq \frac{\phi_*^2 t'}{2} \right\}.$$

**Lemma 12.** *We have* $\mathbb{P}\left(\mathcal{E}_e\right) \geq 1 - \delta$.

*Proof of Lemma 12.* Recall that $\mathcal{F}_t$ is the $\sigma$-algebra generated by $\left( \{\mathbf{x}_{\tau,i}\}_{\tau \in [t], i \in [K]}, \{a_\tau\}_{\tau \in [t]}, \{r_{\tau,a_\tau}\}_{\tau \in [t-1]} \right)$. Fix $j \in [d]$. By sub-Gaussianity of $\eta_t$, $\mathbb{E}\left[e^{s\eta_t} \mid \mathcal{F}_t\right] \leq e^{\frac{s^2 \sigma^2}{2}}$ for all $s \in \mathbb{R}$. Since $(\mathbf{x}_{t,a_t})_j$ is $\mathcal{F}_t$-measurable, we get $\mathbb{E}\left[e^{s\eta_t(\mathbf{x}_{t,a_t})_j} \mid \mathcal{F}_t\right] \leq e^{s^2(\mathbf{x}_{t,a_t})_j^2 \sigma^2/2} \leq e^{s^2 x_{\max}^2 \sigma^2/2}$. Therefore, $\{\eta_t(\mathbf{x}_{t,a_t})_j\}_{t=1}^{M_0}$ is a sequence of conditionally $\sigma x_{\max}$-sub-Gaussian random variables. Then, by the Azuma-Hoeffding inequality, we have

$$\mathbb{P}\left( \left| \sum_{t=1}^{M_0} \eta_t(\mathbf{x}_{t,a_t})_j \right| \leq \sigma x_{\max} \sqrt{2 M_0 \log \frac{2}{\delta}} \right) \leq \delta.$$

Take the union bound over $j \in [d]$ and obtain

$$\mathbb{P}\left(\mathcal{E}_e^{\mathsf{c}}\right) = \mathbb{P}\left( \max_{j \in [d]} \left| \sum_{t=1}^{M_0} \eta_t(\mathbf{x}_{t,a_t})_j \right| \leq \sigma x_{\max} \sqrt{2 M_0 \log \frac{2d}{\delta}} \right)$$

$$\leq \sum_{j=1}^{d} \mathbb{P}\left( \left| \sum_{t=1}^{M_0} \eta_t(\mathbf{x}_{t,a_t})_j \right| \leq \sigma x_{\max} \sqrt{2 M_0 \log \frac{2d}{\delta}} \right)$$

$$\leq \delta.$$

$\square$

**Lemma 13.** *We have* $\mathbb{P}\left(\mathcal{E}_g\right) \geq 1 - \delta$.

*Proof of Lemma 13.* Fix $j \in [d]$. Following the same argument as in the proof of Lemma 12, $\{\eta_t(\mathbf{x}_{t,a_t})_j\}_{t=M_0+1}^{\infty}$ is a sequence of conditionally $\sigma x_{\max}$-sub-Gaussian random variables. By Lemma 27, it holds that

$$\mathbb{P}\left( \left| \sum_{i=M_0+1}^{M_0+t'} \eta_i(\mathbf{x}_{i,a_i})_j \right| \geq 2^{\frac{3}{4}} \sigma x_{\max} \sqrt{t' \log \frac{7(\log 2t')^2}{\delta}} \right) \leq \delta.$$

Taking the union bound over $j \in [d]$ concludes the proof. $\square$

**Lemma 14.** *For any* $n \in \mathbb{N}_0$, *we have* $\mathbb{P}\left(\mathcal{E}_N(n)\right) \geq 1 - \delta$.

*Proof of Lemma 14.* Let $Y_i = \mathbb{1}\left\{a_{M_0+n+i} \neq a^*_{M_0+n+i}\right\}$. Define $\mathcal{F}^+_t$ to be the $\sigma$-algebra generated by $\left(\{\mathbf{x}_{\tau,i}\}_{\tau\in[t],i\in[K]}, \{a_\tau\}_{\tau\in[t]}, \{r_{\tau,a_\tau}\}_{\tau\in[t]}\right)$. Note that the only difference between $\mathcal{F}_t$ and $\mathcal{F}^+_t$ is that $\mathcal{F}^+_t$ is also generated by $r_{t,a_t}$. $Y_i$ is $\mathcal{F}^+_{M_0+n+i}$-measurable. By Lemma 29, with probability at least $1-\delta$, the following holds for all $t' \geq 1$:

$$\sum_{i=1}^{t'} Y_i \leq \frac{5}{4}\sum_{i=1}^{t'}\mathbb{E}\left[Y_i \mid \mathcal{F}^+_{M_0+n+i-1}\right] + 4\log\frac{1}{\delta}. \tag{27}$$

By Lemma 24, $Y_i = 1$ happens only when $\Delta_{t_i} \leq 2x_{\max}\left\|\boldsymbol{\beta}^* - \hat{\boldsymbol{\beta}}_{t_i-1}\right\|_1$, where $t_i = M_0 + n + i$. By Assumption 2, $\mathbb{P}\left(\Delta_{t_i} \leq 2x_{\max}\left\|\boldsymbol{\beta}^* - \hat{\boldsymbol{\beta}}_{t_i-1}\right\|_1 \mid \mathcal{F}^+_{t_i-1}\right) \leq \left(\frac{2x_{\max}}{\Delta_*}\left\|\boldsymbol{\beta}^* - \hat{\boldsymbol{\beta}}_{t_i-1}\right\|_1\right)^\alpha$, where we use the fact that $\hat{\boldsymbol{\beta}}_{t_i-1}$ is $\mathcal{F}^+_{t_i-1}$-measurable and $\Delta_t$ is independent of $\mathcal{F}^+_{t_i-1}$. Then, we have

$$\begin{aligned}
\mathbb{E}\left[Y_i \mid \mathcal{F}^+_{t_i-1}\right] &= \mathbb{P}\left(Y_i = 1 \mid \mathcal{F}^+_{t_i-1}\right) \\
&\leq \mathbb{P}\left(\Delta_{t_i} \leq 2x_{\max}\left\|\boldsymbol{\beta}^* - \hat{\boldsymbol{\beta}}_{t_i-1}\right\|_1 \mid \mathcal{F}^+_{t_i-1}\right) \\
&\leq \left(\frac{2x_{\max}}{\Delta_*}\left\|\boldsymbol{\beta}^* - \hat{\boldsymbol{\beta}}_{t_i-1}\right\|_1\right)^\alpha.
\end{aligned}$$

On the other hand, $\mathbb{E}\left[Y_i \mid \mathcal{F}^+_{t_i-1}\right]$ has a trivial upper bound of 1. Therefore, we deduce that

$$\mathbb{E}\left[Y_i \mid \mathcal{F}^+_{t_i-1}\right] \leq \min\left\{1, \left(\frac{2x_{\max}}{\Delta_*}\left\|\boldsymbol{\beta}^* - \hat{\boldsymbol{\beta}}_{t_i-1}\right\|_1\right)^\alpha\right\} \tag{28}$$

Plug in inequality (28) to (27) and we obtain the desired result. $\qquad\square$

**Lemma 15.** *If* $\tau_2 \geq \tau_1 + \frac{2048x^4_{\max}s_0^2}{\phi^4_*}\left(\log\frac{d^2}{\delta} + 2\log\frac{64x^2_{\max}s_0}{\phi^2_*}\right)$, *then we have* $\mathbb{P}\left(\mathcal{E}^*(\tau_1,\tau_2)\right) \geq 1-\delta$.

*Proof of Lemma 15.* Let $\hat{\mathbf{V}}^*_{t'} = \sum_{t=M_0+\tau_1+1}^{M_0+\tau_1+t'}\mathbf{x}_{t,a^*_t}\mathbf{x}^\top_{t,a^*_t}$. Note that $\mathbb{E}\left[\hat{\mathbf{V}}^*_{t'}\right] = \sum_{t=M_0+\tau_1+1}^{M_0+\tau_1+t'}\mathbb{E}\left[\mathbf{x}_*\mathbf{x}^\top_*\right] = t'\Sigma^*$. By Assumption 3, $\phi^2\left(\mathbb{E}\left[\hat{\mathbf{V}}^*_{t'}\right], S_0\right) \geq \phi^2_* t'$. By Lemma 23, with probability at least $1-\delta$, $\phi^2\left(\hat{\mathbf{V}}^*_{t'}, S_0\right) \geq \frac{\phi^2_* t'}{2}$ holds for all $t' \geq \frac{2048x^4_{\max}s_0^2}{\phi^4_*}\left(\log\frac{d^2}{\delta} + 2\log\frac{64x^2_{\max}s_0}{\phi^2_*}\right) + 1$. Since $\tau_2 \geq \tau_1 + \frac{2048x^4_{\max}s_0^2}{\phi^4_*}\left(\log\frac{d^2}{\delta} + 2\log\frac{64x^2_{\max}s_0}{\phi^2_*}\right)$, $t' \geq \tau_2 - \tau_1 + 1$ implies $t' \geq \frac{2048x^4_{\max}s_0^2}{\phi^4_*}\left(\log\frac{d^2}{\delta} + 2\log\frac{64x^2_{\max}s_0}{\phi^2_*}\right) + 1$. Therefore, we conclude that $\mathcal{E}^*(\tau_1,\tau_2) \geq 1-\delta$. $\qquad\square$

### E.4.2 Proof of Lemma 6

*Proof of Lemma 6.* We apply Lemma 19, using the constraints of $\phi^2\left(\hat{\mathbf{V}}_{M_0+\tau_1}, S_0\right)$. Under the events $\mathcal{E}_e$ and $\mathcal{E}_g$, it holds that for $t \geq M_0$,

$$\begin{aligned}
&\max_{j\in[d]}\left|\sum_{i=1}^{M_0} w\eta_i(\mathbf{x}_{i,a_i})_j + \sum_{i=M_0+1}^{t}\eta_i(\mathbf{x}_{i,a_i})_j\right| \\
&\leq \max_{j\in[d]} w\left|\sum_{i=1}^{M_0}\eta_i(\mathbf{x}_{i,a_i})_j\right| + \max_{j\in[d]}\left|\sum_{i=M_0+1}^{t}\eta_i(\mathbf{x}_{i,a_i})_j\right| \\
&\leq \sigma x_{\max}\left(w\sqrt{2M_0\log\frac{2d}{\delta}} + 2^{\frac{3}{4}}\sqrt{(t-M_0)\log\frac{7d(\log 2(t-M_0))^2}{\delta}}\right),
\end{aligned}$$

which implies

$$\max_{j\in[d]}\left|\sum_{i=1}^{M_0} w\eta_i(\mathbf{x}_{i,a_i})_j + \sum_{i=M_0+1}^{t}\eta_i(\mathbf{x}_{i,a_i})_j\right| \leq \frac{\lambda_t}{4}. \tag{29}$$

For $t' \geq 0$, we have $\phi^2\left(\hat{\mathbf{V}}_{M_0+\tau_1+t'}, S_0\right) \geq \phi^2\left(\hat{\mathbf{V}}_{M_0+\tau_1}, S_0\right) \geq \frac{4x_{\max}s_0}{\Delta_*}\left(\frac{80x_{\max}^2 s_0}{\phi_*^2}\right)^{\frac{1}{\alpha}}\lambda_{M_0+\tau_2}$
by the condition of Proposition 1. By Lemma 19, it holds that

$$
\begin{aligned}
\left\|\boldsymbol{\beta}^* - \hat{\boldsymbol{\beta}}_{M_0+\tau_1+t'}\right\|_1 &\leq \frac{2s_0\lambda_{M_0+\tau_1+t'}}{\frac{4x_{\max}s_0}{\Delta_*}\left(\frac{80x_{\max}^2 s_0}{\phi_*^2}\right)^{\frac{1}{\alpha}}\lambda_{M_0+\tau_2}} \\
&\leq \frac{2s_0}{\frac{4x_{\max}s_0}{\Delta_*}\left(\frac{80x_{\max}^2 s_0}{\phi_*^2}\right)^{\frac{1}{\alpha}}} \\
&= \frac{\Delta_*}{2x_{\max}}\left(\frac{\phi_*^2}{80x_{\max}^2 s_0}\right)^{\frac{1}{\alpha}},
\end{aligned}
$$

where the second inequality holds since $\lambda_t$ is increasing in $t$ and $t' \leq \tau_2 - \tau_1$. $\qquad\square$

### E.4.3 PROOF OF LEMMA 7

*Proof of Lemma 7.* Decompose $\hat{\mathbf{V}}_{M_0+\tau_1+t'}$ as follows:

$$
\begin{aligned}
\hat{\mathbf{V}}_{M_0+\tau_1+t'} &= \hat{\mathbf{V}}_{M_0+\tau_1} + \sum_{i=M_0+\tau_1+1}^{M_0+\tau_1+t'} \mathbf{x}_{i,a_i}\mathbf{x}_{i,a_i}^\top \\
&= \hat{\mathbf{V}}_{M_0+\tau_1} + \sum_{i=M_0+\tau_1+1}^{M_0+\tau_1+t'}\left(\mathbf{x}_{i,a_i}\mathbf{x}_{i,a_i}^\top - \mathbf{x}_{i,a_i^*}\mathbf{x}_{i,a_i^*}^\top\right) + \sum_{i=M_0+\tau_1+1}^{M_0+\tau_1+t'}\mathbf{x}_{i,a_i^*}\mathbf{x}_{i,a_i^*}^\top \\
&= \hat{\mathbf{V}}_{M_0+\tau_1} + \sum_{i=M_0+\tau_1+1}^{M_0+\tau_1+t'}\mathbb{1}\left\{a_i \neq a_i^*\right\}\left(\mathbf{x}_{i,a_i}\mathbf{x}_{i,a_i}^\top - \mathbf{x}_{i,a_i^*}\mathbf{x}_{i,a_i^*}^\top\right) + \sum_{i=M_0+\tau_1+1}^{M_0+\tau_1+t'}\mathbf{x}_{i,a_i^*}\mathbf{x}_{i,a_i^*}^\top \\
&= \hat{\mathbf{V}}_{M_0+\tau_1} + \sum_{i=M_0+\tau_1+1}^{M_0+\tau_1+t'}\mathbb{1}\left\{a_i \neq a_i^*\right\}\mathbf{x}_{i,a_i}\mathbf{x}_{i,a_i}^\top - \sum_{i=M_0+\tau_1+1}^{M_0+\tau_1+t'}\mathbb{1}\left\{a_i \neq a_i^*\right\}\mathbf{x}_{i,a_i^*}\mathbf{x}_{i,a_i^*}^\top \\
&\quad + \sum_{i=M_0+\tau_1+1}^{M_0+\tau_1+t'}\mathbf{x}_{i,a_i^*}\mathbf{x}_{i,a_i^*}^\top.
\end{aligned}
$$

Note that $\phi^2\left(\hat{\mathbf{V}}_{M_0+\tau_1}, S_0\right) \geq 64x_{\max}^2 s_0 \log\frac{1}{\delta}$ holds by the assumption of Proposition 1. By Lemma 21, $\phi^2\left(\sum_{i=M_0+\tau_1+1}^{M_0+\tau_1+t'}\mathbb{1}\{a_i \neq a_i^*\}\mathbf{x}_{i,a_i}\mathbf{x}_{i,a_i}^\top, S_0\right)$ and $\phi^2\left(-\sum_{i=M_0+\tau_1+1}^{M_0+\tau_1+t'}\mathbb{1}\{a_i \neq a_i^*\}\mathbf{x}_{i,a_i^*}\mathbf{x}_{i,a_i^*}^\top, S_0\right)$ are lower bounded by $0$ and $-16x_{\max}^2 s_0 N_{\tau_1}(t')$ respectively. Under the event $\mathcal{E}^*(\tau_1, \tau_2)$, $\phi^2\left(\sum_{i=M_0+\tau_1+1}^{M_0+\tau_1+t'}\mathbf{x}_{i,a_i^*}\mathbf{x}_{i,a_i^*}^\top, S_0\right) \geq \frac{\phi_*^2 t'}{2}$ holds when $t' > \tau_2 - \tau_1$. By combining the lower bounds and by concavity of compatibility constant (Lemma 20), we have

$$
\phi^2\left(\hat{\mathbf{V}}_{M_0+\tau_1+t'}\right) \geq 64x_{\max}^2 s_0 \log\frac{1}{\delta} - 16x_{\max}^2 s_0 N_{\tau_1}(t') + \frac{\phi_*^2 t'}{2}. \tag{30}
$$

Under the event $\mathcal{E}_N(\tau_1)$, we have $N_{\tau_1}(t') \leq \frac{5}{4}\overline{N}(t') + 4\log\frac{1}{\delta}$. We supposed that $\overline{N}(t') \leq \frac{\phi_*^2}{80x_{\max}^2 s_0}t'$. Combining these facts, we have $N_{\tau_1}(t') \leq \frac{\phi_*^2}{64x_{\max}^2 s_0}t' + 4\log\frac{1}{\delta}$. Then, together with Eq. (30), $\phi^2\left(\hat{\mathbf{V}}_{M_0+\tau_1+t'}\right) \geq \frac{\phi_*^2}{4}t'$ holds.

On the other hand, since $t' > \tau_2 - \tau_1 \geq \tau_1$, it holds that $t' \geq \frac{\tau_1+t'}{2}$. Then, we obtain the following lower bound of $\phi^2\left(\hat{\mathbf{V}}_{M_0+\tau_1+t'}\right)$:

$$
\phi^2\left(\hat{\mathbf{V}}_{M_0+\tau_1+t'}\right) \geq \phi^2\left(\hat{\mathbf{V}}_{M_0+\tau_1+\frac{\tau_1+t'}{2}}\right) \geq \frac{\phi_*^2}{8}(\tau_1+t').
$$

As shown in Eq. (29), under the events $\mathcal{E}_e$, $\mathcal{E}_g$, it holds that $\max_{j\in[d]}\left|\sum_{i=1}^{M_0}w\eta_i(\mathbf{x}_{i,a_i})_j + \sum_{i=M_0+1}^{t}\eta_i(\mathbf{x}_{i,a_i})_j\right| \leq \frac{\lambda_t}{4}$. Therefore, by Lemma 19, we

have that

$$\left\|\boldsymbol{\beta}^* - \hat{\boldsymbol{\beta}}_{M_0+\tau_1+t'}\right\|_1 \leq \frac{2s_0\lambda_{M_0+\tau_1+t'}}{\frac{\phi_*^2}{8}(\tau_1+t')}$$

$$= \frac{64\sigma x_{\max}s_0}{\phi_*^2(\tau_1+t')} \left(\sqrt{2w^2 M_0 \log \frac{2d}{\delta}} + 2^{\frac{3}{4}}\sqrt{(\tau_1+t')(2\log\log 2(\tau_1+t') + \log\frac{7d}{\delta})}\right).$$

From $w^2 M_0 \leq \tau_2 \leq \tau_1 + t'$ and $\log\frac{2d}{\delta} \leq 2\log\log 2(\tau_1+t') + \log\frac{7d}{\delta}$, we obtain

$$\left\|\boldsymbol{\beta}^* - \hat{\boldsymbol{\beta}}_{M_0+\tau_1+t'}\right\|_1 \leq \frac{64\sigma x_{\max}s_0}{\phi_*^2(\tau_1+t')} \left(\sqrt{2w^2 M_0 \log \frac{2d}{\delta}} + 2^{\frac{3}{4}}\sqrt{(\tau_1+t')(2\log\log 2(\tau_1+t') + \log\frac{7d}{\delta})}\right)$$

$$\leq \frac{64\sigma x_{\max}s_0}{\phi_*^2(\tau_1+t')} \left(\left(\sqrt{2} + 2^{\frac{3}{4}}\right)\sqrt{(\tau_1+t')(2\log\log 2(\tau_1+t') + \log\frac{7d}{\delta})}\right)$$

$$\leq \frac{200\sigma x_{\max}s_0}{\phi_*^2} \sqrt{\frac{2\log\log 2(\tau_1+t') + \log\frac{7d}{\delta}}{\tau_1+t'}},$$

where the last inequality used the fact $64 \times \left(\sqrt{2} + 2^{\frac{3}{4}}\right) \leq 200$. $\qquad\square$

### E.4.4 PROOF OF LEMMA 8

*Proof of Lemma 8.* By Lemma 6, for $1 \leq t' \leq \tau_2 - \tau_1 + 1$, it holds that

$$\overline{N}(t') \leq \sum_{t=M_0+\tau_1+1}^{M_0+\tau_1+t'} \left(\frac{2x_{\max}}{\Delta_*}\left\|\boldsymbol{\beta}^* - \hat{\boldsymbol{\beta}}_{t-1}\right\|_1\right)^\alpha$$

$$\leq \sum_{t=M_0+\tau_1+1}^{M_0+\tau_1+t'} \frac{\phi_*^2}{80x_{\max}^2 s_0}$$

$$= \frac{\phi_*^2}{80x_{\max}^2 s_0}t'.$$

To prove that the inequality holds for $t' \geq \tau_2 - \tau_1 + 1$, we use mathematical induction on $t'$. Suppose $\overline{N}(t') \leq \frac{\phi_*^2}{80x_{\max}^2 s_0}t'$ holds for some $t' \geq \tau_2 - \tau_1 + 1$. We must prove that it implies $\overline{N}(t'+1) \leq \frac{\phi_*^2}{80x_{\max}^2 s_0}(t'+1)$. By Lemma 7, we have

$$\left\|\boldsymbol{\beta}^* - \hat{\boldsymbol{\beta}}_{M_0+\tau_1+t'}\right\|_1 \leq \frac{200\sigma x_{\max}s_0}{\phi_*^2} \sqrt{\frac{2\log\log 2(\tau_1+t') + \log\frac{7d}{\delta}}{\tau_1+t'}}.$$

Note that for $n \geq \tau_2$, $\frac{2\log\log 2n + \log\frac{7d}{\delta}}{n} \leq \left(\frac{\Delta_*\phi_*^2}{400\sigma x_{\max}^2 s_0}\right)^2 \left(\frac{\phi_*^2}{80x_{\max}^2 s_0}\right)^{\frac{2}{\alpha}}$ holds, which is shown in Eq. (11). Since $\tau_1 + t' \geq \tau_2$, we have

$$\left\|\boldsymbol{\beta}^* - \hat{\boldsymbol{\beta}}_{M_0+\tau_1+t'}\right\|_1 \leq \frac{\Delta_*}{2x_{\max}}\left(\frac{80x_{\max}^2 s_0}{\phi_*^2}\right)^{\frac{1}{\alpha}}.$$

Therefore, we have

$$\overline{N}(t'+1) = \overline{N}(t') + \left(\frac{2x_{\max}}{\Delta_*}\left\|\boldsymbol{\beta}^* - \hat{\boldsymbol{\beta}}_{M_0+\tau_1+t'}\right\|_1\right)^\alpha$$

$$\leq \frac{\phi_*^2}{80x_{\max}^2 s_0}t' + \frac{\phi_*^2}{80x_{\max}^2 s_0}$$

$$= \frac{\phi_*^2}{80x_{\max}^2 s_0}(t'+1).$$

By mathematical induction, $\overline{N}(t') \leq \frac{\phi_*^2}{80x_{\max}^2 s_0}t'$ holds for all $t' \geq \tau_2 - \tau_1 + 1$. $\qquad\square$

### E.4.5 PROOF OF LEMMA 9

*Proof of Lemma 9.* By Lemma 24, the instantaneous regret in round $t \geq \tau + 1$ is at most $\overline{\Delta}_{t-1}$, i.e., $\text{reg}_t \leq 2x_{\max} \|\boldsymbol{\beta}^* - \hat{\boldsymbol{\beta}}_{t-1}\|_1 \leq \overline{\Delta}_{t-1}$. Define $N_\tau(t) = \sum_{i=\tau+1}^{\tau+t} \mathbb{1}\{a_i \neq a_i^*\}$. The cumulative regret from round $t = \tau + 1$ to $T$ is bounded as the following:

$$
\sum_{t=\tau+1}^{T} \text{reg}_t \leq \sum_{t=\tau+1}^{T} \overline{\Delta}_{t-1} \mathbb{1}\{a_t \neq a_t^*\}
$$

$$
= \sum_{t=\tau+1}^{T} \overline{\Delta}_{t-1} \left( N_\tau(t - \tau) - N_\tau(t - \tau - 1) \right)
$$

$$
= \sum_{t'=1}^{T-\tau} \overline{\Delta}_{\tau+t'-1} \left( N_\tau(t') - N_\tau(t' - 1) \right). \tag{31}
$$

To show that the bound above is increasing in $N_\tau(t')$ for $t' \geq 1$, we rewrite Eq. (31) using the summation by parts technique as follows:

$$
\sum_{t'=1}^{T-\tau} \overline{\Delta}_{\tau+t'-1} \left( N_\tau(t') - N_\tau(t' - 1) \right) = \sum_{t'=1}^{T-\tau} \overline{\Delta}_{\tau+t'-1} N_\tau(t') - \sum_{t'=0}^{T-\tau-1} \overline{\Delta}_{\tau+t'} N_\tau(t')
$$

$$
= \overline{\Delta}_{T-1} N_\tau(T - \tau) + \sum_{t'=1}^{T-\tau-1} \left( \overline{\Delta}_{\tau+t'-1} - \overline{\Delta}_{\tau+t'} \right) N_\tau(t'). \tag{32}
$$

Since $\overline{\Delta}_t$ is non-increasing, we have $\overline{\Delta}_{\tau+t'-1} - \overline{\Delta}_{\tau+t'} \geq 0$. One can observe that the value of Eq. (32) increases when $N_\tau(t')$ is replaced by a larger value for $t' \geq 1$. Under the event $\mathcal{E}_N(\tau)$, it holds that $N_\tau(t') \leq \frac{5}{4} \sum_{i=\tau+1}^{\tau+t'} \min\left\{1, \left(\frac{\overline{\Delta}_{i-1}}{\Delta_*}\right)^\alpha\right\} + 4 \log \frac{1}{\delta}$ for all $t' \geq 1$. Replace $N_\tau(t')$ by $\frac{5}{4} \sum_{i=\tau+1}^{\tau+t'} \min\left\{1, \left(\frac{\overline{\Delta}_{i-1}}{\Delta_*}\right)^\alpha\right\} + 4 \log \frac{1}{\delta}$ for $t' \geq 1$ in Eq. (31) and obtain the desired upper bound.

$$
\sum_{t'=1}^{T-\tau} \overline{\Delta}_{\tau+t'-1} \left( N_\tau(t') - N_\tau(t' - 1) \right)
$$

$$
\leq \overline{\Delta}_\tau \left( \frac{5}{4} \min\left\{1, \left(\frac{\overline{\Delta}_\tau}{\Delta_*}\right)^\alpha\right\} + 4 \log \frac{1}{\delta} \right) + \sum_{t=\tau+2}^{T} \overline{\Delta}_{t-1} \cdot \frac{5}{4} \min\left\{1, \left(\frac{\Delta_{t-1}}{\Delta_*}\right)^\alpha\right\}
$$

$$
= 4\overline{\Delta}_\tau \log \frac{1}{\delta} + \frac{5}{4} \sum_{t=\tau}^{T-1} \overline{\Delta}_t \min\left\{1, \left(\frac{\Delta_{t-1}}{\Delta_*}\right)^\alpha\right\}.
$$

$\square$

### E.4.6 PROOF OF LEMMA 10

*Proof of Lemma 10.* Let $\mathcal{F}_t^+$ be the $\sigma$-algebra generated by $\left(\{\mathbf{x}_{\tau,i}\}_{\tau \in [t], i \in [K]}, \{a_\tau\}_{\tau \in [t]}, \{r_{\tau,a_\tau}\}_{\tau \in [t]}\right)$. Then, $\mathbf{x}_{t,a_t}$ and $\hat{\boldsymbol{\beta}}_t$ are $\mathcal{F}_t^+$-measurable. Under the greedy diversity, we have that for all $t \geq 1$,

$$
\phi^2\left(\mathbb{E}\left[\mathbf{x}_{t,a_t} \mathbf{x}_{t,a_t}^\top \mid \mathcal{F}_{t-1}^+\right], S_0\right) = \phi^2\left(\mathbb{E}\left[\mathbf{x}_{\hat{\boldsymbol{\beta}}_{t-1}} \mathbf{x}_{\hat{\boldsymbol{\beta}}_{t-1}}^\top \mid \mathcal{F}_{t-1}^+\right], S_0\right)
$$

$$
\geq \phi_G^2.
$$

By Lemma 23, with probability at least $1 - \delta$, $\phi^2\left(\hat{\mathbf{V}}_t, S_0\right) \geq \frac{\phi_G^2 t}{2}$ holds for all $t \geq \frac{2048 x_{\max}^4 s_0^2}{\phi_G^4} \left(\log \frac{d^2}{\delta} + 2 \log \frac{64 x_{\max}^2 s_0}{\phi_G^2}\right) + 1 = \tau_0 + 1$. $\square$

### E.4.7 PROOF OF LEMMA 11

*Proof of Lemma 11.* By Lemma 19, under the events $\mathcal{E}_g$ and $\mathcal{E}_{\mathrm{GD}}$, the estimation error of $\hat{\boldsymbol{\beta}}_t$ for $t \geq \tau_0 + 1$ is bounded as follows:

$$
\begin{aligned}
\left\| \boldsymbol{\beta}^* - \hat{\boldsymbol{\beta}}_t \right\|_1 &\leq \frac{2 s_0 \lambda_t}{\frac{\phi_{\mathrm{G}}^2 t}{2}} \\
&= \frac{2^{\frac{19}{4}} \sigma x_{\max} s_0}{\phi_{\mathrm{G}}^2} \sqrt{\frac{2 \log \log 2t + \log \frac{7d}{\delta}}{t}} \\
&\leq \frac{27 \sigma x_{\max} s_0}{\phi_{\mathrm{G}}^2} \sqrt{\frac{2 \log \log 2t + \log \frac{7d}{\delta}}{t}} \ .
\end{aligned}
\tag{33}
$$

Define $\overline{\Delta}_t$ as follows:

$$
\overline{\Delta}_t = \frac{54 \sigma x_{\max}^2 s_0}{\phi_{\mathrm{G}}^2} \sqrt{\frac{2 \log \log 2t + \log \frac{7d}{\delta}}{t}} \ .
$$

Then, $2 x_{\max} \left\| \boldsymbol{\beta}^* - \hat{\boldsymbol{\beta}}_t \right\|_1 \leq \overline{\Delta}_t$ for all $t \geq \tau_0 + 1$, and $\overline{\Delta}_t$ is decreasing in $t$. Therefore, we can use Lemma 9 with $\tau = \tau_0$, which gives the following upper bound of cumulative regret:

$$
\sum_{t=\tau_0+1}^{T} \mathrm{reg}_t \leq 4 \overline{\Delta}_{\tau_0} \log \frac{1}{\delta} + \frac{5}{4} \sum_{t=\tau_0}^{T-1} \overline{\Delta}_t \min \left\{ 1, \left( \frac{\overline{\Delta}_t}{\Delta_*} \right)^{\alpha} \right\} \ .
$$

We first address the case where $\alpha \leq 1$. Plugging in the definition of $\overline{\Delta}_t$, We have

$$
\begin{aligned}
\sum_{t=\tau_0+1}^{T} \mathrm{reg}_t &\leq 4 \overline{\Delta}_{\tau_0} \log \frac{1}{\delta} + \frac{5}{4} \sum_{t=\tau_0}^{T-1} \frac{\overline{\Delta}_t^{1+\alpha}}{\Delta_*^{\alpha}} \\
&= 4 \overline{\Delta}_{\tau_0} \log \frac{1}{\delta} + \frac{5}{4 \Delta_*^{\alpha}} \left( \frac{54 \sigma x_{\max}^2 s_0}{\phi_{\mathrm{G}}^2} \right)^{1+\alpha} \sum_{t=\tau_0}^{T-1} \left( \frac{2 \log \log 2t + \log \frac{7d}{\delta}}{t} \right)^{\frac{1+\alpha}{2}} \ .
\end{aligned}
\tag{34}
$$

By Lemma 26, we bound the sum as the following:

$$
\sum_{t=\tau_0}^{T-1} \left( \frac{2 \log \log 2t + \log \frac{7d}{\delta}}{t} \right)^{\frac{1+\alpha}{2}} \leq \begin{cases} \frac{2}{1-\alpha} T^{\frac{1-\alpha}{2}} \left( 2 \log \log 2T + \log \frac{7d}{\delta} \right) & \alpha \in [0, 1) \\ (\log T) \left( 2 \log \log 2T + \log \frac{7d}{\delta} \right) & \alpha = 1 \, . \end{cases}
\tag{35}
$$

By combining inequalities (34) and (35), we conclude that

$$
\sum_{t=\tau_0+1}^{T} \mathrm{reg}_t \leq 4 \overline{\Delta}_{\tau_0} \log \frac{1}{\delta} + I_2(T) \, ,
$$

where

$$
I_2(T) = \begin{cases} \mathcal{O} \left( \frac{1}{(1-\alpha) \Delta_*^{\alpha}} \left( \frac{\sigma x_{\max}^2 s_0}{\phi_{\mathrm{G}}^2} \right)^{1+\alpha} T^{\frac{1-\alpha}{2}} \left( \log d + \log \frac{\log T}{\delta} \right) \right) & \alpha \in [0, 1) \, , \\ \mathcal{O} \left( \left( \frac{\sigma x_{\max}^2 s_0}{\phi_{\mathrm{G}}^2} \right)^2 (\log T) \left( \log d + \log \frac{\log T}{\delta} \right) \right) & \alpha = 1 \, . \end{cases}
$$

Now, suppose $\alpha > 1$. We need more sophisticated analysis to bound the regret in this case. Let $\tau_0'$ be a constant that satisfies the following:

$$
\forall n \geq \tau_0', \quad \frac{2 \log \log 2\tau_0' + \log \frac{7d}{\delta}}{\tau_0'} \leq \left( \frac{54 \sigma x_{\max}^2 s_0}{\Delta_* \phi_{\mathrm{G}}^2} \right)^{-2} \ .
\tag{36}
$$

By Lemma 25, it is sufficient to take $\tau_0' = C_0' \log \frac{7d}{\delta} + 2 C_0' \log \log \frac{28 d C_0'^2}{\delta}$, where $C_0' = \max \left\{ 2, \left( \frac{54 \sigma x_{\max}^2 s_0}{\Delta_* \phi_{\mathrm{G}}^2} \right)^2 \right\}$. Now, we bound the cumulative regret as the following:

$$
\sum_{t=\tau_0+1}^{T} \mathrm{reg}_t \leq 4 \overline{\Delta}_{\tau_0} \log \frac{1}{\delta} + \frac{5}{4} \sum_{t=\tau_0}^{\tau_0'} \overline{\Delta}_t + \frac{5}{4} \sum_{t=\tau_0'+1}^{T-1} \frac{\overline{\Delta}_t^{1+\alpha}}{\Delta_*^{\alpha}} \, ,
\tag{37}
$$

where the sum $\sum_{t=\tau_0}^{\tau_0'} \overline{\Delta}_t$ is treated as 0 when $\tau_0 > \tau_0'$. Plug the definition of $\overline{\Delta}_t$ into the first summation and obtain

$$\sum_{t=\tau_0}^{\tau_0'} \overline{\Delta}_t = \frac{54\sigma x_{\max}^2 s_0}{\phi_G^2} \sum_{t=\tau_0}^{\tau_0'} \sqrt{\frac{2\log\log 2t + \log \frac{7d}{\delta}}{t}} \, .$$

By Lemma 26 with $r = \frac{1}{2}$, we have

$$\sum_{t=\tau_0}^{\tau_0'} \sqrt{\frac{2\log\log 2t + \log \frac{7d}{\delta}}{t}} \leq 2\sqrt{\tau_0'\left(2\log\log 2\tau_0' + \log\frac{7d}{\delta}\right)}$$

$$= 2\tau_0'\sqrt{\frac{2\log\log 2\tau_0' + \log\frac{7d}{\delta}}{\tau_0'}} \, .$$

By constraint (36) of $\tau_0'$, we achieve

$$\frac{5}{4}\sum_{t=\tau_0}^{\tau_0'} \overline{\Delta}_t \leq \frac{5}{4}\left(\frac{54\sigma x_{\max}^2 s_0}{\phi_G^2}\right) \cdot 2\tau_0'\sqrt{\frac{2\log\log 2\tau_0' + \log\frac{7d}{\delta}}{\tau_0'}}$$

$$\leq \frac{5\tau_0'}{2}\left(\frac{54\sigma x_{\max}^2 s_0}{\phi_G^2}\right)\left(\frac{54\sigma x_{\max}^2 s_0}{\Delta_* \phi_G^2}\right)^{-1}$$

$$\leq \frac{5\Delta_* \tau_0'}{2}$$

$$= \mathcal{O}\left(\frac{1}{\Delta_*}\left(\frac{\sigma x_{\max}^2 s_0}{\phi_G^2}\right)^2\left(\log d + \log\frac{1}{\delta}\right)\right) \, . \tag{38}$$

For the last summation in inequality (37), we have

$$\sum_{t=\tau_0'+1}^{T-1} \overline{\Delta}_t^{1+\alpha} = \left(\frac{54\sigma x_{\max}^2 s_0}{\phi_G^2}\right)^{1+\alpha} \sum_{t=\tau_0'+1}^{T-1}\left(\frac{2\log\log 2t + \log\frac{7d}{\delta}}{t}\right)^{\frac{1+\alpha}{2}}$$

$$\leq \left(\frac{54\sigma x_{\max}^2 s_0}{\phi_G^2}\right)^{1+\alpha} \cdot \frac{4\alpha}{(\alpha-1)^2} \cdot \frac{\left(2\log\log 2\tau_0' + \log\frac{7d}{\delta}\right)^{\frac{\alpha+1}{2}}}{\tau_0'^{\frac{\alpha-1}{2}}} \, ,$$

where the equality holds by the definition of $\overline{\Delta}_t$, and the inequality comes from Lemma 26. Again by constraint (36), we have

$$\frac{\left(2\log\log 2\tau_0' + \frac{7d}{\delta}\right)^{\frac{\alpha+1}{2}}}{\tau_0'^{\frac{\alpha-1}{2}}} \leq \left(\frac{54\sigma x_{\max}^2 s_0}{\Delta_* \phi_G^2}\right)^{1-\alpha}\left(2\log\log 2\tau_0' + \log\frac{7d}{\delta}\right) \, .$$

Then, we have

$$\frac{5}{4}\sum_{t=\tau_0'+1}^{T-1} \frac{\overline{\Delta}_t^{1+\alpha}}{\Delta_*^\alpha} \leq \frac{5\alpha}{(\alpha-1)^2}\left(\frac{54\sigma x_{\max}^2 s_0}{\phi_G^2}\right)^2\left(2\log\log 2\tau_0' + \log\frac{7d}{\delta}\right)$$

$$= \mathcal{O}\left(\frac{\alpha}{(\alpha-1)^2\Delta_*}\left(\frac{\sigma x_{\max}^2 s_0}{\phi_G^2}\right)^2\left(\log d + \log\frac{1}{\delta}\right)\right) \, . \tag{39}$$

Plugging in inequalities of Eq. (38) and Eq. (39) into Eq. (37) yields

$$\sum_{t=\tau_0+1}^{T} \text{reg}_t \leq 4\overline{\Delta}_{\tau_0}\log\frac{1}{\delta} + I_2(T) \, ,$$

where

$$I_2(T) = \mathcal{O}\left(\frac{\alpha^2}{(\alpha-1)^2\Delta_*}\left(\frac{\sigma x_{\max}^2 s_0}{\phi_G^2}\right)^2\left(\log d + \log\frac{1}{\delta}\right)\right)$$

in case $\alpha > 1$.

Putting all together, for any $\alpha \geq 0$, we obtain

$$\sum_{t=\tau_0+1}^{T} \text{reg}_t \leq 4\Delta_{\tau_0} \log \frac{1}{\delta} + I_2(T), \tag{40}$$

where

$$I_2(T) = \begin{cases} \mathcal{O}\left( \frac{1}{(1-\alpha)\Delta_*^{\alpha}} \left( \frac{\sigma x_{\max}^2 s_0}{\phi_G^2} \right)^{1+\alpha} T^{\frac{1-\alpha}{2}} \left( \log d + \log \frac{\log T}{\delta} \right) \right) & \alpha \in [0,1], \\[2ex] \mathcal{O}\left( \left( \frac{\sigma x_{\max}^2 s_0}{\phi_G^2} \right)^2 (\log T) \left( \log d + \log \frac{\log T}{\delta} \right) \right) & \alpha = 1, \\[2ex] \mathcal{O}\left( \frac{\alpha^2}{(\alpha-1)^2 \Delta_*} \left( \frac{\sigma x_{\max}^2 s_0}{\phi_G^2} \right)^2 \left( \log d + \log \frac{1}{\delta} \right) \right) & \alpha > 1. \end{cases}$$

We bound the cumulative regret of first $\tau_0$ rounds by $2x_{\max} b \tau_0$, which is the maximum regret possible. We also bound $\overline{\Delta}_{\tau_0} \leq 2x_{\max} b$, since $\overline{\Delta}_{\tau_0}$ represents the maximum instantaneous regret in round $t = \tau_0 + 1$. Together with Eq. (40), we obtain

$$\sum_{t=1}^{T} \text{reg}_t \leq 2 \max b \left( \tau_0 + 4 \log \frac{1}{\delta} \right) + I_2(T).$$

$\square$

## F   Forced Sampling with Lasso (FS-Lasso)

In this section, we present `FS-Lasso`, an algorithm that uses forced-sampling adaptively. We prove that `FS-Lasso` is capable of bounding the expected regret even when $T$ is unknown. The regret bound matches the regret bound of `FS-WLasso`.

Forced-sampling algorithms in the existing literature (Goldenshluger & Zeevi, 2013; Bastani & Bayati, 2020) are designed for the multiple parameter setting where each arm has its own hidden parameter and one context feature vector is given at each round. Additionally, the compatibility assumptions employed by Bastani & Bayati (2020) (Assumption 4 in (Bastani & Bayati, 2020)) involve the compatibility condition of the expected Gram matrix of the optimal context vectors when the gap is large enough (measured by $h$ in (Bastani & Bayati, 2020)). This assumption enables a more straightforward regret analysis because it implies that a small estimation error is guaranteed if the agent chooses the optimal arm only when it is clearly distinguishable from the others. However, our assumption (Assumption 3) does not imply such a convenient guarantee. Furthermore, Bastani & Bayati (2020) make an additional assumption (Assumption 3 in (Bastani & Bayati, 2020)), stating that some subset of arms is always sub-optimal with a gap of at least $h$ (denoted by $\mathcal{K}_{\text{sub}}$ in (Bastani & Bayati, 2020)), and the probability of observing an optimal context corresponding to the rest of the arms with a sub-optimality gap $h$ is lower-bounded by $p_*$.

We consider the single parameter setting where there is one unknown reward parameter vector and multiple feature vectors for each arm are given at each round. We emphasize that directly translating assumptions or theoretical guarantees across these different settings is either not trivial or not optimal, or usually both. Under Assumptions 1-3, we show that `FS-Lasso` achieves the same regret bound as `FS-WLasso` without constraining the expected Gram matrix of the optimal arms only to cases where the sub-optimality gap is large, or a lower bound on the probability of observing such a large sub-optimality gap.

### F.1   Algorithm: FS-Lasso

For a non-empty set of index $\mathcal{I}$, let us define $L_{\mathcal{I}}(\boldsymbol{\beta})$ as follows:

$$L_{\mathcal{I}}(\boldsymbol{\beta}) := \frac{1}{|\mathcal{I}|} \sum_{i \in \mathcal{I}} \left( \mathbf{x}_{i,a_i}^{\top} \boldsymbol{\beta} - r_{i,a_i} \right)^2$$

---

**Algorithm 2** FS-Lasso (*Forced Sampling with Lasso*)

---

1: **Input:** Forced sampling function $q : \mathbb{N}_0 \to \mathbb{R}_{\geq 0}$, localization parameter $h > 0$, regularization parameters $\lambda_1, \{\lambda_{2,t}\}_{t \geq 1}$
2: **Initialize:** $\mathcal{T}_e(1) = \mathcal{T}_g(1) = \emptyset$, $\widetilde{\boldsymbol{\beta}}_0 = \hat{\boldsymbol{\beta}}_0 = \mathbf{0}_d$
3: **for** $t = 1, 2, ..., T$ **do**
4:      Observe $\{\mathbf{x}_{t,k}\}_{k=1}^K$
5:      **if** $|\mathcal{T}_e(t)| \leq q(|\mathcal{T}_g(t)|)$ **then**
6:          Choose $a_t \sim \text{Unif}(\mathcal{A})$ and observe $r_{t,a_t}$
7:          $\mathcal{T}_e(t + 1) = \mathcal{T}_e(t) \cup \{t\}$
8:          $\widetilde{\boldsymbol{\beta}}_{|\mathcal{T}_e(t+1)|} = \text{argmin}_{\boldsymbol{\beta}} L_{\mathcal{T}_e(t+1)}(\boldsymbol{\beta}) + \lambda_1 \|\boldsymbol{\beta}\|_1$
9:      **else**
10:          $\widetilde{a}_t = \text{argmax}_{k \in [K]} \mathbf{x}_{t,k}^\top \widetilde{\boldsymbol{\beta}}_{|\mathcal{T}_e(t)|}$
11:          **if** $\mathbf{x}_{t,\widetilde{a}_t}^\top \widetilde{\boldsymbol{\beta}}_{|\mathcal{T}_e(t)|} > \max_{k \neq \widetilde{a}_t} \mathbf{x}_{t,k}^\top \widetilde{\boldsymbol{\beta}}_{|\mathcal{T}_e(t)|} + h$ **then**
12:              Choose $a_t = \widetilde{a}_t$
13:          **else**
14:              Choose $a_t = \text{argmax}_{k \in [K]} \mathbf{x}_{t,k}^\top \hat{\boldsymbol{\beta}}_{|\mathcal{T}_g(t)|}$
15:          **end if**
16:          Observe $r_{t,a_t}$
17:          $\mathcal{T}_g(t + 1) = \mathcal{T}_g(t) \cup \{t\}$
18:          Update $\hat{\boldsymbol{\beta}}_{|\mathcal{T}_g(t+1)|} = \text{argmin}_{\boldsymbol{\beta}} L_{\mathcal{T}_g(t+1)} + \lambda_{2,t} \|\boldsymbol{\beta}\|_1$
19:      **end if**
20: **end for**

---

## F.2 REGRET BOUND OF FS-Lasso

**Theorem 4.** *Suppose Assumptions 1-3 hold. If the agent runs Algorithm 2 with the input parameters as*

$$q(n) = \frac{512 \rho^2 x_{\max}^4 s_0^2 \log 2d^2(n + 1)^3}{\phi_*^4} \max \left\{ 4, \frac{4\sigma^2}{\Delta_*^2} \left( \frac{128 x_{\max}^2 s_0}{\phi_*^2} \right)^{\frac{2}{\alpha}} \right\}, h = \frac{\Delta_*}{2} \left( \frac{\phi_*^2}{128 x_{\max}^2 s_0} \right)^{\frac{1}{\alpha}},$$

$$\lambda_1 = \frac{\phi_*^2 h}{2 \rho x_{\max} s_0}, \quad \lambda_{2,t} = 4 \sigma x_{\max} \sqrt{\frac{2 \log 4d(|\mathcal{T}_g(t)| + 1)^2}{t}},$$

*then, the expected cumulative regret is bounded as follows:*

$$\mathbb{E}\left[ \sum_{t=1}^T reg_t \right] \leq 2 x_{\max} b I_0 + I_T,$$

*where*

$$I_0 = \mathcal{O}\left( q(T) + \frac{x_{\max}^4 s_0^2}{\phi_*^4} \log d \right),$$

$$I_T \leq \begin{cases} \mathcal{O}\left( \frac{1}{(1-\alpha)\Delta_*^\alpha} \left( \frac{\sigma x_{\max}^2 s_0}{\phi_*^2} \right)^{1+\alpha} T^{\frac{1-\alpha}{2}} (\log d + \log T)^{\frac{1+\alpha}{2}} \right) & \alpha \in (0, 1), \\ \mathcal{O}\left( \frac{1}{\Delta_*} \left( \frac{\sigma x_{\max}^2 s_0}{\phi_*^2} \right) (\log T)(\log d + \log T) \right) & \alpha = 1, \\ \mathcal{O}\left( \frac{1}{(\alpha-1)\Delta_*} \left( \frac{\sigma x_{\max}^2 s_0}{\phi_*^2} \right)^2 (\log d + \log T) \right) & \alpha > 1. \end{cases}$$

## F.3 PROOF OF THEOREM 4

*Proof of Theorem 4.* We define $\mathcal{T}_g$ to be the set of rounds that take greedy actions, and $\mathcal{T}_e$ to be the set of rounds that take random actions. We define $n_g(t) = |\mathcal{T}_g \cap [t]|$ to be the number of greedy selections up to round $t$, and $n_e(t) = |\mathcal{T}_e \cap [t]|$ to be the number of random selections up to round $t$. We first bound the estimation error of $\widetilde{\boldsymbol{\beta}}$, the estimator obtained by forced-sampled arms.

**Lemma 16.** *Suppose $q(n)$ and $\lambda_1$ of Algorithm 2 satisfy $q(n) \geq \frac{\rho^2 x_{\max}^4 s_0^2}{\phi_*^4} \max \left\{ 2048 \log 2d^2(n+1)^3, \frac{512\sigma^2}{h^2} \log 2d(n+1)^3 \right\}$ and $\lambda_1 = \frac{\phi_*^2 h}{4\rho x_{\max} s_0}$. Define an event $\Gamma_e(t) = \left\{ \omega \in \Omega : \left\| \boldsymbol{\beta}^* - \widetilde{\boldsymbol{\beta}}_{|\mathcal{T}_e(t)|} \right\|_1 \leq \frac{h}{2x_{\max}} \right\}$. Then, for all $t \in \mathcal{T}_g$, $\mathbb{P}\left(\Gamma_e(t)^c\right) \leq \frac{2}{n_g(t)^3}$.*

We further define a set $\mathcal{T}_g^-(t) = \left\{ i \in \mathcal{T}(t+1) \mid n_g(i) \geq \left\lfloor \frac{n_g(t)+1}{2} \right\rfloor + 1 \right\}$. $\mathcal{T}_g^-(t)$ is the set of rounds that the latter half of the greedy actions are made, rounded up. Note that $\left|\mathcal{T}_g^-(t)\right| = \left\lceil \frac{n_g(t)}{2} \right\rceil$. We show that the number of sub-optimal arm selections during the latter half of the greedy actions is bounded with high probability.

**Lemma 17.** *Let $N^-(t) = \sum_{i \in \mathcal{T}_g^-(t)} \mathbb{1}\left\{a_i \neq a_i^*\right\}$. $N^-(t)$ is the number of sub-optimal arm selections during the latter half of the greedy actions. Let $\Gamma_{N^-}(t) = \left\{ \omega \in \Omega : N^-(t) \leq \frac{\phi_*^2}{64 x_{\max}^2 s_0} \left\lceil \frac{n_g(t)}{2} \right\rceil \right\}$. If the input parameters of Algorithm 2 satisfy $h \leq \frac{\Delta_*}{2} \left( \frac{\phi_*}{128 x_{\max}^2 s_0} \right)^{\frac{1}{\alpha}}$, $q(n) \geq \frac{\rho^2 x_{\max}^4 s_0^2}{\phi_*^4} \max \left\{ 2048 \log 2d^2(n+1)^3, \frac{512\sigma^2}{h^2} \log 2d(n+1)^3 \right\} \log 2d^2(n+1)^3$, and $\lambda_1 = \frac{\phi_*^2 h}{4\rho x_{\max} s_0}$, then $\mathbb{P}\left(\Gamma_{N^-}(t)^c\right) \leq \frac{19}{n_g(t)^2} + \exp\left(-\frac{n_g(t)\phi_*^4}{16384 x_{\max}^4 s_0^2}\right)$.*

Finally, we bound the estimation error of $\hat{\boldsymbol{\beta}}$ when the majority of the samples are obtained from greedy actions.

**Lemma 18.** *Suppose $t \in \mathcal{T}_g$, $\lambda_{2,t} = 4\sigma x_{\max} \sqrt{\frac{2\log 4dn_g(t)^2}{t}}$, and $n_g(t) \geq n_e(t)$. Define an event $\Gamma_g(t) = \left\{ \omega \in \Omega : \left\| \boldsymbol{\beta}^* - \hat{\boldsymbol{\beta}}_{|\mathcal{T}_g(t)|} \right\|_1 < \frac{128\sigma x_{\max} s_0}{\phi_*^2} \sqrt{\frac{2\log 4dn_g(t)^2}{t}} \right\}$. Then, $\mathbb{P}\left(\Gamma_g(t)^c\right) \leq \frac{20}{n_g(t)^2} + \exp\left(-\frac{\phi_*^4 n_g(t)}{16384 x_{\max}^4 s_0^2}\right) + 2d^2 \exp\left(-\frac{\phi_*^4 n_g(t)}{4096 x_{\max}^4 s_0^2}\right)$.*

Now, we bound the total regret of Algorithm 2. We observe that there are at most $n_e(T)$ random actions. We set $T_0 = \max\left\{ n_e(T), \frac{8192 x_{\max}^4 s_0^2}{\phi_*^4} \log d \right\}$. For all random actions and the first $T_0$ greedy actions, we bound the incurred regret by $2x_{\max} b \cdot 2T_0$, which is the maximum regret possible. Now, we bound the regret incurred by the greedy selections from $n_g(t) = T_0 + 1$. We decompose the expected instantaneous regret in round $t$ as follows:

$$\mathbb{E}\left[\mathrm{reg}_t\right] \leq \mathbb{E}\left[\mathrm{reg}_t \mathbb{1}\left\{\Gamma_e(t)^c\right\}\right] + \mathbb{E}\left[\mathrm{reg}_t \mathbb{1}\left\{\Gamma_g(t)^c\right\}\right] + \mathbb{E}\left[\mathrm{reg}_t \mathbb{1}\left\{\mathrm{reg}_t > 0, \Gamma_e(t), \Gamma_g(t)\right\}\right].$$

The first two terms are the regret when good events do not hold. We take $2x_{\max} b$ as the upper bound of the instantaneous regret in this case, and bound the terms using Lemmas 16 and 18.

$$
\begin{aligned}
&\mathbb{E}\left[\mathrm{reg}_t \mathbb{1}\left\{\Gamma_e(t)^c\right\}\right] + \mathbb{E}\left[\mathrm{reg}_t \mathbb{1}\left\{\Gamma_g(t)^c\right\}\right] \\
&\leq 2x_{\max} b \left(\mathbb{P}\left(\Gamma_e(t)^c\right) + \mathbb{P}\left(\Gamma_g(t)^c\right)\right) \\
&\leq 2x_{\max} b \left(\frac{2}{n_g(t)^3} + \frac{20}{n_g(t)^2} + \exp\left(-\frac{\phi_*^4 n_g(t)}{16384 x_{\max}^4 s_0^2}\right) + 2d^2 \exp\left(-\frac{\phi_*^4 n_g(t)}{4096 x_{\max}^4 s_0^2}\right)\right) \\
&\leq 2x_{\max} b \left(\frac{22}{n_g(t)^2} + \exp\left(-\frac{\phi_*^4 n_g(t)}{16384 x_{\max}^4 s_0^2}\right) + 2d^2 \exp\left(-\frac{\phi_*^4 n_g(t)}{4096 x_{\max}^4 s_0^2}\right)\right).
\end{aligned}
$$

The sum of the expected regret when the good events do not hold is bounded as the following:

$$\sum_{n_g(t)=T_0+1}^{n_g(T)} \mathbb{E}\left[\mathrm{reg}_t \mathbb{1}\left\{\Gamma_e(t)^{\mathsf{c}}\right\}\right] + \mathbb{E}\left[\mathrm{reg}_t \mathbb{1}\left\{\Gamma_g(t)^{\mathsf{c}}\right\}\right]$$

$$\leq \sum_{n_g(t)=T_0+1}^{n_g(T)} 2x_{\max}b\left(\frac{22}{n_g(t)^2} + \exp\left(-\frac{\phi_*^4 n_g(t)}{16384 x_{\max}^4 s_0^2}\right) + 2d^2 \exp\left(-\frac{\phi_*^4 n_g(t)}{4096 x_{\max}^4 s_0^2}\right)\right)$$

$$\leq 88x_{\max}b + 2x_{\max}b \int_{T_0}^{\infty} \exp\left(-\frac{\phi_*^4 x}{16384 x_{\max}^4 s_0^2}\right) + 2d^2 \exp\left(-\frac{\phi_*^4 x}{4096 x_{\max}^4 s_0^2}\right) dx$$

$$\leq 88x_{\max}b + 2x_{\max}b\left(\frac{16384 x_{\max}^4 s_0^2}{\phi_*^4} \exp\left(-\frac{\phi_*^4 T_0}{16384 x_{\max}^4 s_0^2}\right) + \frac{8192 d^2 x_{\max}^4 s_0^2}{\phi_*^4} \exp\left(-\frac{\phi_*^4 T_0}{4096 x_{\max}^4 s_0^2}\right)\right).$$

By the fact that $T_0 \geq \frac{8192 x_{\max}^4 s_0^2}{\phi_*^4} \log d$, the exponential in the last term is bounded by $\exp\left(-\frac{\phi_*^4 T_0}{4096 x_{\max}^4 s_0^2}\right) \leq \frac{1}{d^2}$. We obtain the bound of cumulative regret without the good events, which is a constant independent of $T$.

$$\sum_{n_g(t)=T_0+1}^{n_g(T)} \mathbb{E}\left[\mathrm{reg}_t \mathbb{1}\left\{\Gamma_e(t)^{\mathsf{c}}\right\}\right] + \mathbb{E}\left[\mathrm{reg}_t \mathbb{1}\left\{\Gamma_g(t)^{\mathsf{c}}\right\}\right] \leq 88x_{\max}b + \frac{49152 x_{\max}^5 b s_0^2}{\phi_*^4}.$$

Now, we are left to bound the cumulative regret when the good events $\Gamma_g(t), \Gamma_e(t)$ hold. We first show that if the agent chooses $a_t = \widetilde{a}_t$ by the if clause in line 11, since $\mathbf{x}_{t,\widetilde{a}_t}^{\top}\widetilde{\boldsymbol{\beta}}_{|\mathcal{T}_e(t)|} > \max_{k \neq \widetilde{a}_t} \mathbf{x}_{t,k}^{\top}\widetilde{\boldsymbol{\beta}}_{|\mathcal{T}_e(t)|} + h$ is satisfied, then under $\Gamma_e(t)$, $a_t = a_t^*$ holds. Suppose not, then we have $\mathbf{x}_{t,\widetilde{a}_t}^{\top}\widetilde{\boldsymbol{\beta}}_{n_e(t)} > \mathbf{x}_{t,a_t^*}^{\top}\widetilde{\boldsymbol{\beta}}_{n_e(t)} + h$. On the other hand, we have $\mathbf{x}_{t,a_t^*}^{\top}\boldsymbol{\beta}^* - \mathbf{x}_{t,\widetilde{a}_t}^{\top}\boldsymbol{\beta}^* \geq 0$. Combining these two inequalities, we obtain

$$h < \left(\mathbf{x}_{t,\widetilde{a}_t}^{\top}\widetilde{\boldsymbol{\beta}}_{n_e(t)} - \mathbf{x}_{t,a_t^*}^{\top}\widetilde{\boldsymbol{\beta}}_{n_e(t)}\right) + \left(\mathbf{x}_{t,a_t^*}^{\top}\boldsymbol{\beta}^* - \mathbf{x}_{t,\widetilde{a}_t}^{\top}\boldsymbol{\beta}^*\right)$$

$$= \mathbf{x}_{t,\widetilde{a}_t}^{\top}\left(\widetilde{\boldsymbol{\beta}}_{n_e(t)} - \boldsymbol{\beta}^*\right) + \mathbf{x}_{t,a_t^*}^{\top}\left(\boldsymbol{\beta}^* - \widetilde{\boldsymbol{\beta}}_{n_e(t)}\right)$$

$$\leq 2x_{\max}\left\|\boldsymbol{\beta}^* - \widetilde{\boldsymbol{\beta}}_{n_e(t)}\right\|_1,$$

where we apply the Cauchy-Schwarz inequality for the last inequality. However, under $\Gamma_e(t)$, it holds that $\left\|\boldsymbol{\beta}^* - \widetilde{\boldsymbol{\beta}}_{n_e(t)}\right\|_1 \leq \frac{h}{2x_{\max}}$, which is a contradiction since $h < h$.

Therefore, under the event $\Gamma_e(t)$, $a_t \neq A_t^*$ occurs only when the agent performs a greedy action according to $\hat{\boldsymbol{\beta}}_{|\mathcal{T}_g(t)|}$ by the else clause in line 13. By Lemma 24, the instantaneous regret is at most $2x_{\max}\left\|\boldsymbol{\beta}^* - \hat{\boldsymbol{\beta}}_{|\mathcal{T}_g(t)|}\right\|_1 \leq \frac{256\sigma x_{\max}^2 s_0}{\phi_*^2}\sqrt{\frac{2\log 4dn_g(t)^2}{t}}$. Lemma 24 further tells us that the regret is greater than 0 only when $\Delta_t \leq \frac{256\sigma x_{\max}^2 s_0}{\phi_*^2}\sqrt{\frac{2\log 4dn_g(t)^2}{t}}$. Therefore, we deduce that

$$\mathbb{E}\left[\mathrm{reg}_t \mathbb{1}\left\{\mathrm{reg}_t > 0, \Gamma_e(t), \Gamma_g(t)\right\}\right]$$

$$\leq \mathbb{E}\left[\frac{256\sigma x_{\max}^2 s_0}{\phi_*^2}\sqrt{\frac{2\log 4dn_g(t)^2}{t}} \cdot \mathbb{1}\left\{\Delta_t \leq \frac{256\sigma x_{\max}^2 s_0}{\phi_*^2}\sqrt{\frac{2\log 4dn_g(t)^2}{t}}\right\}\right]$$

$$\leq \left(\frac{256\sigma x_{\max}^2 s_0}{\phi_*^2}\sqrt{\frac{2\log 4dn_g(t)^2}{t}}\right) \mathbb{P}\left(\Delta_t \leq \frac{256\sigma x_{\max}^2 s_0}{\phi_*^2}\sqrt{\frac{2\log 4dn_g(t)^2}{t}}\right)$$

$$\leq \left(\frac{256\sigma x_{\max}^2 s_0}{\phi_*^2}\sqrt{\frac{2\log 4dn_g(t)^2}{t}}\right) \min\left\{1, \left(\frac{256\sigma x_{\max}^2 s_0}{\Delta_*\phi_*^2}\sqrt{\frac{2\log 4dn_g(t)^2}{t}}\right)^{\alpha}\right\}$$

$$\leq \left(\frac{256\sigma x_{\max}^2 s_0}{\phi_*^2}\sqrt{\frac{2\log 4dT^2}{n_g(t)}}\right) \min\left\{1, \left(\frac{256\sigma x_{\max}^2 s_0}{\Delta_*\phi_*^2}\sqrt{\frac{2\log 4dT^2}{n_g(t)}}\right)^{\alpha}\right\},$$

where the third inequality holds by the margin condition, and the last inequality by $n_g(t) \le t \le T$. We separately deal with the cases $\alpha \le 1$ and $\alpha > 1$. The expected cumulative regret under the good events when $\alpha \le 1$ is bounded as the following:

$$\sum_{n_g(t)=T_0+1}^{n_g(T)} \mathbb{E}\left[\text{reg}_t \mathbb{1}\left\{\text{reg}_t > 0, \Gamma_e(t), \Gamma_g(t)\right\}\right]$$

$$\le \sum_{n_g(t)=T_0+1}^{n_g(T)} \left(\frac{256\sigma x_{\max}^2 s_0}{\phi_*^2}\sqrt{\frac{2\log 4dT^2}{t}}\right)\min\left\{1, \left(\frac{256\sigma x_{\max}^2 s_0}{\Delta_* \phi_*^2}\sqrt{\frac{2\log 4dT^2}{n_g(t)}}\right)^\alpha\right\}$$

$$\le \sum_{n_g(t)=T_0+1}^{n_g(T)} \frac{1}{\Delta_*^\alpha}\left(\frac{256\sigma x_{\max}^2 s_0}{\phi_*^2}\sqrt{\frac{2\log 4dT^2}{n_g(t)}}\right)^{1+\alpha}$$

$$\le \frac{1}{\Delta_*^\alpha}\left(\frac{256\sigma x_{\max}^2 s_0 \sqrt{2\log 4dT^2}}{\phi_*^2}\right)^{1+\alpha}\sum_{n_g(t)=T_0+1}^{n_g(T)}\frac{1}{n_g(t)^{\frac{1+\alpha}{2}}}$$

$$\le \frac{1}{\Delta_*^\alpha}\left(\frac{256\sigma x_{\max}^2 s_0 \sqrt{2\log 4dT^2}}{\phi_*^2}\right)^{1+\alpha}\sum_{n=T_0+1}^{T}\frac{1}{n^{\frac{1+\alpha}{2}}}.$$

If $\alpha < 1$, we have $\sum_{n=T_0+1}^{T} n^{-\frac{1+\alpha}{2}} \le \frac{2}{1-\alpha}T^{\frac{1-\alpha}{2}}$. If $\alpha = 1$, then $\sum_{n=T_0+1}^{T} n^{-1} \le \log T$. Then, we obtain the desired upper bound of the expected cumulative regret under the good events.

$$\sum_{n_g(t)=T_0+1}^{n_g(T)} \mathbb{E}\left[\text{reg}_t \mathbb{1}\left\{\text{reg}_t > 0, \Gamma_e(t), \Gamma_g(t)\right\}\right] \le$$

$$\begin{cases} \mathcal{O}\left(\frac{1}{(1-\alpha)\Delta_*^\alpha}\left(\frac{\sigma x_{\max}^2 s_0}{\phi_*^2}\right)^{1+\alpha}T^{\frac{1-\alpha}{2}}\left(\log d + \log T\right)^{\frac{1+\alpha}{2}}\right) & \alpha \in (0,1) \\ \mathcal{O}\left(\frac{1}{\Delta_*}\left(\frac{\sigma x_{\max}^2 s_0}{\phi_*^2}\right)(\log T)(\log d + \log T)\right) & \alpha = 1. \end{cases} \tag{41}$$

Now, we address the case where $\alpha > 1$. Let $T_1 = \left(\frac{256\sigma x_{\max}^2 s_0}{\Delta_* \phi_*^2}\right)^2 \cdot \left(2\log 4dT^2\right)$. We first sum the regret until $n_g(t) = T_1$.

$$\sum_{n_g(t)=T_0+1}^{T_1} \mathbb{E}\left[\text{reg}_t \mathbb{1}\left\{\text{reg}_t > 0, \Gamma_e(t), \Gamma_g(t)\right\}\right]$$

$$\le \sum_{n_g(t)=T_0+1}^{T_1} \left(\frac{256\sigma x_{\max}^2 s_0}{\phi_*^2}\sqrt{\frac{2\log 4dT^2}{n_g(t)}}\right)\min\left\{1, \left(\frac{256\sigma x_{\max}^2 s_0}{\Delta_* \phi_*^2}\sqrt{\frac{2\log 4dT^2}{n_g(t)}}\right)^\alpha\right\}$$

$$\le \sum_{n_g(t)=T_0+1}^{T_1} \frac{256\sigma x_{\max}^2 s_0}{\phi_*^2}\sqrt{\frac{2\log 4dT^2}{n_g(t)}}$$

$$= \frac{256\sigma x_{\max}^2 s_0 \sqrt{2\log 4dT^2}}{\phi_*^2}\sum_{n_g(t)=T_0+1}^{T_1}\frac{1}{\sqrt{n_g(t)}}$$

$$\le \frac{256\sigma x_{\max}^2 s_0 \sqrt{2\log 4dT^2}}{\phi_*^2}\cdot\frac{\sqrt{T_1}}{2}$$

$$= \frac{1}{2\Delta_*}\left(\frac{256\sigma x_{\max}^2 s_0}{\phi_*^2}\right)^2\left(2\log 4dT^2\right).$$

Then, we bound the sum of regret from $n_g(t) = T_1 + 1$ to $T$.

$$\sum_{n_g(t)=T_1+1}^{n_g(T)} \mathbb{E}\left[\text{reg}_t \mathbb{1}\left\{\text{reg}_t > 0, \Gamma_e(t), \Gamma_g(t)\right\}\right]$$

$$\leq \sum_{n_g(t)=T_1+1}^{T} \left(\frac{256\sigma x_{\max}^2 s_0}{\phi_*^2}\sqrt{\frac{2\log 4dT^2}{n_g(t)}}\right) \min\left\{1, \left(\frac{256\sigma x_{\max}^2 s_0}{\Delta_*\phi_*^2}\sqrt{\frac{2\log 4dT^2}{n_g(t)}}\right)^\alpha\right\}$$

$$\leq \sum_{n_g(t)=T_1+1}^{T} \left(\frac{256\sigma x_{\max}^2 s_0}{\phi_*^2}\sqrt{\frac{2\log 4dT^2}{n_g(t)}}\right) \left(\frac{256\sigma x_{\max}^2 s_0}{\Delta_*\phi_*^2}\sqrt{\frac{2\log 4dT^2}{n_g(t)}}\right)^\alpha$$

$$= \frac{1}{\Delta_*^\alpha}\left(\frac{256\sigma x_{\max}^2 s_0 \sqrt{2\log 4dT^2}}{\phi_*^2}\right)^{1+\alpha} \sum_{n_g(t)=T_1+1}^{T} \frac{1}{n_g(t)^{\frac{1+\alpha}{2}}}.$$

The summation is upper bounded by

$$\sum_{n_g(t)=T_1+1}^{T} \frac{1}{n_g(t)^{\frac{1+\alpha}{2}}} \leq \int_{T_1}^{T} \frac{1}{x^{\frac{1+\alpha}{2}}}\, dx$$

$$\leq \int_{T_1}^{\infty} \frac{1}{x^{\frac{1+\alpha}{2}}}\, dx$$

$$\leq \frac{2}{\alpha - 1} T_1^{\frac{1-\alpha}{2}}$$

$$= \frac{2}{\alpha - 1}\left(\frac{256\sigma x_{\max}^2 s_0 \sqrt{2\log 4dT^2}}{\Delta_*\phi_*^2}\right)^{1-\alpha}.$$

Therefore, we obtain that

$$\sum_{n_g(t)=T_1+1}^{n_g(T)} \mathbb{E}\left[\text{reg}_t \mathbb{1}\left\{\text{reg}_t > 0, \Gamma_e(t), \Gamma_g(t)\right\}\right] \leq \frac{2}{(\alpha-1)\Delta_*}\left(\frac{256\sigma x_{\max}^2 s_0}{\phi_*^2}\right)^2 \left(2\log 4dT^2\right).$$

$$(42)$$

Combining inequalities of Eq. (41) and Eq. (42), we obtain that

$$\sum_{n_g(t)=T_0+1}^{n_g(T)} \mathbb{E}\left[\text{reg}_t \mathbb{1}\left\{\text{reg}_t > 0, \Gamma_e(t), \Gamma_g(t)\right\}\right] \leq I_T\,,$$

where

$$I_T \leq \begin{cases} \mathcal{O}\left(\frac{1}{(1-\alpha)\Delta_*^\alpha}\left(\frac{\sigma x_{\max}^2 s_0}{\phi_*^2}\right)^{1+\alpha} T^{\frac{1-\alpha}{2}}\left(\log d + \log T\right)^{\frac{1+\alpha}{2}}\right) & \alpha \in (0,1)\,, \\ \mathcal{O}\left(\frac{1}{\Delta_*}\left(\frac{\sigma x_{\max}^2 s_0}{\phi_*^2}\right)(\log T)(\log d + \log T)\right) & \alpha = 1\,, \\ \mathcal{O}\left(\frac{1}{(\alpha-1)\Delta_*}\left(\frac{\sigma x_{\max}^2 s_0}{\phi_*^2}\right)^2 (\log d + \log T)\right) & \alpha > 1\,. \end{cases}$$

Putting all together, we obtain

$$\mathbb{E}\left[\sum_{t=1}^{T} \text{reg}_t\right] \leq 4x_{\max}bT_0 + 88x_{\max}b + \frac{49152 x_{\max}^5 b s_0^2}{\phi_*^4} + I_T\,.$$

which is the desired result. $\qquad\qquad\square$

## F.4 PROOF OF TECHNICAL LEMMAS

### F.4.1 PROOF OF LEMMA 16

*Proof of Lemma 16.* We use Lemma 19 with $w_t = \frac{1}{|\mathcal{T}_e(t)|}$. Define $\hat{\mathbf{\Sigma}}_t^g = \frac{1}{|\mathcal{T}_e(t)|}\sum_{i \in \mathcal{T}_e(t)} \mathbf{x}_{i,a_i}\mathbf{x}_{i,a_i}^\top$.
The lemma requires two events to hold: lower-boundedness of $\phi^2\left(\hat{\mathbf{\Sigma}}_t^g, S_0\right)$ and

$\max_{j \in [d]} \frac{1}{|\mathcal{T}_e(t)|} \left| \sum_{i \in \mathcal{T}_e(t)} \eta_i(\mathbf{x}_{i,a_i})_j \right| \leq \frac{\lambda_1}{4}$. Since $\hat{\boldsymbol{\Sigma}}_t^g$ is the empirical Gram matrix of randomly chosen features, its expectation is $\boldsymbol{\Sigma} = \frac{1}{K} \mathbb{E} \left[ \sum_{k=1}^K \mathbf{x}_{t,k} \mathbf{x}_{t,k}^\top \right]$. Then, by Lemma 22, with probability at least $1 - 2d^2 \exp\left( -\frac{\phi_*^4 |\mathcal{T}_e(t)|}{2048 \rho^2 x_{\max}^4 s_0^2} \right)$, $\phi^2 \left( \hat{\boldsymbol{\Sigma}}_t^g, S_0 \right) \geq \frac{\phi_*^2}{2\rho}$. Since $\{\eta_i(\mathbf{x}_{i,a_i})_j\}_{i \mathcal{T}_e(t)}$ is a sequence of conditionally $\sigma x_{\max}$ sub-Gaussian random variables as shown in the proof of Lemma 12, we apply the Azuma-Hoeffding inequality and obtain

$$\mathbb{P}\left( \frac{1}{|\mathcal{T}_e(t)|} \left| \sum_{i \in \mathcal{T}_e(t)} \eta_i(\mathbf{x}_{i,a_i})_j \right| \geq \frac{\lambda_1}{4} \right) \leq 2 \exp\left( -\frac{\lambda_1^2 |\mathcal{T}_e(t)|}{32 \sigma^2 x_{\max}^2} \right) .$$

Taking the union bound over $j \in [d]$ and plugging in the definition of $\lambda_1$ yields

$$\mathbb{P}\left( \max_{j \in [d]} \frac{1}{|\mathcal{T}_e(t)|} \left| \sum_{i \in \mathcal{T}_e(t)} \eta_i(\mathbf{x}_{i,a_i})_j \right| \geq \frac{\lambda_1}{4} \right) \leq 2d \exp\left( -\frac{\phi_*^4 h^2 |\mathcal{T}_e(t)|}{512 \rho^2 \sigma^2 x_{\max}^4 s_0^2} \right) .$$

Lemma 19 guarantees that under the two events, it holds that

$$\left\| \boldsymbol{\beta}^* - \widetilde{\boldsymbol{\beta}}_{|\mathcal{T}_e(t)|} \right\|_1 \leq \frac{2 s_0 \lambda_1}{\frac{\phi_*^2}{2\rho}}$$
$$= \frac{h}{2 x_{\max}} .$$

By taking the union bound over the two events, we conclude that

$$\mathbb{P}\left( \Gamma_e(t)^{\mathsf{c}} \right) \leq 2d^2 \exp\left( -\frac{\phi_*^4 |\mathcal{T}_e(t)|}{2048 \rho^2 x_{\max}^4 s_0^2} \right) + 2d \exp\left( -\frac{\phi_*^4 h^2 |\mathcal{T}_e(t)|}{512 \rho^2 \sigma^2 x_{\max}^4 s_0^2} \right) .$$

Since $t \in \mathcal{T}_g$, we know that $|\mathcal{T}_e(t)| > q(|\mathcal{T}_g(t)|)$ and $\mathcal{T}_g(t) + 1 = n_g(t)$. By $q(n) \geq \frac{\rho^2 x_{\max}^4 s_0^2}{\phi_*^4} \max\left\{ 2048 \log 2d^2(n+1)^3, \frac{512\sigma^2}{h^2} \log 2d(n+1)^3 \right\}$, we obtain

$$2d^2 \exp\left( -\frac{\phi_*^4 |\mathcal{T}_e(t)|}{2048 \rho^2 x_{\max}^4 s_0^2} \right) + 2d \exp\left( -\frac{\phi_*^4 h^2 |\mathcal{T}_e(t)|}{512 \rho^2 \sigma^2 x_{\max}^4 s_0^2} \right)$$
$$\leq 2d^2 \exp\left( -\frac{\phi_*^4 q(|\mathcal{T}_g(t)|)}{2048 \rho^2 x_{\max}^4 s_0^2} \right) + 2d \exp\left( -\frac{\phi_*^4 h^2 q(|\mathcal{T}_g(t)|)}{512 \rho^2 \sigma^2 x_{\max}^4 s_0^2} \right)$$
$$\leq 2d^2 \exp\left( -\log 2d^2 (|\mathcal{T}_g(t)| + 1)^3 \right) + 2d \exp\left( -\log 2d (|\mathcal{T}_g(t)| + 1)^3 \right)$$
$$= \frac{1}{(|\mathcal{T}_g(t)| + 1)^3} + \frac{1}{(|\mathcal{T}_g(t)| + 1)^3}$$
$$= \frac{2}{n_g(t)^3} ,$$

which is the desired result. $\qquad \square$

### F.4.2 PROOF OF LEMMA 17

*Proof of Lemma 17.* By the union bound, we have

$$\mathbb{P}\left( \Gamma_{N^-}(t)^{\mathsf{c}} \right) \leq \mathbb{P}\left( \Gamma_{N^-}(t)^{\mathsf{c}}, \bigcup_{i \in \mathcal{T}_g^-(t)} \Gamma_e(i) \right) + \sum_{i \in \mathcal{T}_g^-(t)} \mathbb{P}\left( \Gamma_e(i)^{\mathsf{c}} \right) .$$

By Lemma 16, the summation is bounded as the following:

$$\sum_{i \in \mathcal{T}_g^-(t)} \mathbb{P}\left(\Gamma_e(i)^{\mathsf{c}}\right) \leq \sum_{i \in \mathcal{T}_g^-(t)} \frac{2}{n_g(i)^3}$$

$$\leq \frac{2}{\left(\left\lfloor \frac{n_g(t)}{2} \right\rfloor + 1\right)^3} + \sum_{n_g = \left\lceil \frac{n_g(t)}{2} \right\rceil + 1} \frac{2}{n_g^3}$$

$$\leq \frac{16}{n_g(t)^3} + \int_{\frac{n_g(t)}{2}}^{n_g(t)} \frac{2}{x^3}\, dx$$

$$= \frac{16}{n_g(t)^3} + \frac{3}{n_g(t)^2}$$

$$\leq \frac{19}{n_g(t)^2}\,.$$

Under the event $\Gamma_e(i)$, $\Delta_i > 2h$ implies that for any $a \neq a_i^*$, it holds that

$$\mathbf{x}_{i,a_i^*}^\top \widetilde{\boldsymbol{\beta}}_{|\mathcal{T}_e(i)|} - \mathbf{x}_{i,a}^\top \widetilde{\boldsymbol{\beta}}_{|\mathcal{T}_e(i)|} > (\mathbf{x}_{i,a_i^*}^\top \widetilde{\boldsymbol{\beta}}_{|\mathcal{T}_e(i)|} - \mathbf{x}_{i,a}^\top \widetilde{\boldsymbol{\beta}}_{|\mathcal{T}_e(i)|}) - \left(\mathbf{x}_{i,a_i^*}^\top \boldsymbol{\beta}^* - \mathbf{x}_{i,a}^\top \boldsymbol{\beta}^*\right) + 2h$$

$$= \mathbf{x}_{i,a_i^*}^\top \left(\widetilde{\boldsymbol{\beta}}_{|\mathcal{T}_e(i)|} - \boldsymbol{\beta}^*\right) + \mathbf{x}_{i,a}^\top \left(\boldsymbol{\beta}^* - \widetilde{\boldsymbol{\beta}}_{|\mathcal{T}_e(i)|}\right) + 2h$$

$$\geq -2x_{\max} \left\|\widetilde{\boldsymbol{\beta}}_{|\mathcal{T}_e(i)|} - \boldsymbol{\beta}^*\right\|_1 + 2h$$

$$\geq h\,.$$

Then, the agent chooses $a_i = a_i^*$ in round $i$. Taking the contraposition, it means that $a_i \neq a_i^*$ implies $\Delta_i \leq 2h$ under the event $\Gamma_e(i)$. Then, we have that

$$\mathbb{P}\left(\Gamma_{N^-}(t)^{\mathsf{c}}, \bigcup_{i \in \mathcal{T}_g^-(t)} \Gamma_e(i)\right) \leq \mathbb{P}\left(\sum_{i \in \mathcal{T}_g^-(t)} \mathbb{1}\left\{\Delta_i \leq 2h\right\} > \frac{\phi_*^2}{64 x_{\max}^2 s_0} \left\lceil \frac{n_g(t)}{2} \right\rceil\right)\,.$$

$\{\mathbb{1}\{\Delta_i \leq 2h\}\}_{i \in \mathcal{T}_g^-(t)}$ is a sequence of independent Bernoulli random variables, whose expectation is at most $\left(\frac{2h}{\Delta_*}\right)^\alpha = \frac{\phi_*^2}{128 x_{\max}^2 s_0}$ by the margin condition and the definition of $h$. Then, by Hoeffding's inequality, we have

$$\mathbb{P}\left(\sum_{i \in \mathcal{T}_g^-(t)} \mathbb{1}\left\{\Delta_i \leq 2h\right\} > \frac{\phi_*^2}{64 x_{\max}^2 s_0} \left\lceil \frac{n_g(t)}{2} \right\rceil\right)$$

$$= \mathbb{P}\left(\sum_{i \in \mathcal{T}_g^-(t)} \left(\mathbb{1}\left\{\Delta_i \leq 2h\right\} - \mathbb{E}\left[\mathbb{1}\left\{\Delta_i \leq 2h\right\}\right]\right) > \frac{\phi_*^2}{64 x_{\max}^2 s_0} \left\lceil \frac{n_g(t)}{2} \right\rceil - \sum_{i \in \mathcal{T}_g^-(t)} \mathbb{E}\left[\mathbb{1}\left\{\Delta_i \leq 2h\right\}\right]\right)$$

$$\leq \mathbb{P}\left(\sum_{i \in \mathcal{T}_g^-(t)} \left(\mathbb{1}\left\{\Delta_i \leq 2h\right\} - \mathbb{E}\left[\mathbb{1}\left\{\Delta_i \leq 2h\right\}\right]\right) > \frac{\phi_*^2}{128 x_{\max}^2 s_0} \left\lceil \frac{n_g(t)}{2} \right\rceil\right)$$

$$\leq \exp\left(-2 \left\lceil \frac{n_g(t)}{2} \right\rceil \left(\frac{\phi_*^2}{128 x_{\max}^2 s_0}\right)^2\right)$$

$$\leq \exp\left(-\frac{n_g(t)\phi_*^4}{16384 x_{\max}^4 s_0^2}\right)\,.$$

Combining all together, we obtain

$$\mathbb{P}\left(\Gamma_{N^-}(t)^{\mathsf{c}}\right) \leq \frac{19}{n_g(t)^2} + \exp\left(-\frac{n_g(t)\phi_*^4}{16384 x_{\max}^4 s_0^2}\right)\,.$$

$\square$

### F.4.3 PROOF OF LEMMA 18

*Proof of Lemma 18.* Define the empirical Gram matrix of the latter half of the greedy actions as $\hat{\boldsymbol{\Sigma}}_t^- = \frac{1}{|\mathcal{T}_g^-(t)|} \sum_{i \in \mathcal{T}_g^-(t)} \mathbf{x}_{i,a_i} \mathbf{x}_{i,a_i}^\top$. Define the empirical Gram matrix of optimal features of the latter half of the greedy actions as $\hat{\boldsymbol{\Sigma}}_t^{*-} = \frac{1}{|\mathcal{T}_g^-(t)|} \sum_{i \in \mathcal{T}_g^-(t)} \mathbf{x}_{i,a_i^*} \mathbf{x}_{i,a_i^*}^\top$. We decompose $\hat{\boldsymbol{\Sigma}}_t^-$ as follows:

$$
\begin{aligned}
\hat{\boldsymbol{\Sigma}}_t^- &= \frac{1}{|\mathcal{T}_g^-(t)|} \sum_{i \in \mathcal{T}_g^-(t)} \mathbf{x}_{i,a_i} \mathbf{x}_{i,a_i}^\top \\
&= \frac{1}{|\mathcal{T}_g^-(t)|} \sum_{i \in \mathcal{T}_g^-(t)} \mathbf{x}_{i,a_i^*} \mathbf{x}_{i,a_i^*}^\top + \frac{1}{|\mathcal{T}_g^-(t)|} \sum_{i \in \mathcal{T}_g^-(t)} \mathbb{1}\{a_i \neq a_i^*\} \left( \mathbf{x}_{i,a_i} \mathbf{x}_{i,a_i}^\top - \mathbf{x}_{i,a_i^*} \mathbf{x}_{i,a_i^*}^\top \right) \\
&= \hat{\boldsymbol{\Sigma}}_t^{*-} + \frac{1}{|\mathcal{T}_g^-(t)|} \sum_{i \in \mathcal{T}_g^-(t)} \mathbb{1}\{a_i \neq a_i^*\} \mathbf{x}_{i,a_i} \mathbf{x}_{i,a_i}^\top - \frac{1}{|\mathcal{T}_g^-(t)|} \sum_{i \in \mathcal{T}_g^-(t)} \mathbb{1}\{a_i \neq a_i^*\} \mathbf{x}_{i,a_i^*} \mathbf{x}_{i,a_i^*}^\top .
\end{aligned}
$$

By Lemma 22, with probability at least $1 - 2d^2 \exp\left(-\frac{n_g(t)\phi_*^4}{4096 x_{\max}^4 s_0^2}\right)$, $\phi^2(\hat{\boldsymbol{\Sigma}}_{t-}^*, S_0) \geq \frac{\phi_*^2}{2}$. The compatibility constant of the second term is lower bounded by 0. The compatibility constant of the last term is lower bounded by $-\frac{N^-(t)}{|\mathcal{T}_g^-(t)|} \cdot 16 x_{\max}^2 s_0$ by Lemma 21. By the concavity of the compatibility constant, we have

$$
\phi^2\left(\hat{\boldsymbol{\Sigma}}_t^-, S_0\right) \geq \frac{\phi_*^2}{2} - \frac{16 x_{\max}^2 s_0 N^-(t)}{|\mathcal{T}_g^-(t)|} .
$$

Under the event $\Gamma_{N^-}(t)$, it holds that $\frac{16 x_{\max}^2 s_0 N^-(t)}{|\mathcal{T}_g^-(t)|} \geq \frac{\phi_*^2}{4}$. Therefore, we have $\phi^2\left(\hat{\boldsymbol{\Sigma}}_t^-, S_0\right) \geq \frac{\phi_*^2}{4}$. Let $\hat{\boldsymbol{\Sigma}}_t = \frac{1}{t} \sum_{i=1}^t \mathbf{x}_{i,a_i} \mathbf{x}_{i,a_i}$. Then, since $n_g(t) \geq n_e(t)$ and $|\mathcal{T}_g^-(t)| = \lceil \frac{n_g(t)}{2} \rceil$, we deduce that $|\mathcal{T}_g^-(t)| \geq \frac{t}{4}$. Then, it holds that

$$
\begin{aligned}
\phi^2\left(\hat{\boldsymbol{\Sigma}}_t, S_0\right) &\geq \frac{|\mathcal{T}_g(t)|}{t} \phi^2\left(\hat{\boldsymbol{\Sigma}}_t^-\right) \\
&\geq \frac{1}{4} \cdot \frac{\phi_*^2}{4} \\
&= \frac{\phi_*^2}{16} .
\end{aligned}
$$

By the choice of $\lambda_{2,t} = 4\sigma x_{\max} \sqrt{\frac{2 \log 4 d n_g(t)^2}{t}}$ and Lemma 19, for $t \in \mathcal{T}_g$,

$$
\mathbb{P}\left( \left\| \hat{\boldsymbol{\beta}}_{n_g(t)} - \boldsymbol{\beta}^* \right\|_1 \geq \frac{128 \sigma x_{\max} s_0}{\phi_*^2} \sqrt{\frac{2 \log 4 d n_g(t)^2}{t}}, \phi^2(\hat{\boldsymbol{\Sigma}}_t^-, S_0) \geq \frac{\phi_*^2}{2}, \Gamma_{N^-}(t) \right) \leq \frac{1}{n_g(t)^2} .
$$

By the union bound, we have

$$
\mathbb{P}\left( \Gamma_g(t)^{\mathsf{c}} \right) \leq \mathbb{P}\left( \Gamma_g(t)^{\mathsf{c}}, \phi^2(\hat{\boldsymbol{\Sigma}}_t^-, S_0) \geq \frac{\phi_*^2}{2}, \Gamma_{N^-}(t) \right) + \mathbb{P}\left( \phi^2(\hat{\boldsymbol{\Sigma}}_t^-, S_0) < \frac{\phi_*^2}{2} \right) + \mathbb{P}\left( \Gamma_{N^-}(t)^{\mathsf{c}} \right)
$$

$$
\leq \frac{1}{n_g(t)^2} + \frac{19}{n_g(t)^2} + 2d^2 \exp\left(-\frac{\phi_*^4 n_g(t)}{4096 x_{\max}^4 s_0^2}\right) + \exp\left(-\frac{\phi_*^4 n_g(t)}{16384 x_{\max}^4 s_0^2}\right) ,
$$

which completes the proof. □

## G TECHNICAL LEMMAS FOR APPENDICES E AND F

In this section, we state and prove the lemmas used for the analysis of Appendices E and F.

### G.1 ORACLE INEQUALITY FOR WEIGHTED SQUARED ERROR LASSO ESTIMATOR

We present the oracle inequality for the weighted squared error Lasso estimator. The proof mainly follows the proof of the standard Lasso oracle inequality with the compatibility condition (Bühlmann & Van De Geer, 2011), but with adaptive samples and weights. We provide the whole proof for completeness.

**Lemma 19.** *Let $\boldsymbol{\beta}^* \in \mathbb{R}^d$ be the true parameter vector and $\{\mathbf{x}_t\}_{t=1}^n$ be a sequence of random vectors in $\mathbb{R}^d$ adapted to a filtration $\{\mathcal{F}_t\}_{t=0}^n$. Let $r_t$ be the noised observation given by $\mathbf{x}_t^\top \boldsymbol{\beta}^* + \eta_t$, where $\eta_t$ is a real-valued random variable that is $\mathcal{F}_{t+1}$-measurable. For non-negative constants $w_1, w_2, \ldots, w_n$ and $\lambda_n > 0$, define the weighted squared error Lasso estimator by*

$$\hat{\boldsymbol{\beta}} = \operatorname*{argmin}_{\boldsymbol{\beta} \in \mathbb{R}^d} \lambda_n \|\boldsymbol{\beta}\|_1 + \sum_{t=1}^n w_t \left(r_t - \mathbf{x}_t^\top \boldsymbol{\beta}\right)^2 . \tag{43}$$

*Let $\hat{\mathbf{V}}_n = \sum_{t=1}^n w_t \mathbf{x}_t \mathbf{x}_t^\top$ and assume $\phi^2\left(\hat{\mathbf{V}}_n, S_0\right) \geq \phi_n^2 > 0$. Then, under the event $\left\{\omega \in \Omega : \max_{j \in [d]} \left|\sum_{t=1}^n w_t \eta_t\left(\mathbf{x}_t\right)_j\right| \leq \frac{\lambda_n}{4}\right\}$, $\hat{\boldsymbol{\beta}}$ satisfies*

$$\left\|\boldsymbol{\beta}^* - \hat{\boldsymbol{\beta}}\right\|_1 \leq \frac{2\lambda_n s_0}{\phi_n^2} .$$

*Proof of Lemma 19.* Define $\mathbf{X_w} = \left(\sqrt{w_1}\mathbf{x}_1 \quad \sqrt{w_2}\mathbf{x}_2 \quad \cdots \sqrt{w_n}\mathbf{x}_n\right) \in \mathbb{R}^{d \times n}$, $\mathbf{r_w} = \left(\sqrt{w_1}r_1 \quad \sqrt{w_2}r_2 \quad \cdots \quad \sqrt{w_n}r_n\right)^\top \in \mathbb{R}^n$, and $\boldsymbol{\eta_w} = \left(\sqrt{w_1}\eta_1 \quad \sqrt{w_2}r_2 \quad \cdots \sqrt{w_n}\eta_n\right)^\top \in \mathbb{R}^n$. The minimization problem (43) can be rewritten as

$$\operatorname*{argmin}_{\boldsymbol{\beta} \in \mathbb{R}^d} \lambda_n \|\boldsymbol{\beta}\|_1 + \left\|\mathbf{r_w} - \mathbf{X_w}^\top \boldsymbol{\beta}\right\|_2^2 .$$

Since $\hat{\boldsymbol{\beta}}$ achieves the minimum, it holds that

$$\lambda_n \|\hat{\boldsymbol{\beta}}\|_1 + \left\|\mathbf{r_w} - \mathbf{X_w}^\top \hat{\boldsymbol{\beta}}\right\|_2^2 \leq \lambda_n \|\boldsymbol{\beta}^*\|_1 + \left\|\mathbf{r_w} - \mathbf{X_w}^\top \boldsymbol{\beta}^*\right\|_2^2 . \tag{44}$$

Using that $\mathbf{r_w} = \boldsymbol{\eta_w} + \mathbf{X_w}^\top \boldsymbol{\beta}^*$, expand the squares as

$$\left\|\mathbf{r_w} - \mathbf{X_w}^\top \hat{\boldsymbol{\beta}}\right\|_2^2 = \left\|\boldsymbol{\eta_w} + \mathbf{X_w}^\top (\boldsymbol{\beta}^* - \hat{\boldsymbol{\beta}})\right\|_2^2$$

$$= \|\boldsymbol{\eta_w}\|_2^2 + 2\boldsymbol{\eta_w}^\top \mathbf{X_w}^\top (\boldsymbol{\beta}^* - \hat{\boldsymbol{\beta}}) + \left\|\mathbf{X_w}^\top (\boldsymbol{\beta}^* - \hat{\boldsymbol{\beta}})\right\|_2^2 . \tag{45}$$

By plugging Eq. (45) into Eq. (44) and reordering the terms, we have

$$\left\|\mathbf{X_w}^\top (\boldsymbol{\beta}^* - \hat{\boldsymbol{\beta}})\right\|_2^2 \leq \lambda_n \left(\|\boldsymbol{\beta}^*\|_1 - \|\hat{\boldsymbol{\beta}}\|_1\right) + 2\boldsymbol{\eta_w}^\top \mathbf{X_w}^\top (\hat{\boldsymbol{\beta}} - \boldsymbol{\beta}^*)$$

$$\leq \lambda_n \left(\|\boldsymbol{\beta}^*\|_1 - \|\hat{\boldsymbol{\beta}}\|_1\right) + 2 \|\mathbf{X_w}\boldsymbol{\eta_w}\|_\infty \|\boldsymbol{\beta}^* - \hat{\boldsymbol{\beta}}\|_1 . \tag{46}$$

Note that $\mathbf{X_w}\boldsymbol{\eta_w}$ is a $d$-dimensional vector whose $j$-th component is $(\mathbf{X_w}\boldsymbol{\eta_w})_j = \sum_{t=1}^n w_t \eta_i(\mathbf{x}_i)_j$. Under the event $\left\{\omega \in \Omega : \max_{j \in [d]} \left|\sum_{t=1}^n w_t \eta_t(\mathbf{x}_t)_j\right| \leq \frac{\lambda_n}{4}\right\}$, we have $\|\mathbf{X_w}\boldsymbol{\eta_w}\|_\infty \leq \frac{\lambda_n}{4}$. Plug it into the Eq. (46) and obtain

$$\left\|\mathbf{X_w}^\top (\boldsymbol{\beta}^* - \hat{\boldsymbol{\beta}})\right\|_2^2 \leq \lambda_n \left(\|\boldsymbol{\beta}^*\|_1 - \|\hat{\boldsymbol{\beta}}\|_1\right) + \frac{\lambda_n}{2} \|\boldsymbol{\beta}^* - \hat{\boldsymbol{\beta}}\|_1 . \tag{47}$$

On the other hand, by the definition of $S_0$, we have

$$\|\boldsymbol{\beta}^*\|_1 - \|\hat{\boldsymbol{\beta}}\|_1 = \|\boldsymbol{\beta}^*_{S_0}\|_1 - \|\hat{\boldsymbol{\beta}}_{S_0}\|_1 - \|\hat{\boldsymbol{\beta}}_{S_0^c}\|_1$$

$$\leq \|(\boldsymbol{\beta}^* - \hat{\boldsymbol{\beta}})_{S_0}\|_1 - \|\hat{\boldsymbol{\beta}}_{S_0^c}\|_1$$

$$= \|(\boldsymbol{\beta}^* - \hat{\boldsymbol{\beta}})_{S_0}\|_1 - \|(\boldsymbol{\beta}^* - \hat{\boldsymbol{\beta}})_{S_0^c}\|_1 . \tag{48}$$

Also, note that
$$\|\boldsymbol{\beta}^* - \hat{\boldsymbol{\beta}}\|_1 = \|(\boldsymbol{\beta}^* - \hat{\boldsymbol{\beta}})_{S_0}\|_1 + \|(\boldsymbol{\beta}^* - \hat{\boldsymbol{\beta}})_{S_0^c}\|_1 . \tag{49}$$
By plugging Eq. (48) and Eq. (49) into Eq. (47), we have
$$0 \le \left\|\mathbf{X}_{\mathbf{w}}^\top(\boldsymbol{\beta}^* - \hat{\boldsymbol{\beta}})\right\|_2^2 \le \frac{3\lambda_n}{2}\|(\boldsymbol{\beta}^* - \hat{\boldsymbol{\beta}})_{S_0}\|_1 - \frac{\lambda_n}{2}\|(\boldsymbol{\beta}^* - \hat{\boldsymbol{\beta}})_{S_0^c}\|_1 . \tag{50}$$
Eq. (50) implies $\|(\boldsymbol{\beta}^* - \hat{\boldsymbol{\beta}})_{S_0^c}\|_1 \le 3\|(\boldsymbol{\beta}^* - \hat{\boldsymbol{\beta}})_{S_0}\|_1$, by which we conclude $\boldsymbol{\beta}^* - \hat{\boldsymbol{\beta}} \in \mathbb{C}(S_0)$.

Then, we have the following result:
$$\begin{aligned}
\left\|\mathbf{X}_{\mathbf{w}}^\top(\boldsymbol{\beta}^* - \hat{\boldsymbol{\beta}})\right\|_2^2 + \frac{\lambda_n}{2}\|\boldsymbol{\beta}^* - \hat{\boldsymbol{\beta}}\|_1 &= \left\|\mathbf{X}_{\mathbf{w}}^\top(\boldsymbol{\beta}^* - \hat{\boldsymbol{\beta}})\right\|_2^2 + \frac{\lambda_n}{2}\left(\|(\boldsymbol{\beta}^* - \hat{\boldsymbol{\beta}})_{S_0}\|_1 + \|(\boldsymbol{\beta}^* - \hat{\boldsymbol{\beta}})_{S_0^c}\|_1\right) \\
&\le 2\lambda_n\|(\boldsymbol{\beta}^* - \hat{\boldsymbol{\beta}})_{S_0}\|_1 \\
&\le 2\lambda_n\sqrt{\frac{s_0\left\|\mathbf{X}_{\mathbf{w}}(\boldsymbol{\beta}^* - \hat{\boldsymbol{\beta}})\right\|_2^2}{\phi_n^2}} \\
&\le \left\|\mathbf{X}_{\mathbf{w}}^\top(\boldsymbol{\beta}^* - \hat{\boldsymbol{\beta}})\right\|_2^2 + \frac{\lambda_n^2 s_0}{\phi_1^2} ,
\end{aligned}$$
where the first inequality comes from Eq. (50), the second inequality holds due to the compatibility condition of $\hat{\mathbf{V}}_n = \mathbf{X}_{\mathbf{w}}\mathbf{X}_{\mathbf{w}}^\top$, and the last inequality is the AM-GM inequality, namely $2\sqrt{ab} \le a+b$. Therefore, we have $\|\boldsymbol{\beta}^* - \hat{\boldsymbol{\beta}}\|_1 \le \frac{2\lambda_n s_0}{\phi_n^2}$. $\square$

## G.2 PROPERTIES OF COMPATIBILITY CONSTANTS

For this subsection, we assume that $S_0 \subset [d]$ is a fixed set and denote the compatibility constant of a matrix $\mathbf{A}$ as $\phi^2(\mathbf{A})$ instead of $\phi^2(\mathbf{A}, S_0)$ for simplicity.

**Lemma 20** (Concavity of Compatibility Constant)**.** *Let* $\mathbf{A}, \mathbf{B} \in \mathbb{R}^{d\times d}$ *be square matrices. Then,*
$$\phi^2(\mathbf{A} + \mathbf{B}) \ge \phi^2(\mathbf{A}) + \phi^2(\mathbf{B}) .$$

*Proof of Lemma 20.* By definition,
$$\begin{aligned}
\phi^2(\mathbf{A} + \mathbf{B}) &= \inf_{\boldsymbol{\beta}\in\mathbb{C}(S_0)\backslash\{\mathbf{0}_d\}} \frac{s_0\boldsymbol{\beta}^\top(\mathbf{A} + \mathbf{B})\boldsymbol{\beta}}{\|\boldsymbol{\beta}_{S_0}\|_1^2} \\
&= \inf_{\boldsymbol{\beta}\in\mathbb{C}(S_0)\backslash\{\mathbf{0}_d\}} \left(\frac{s_0\boldsymbol{\beta}^\top\mathbf{A}\boldsymbol{\beta}}{\|\boldsymbol{\beta}_{S_0}\|_1^2} + \frac{s_0\boldsymbol{\beta}^\top\mathbf{B}\boldsymbol{\beta}}{\|\boldsymbol{\beta}_{S_0}\|_1^2}\right) \\
&\ge \inf_{\boldsymbol{\beta}\in\mathbb{C}(S_0)\backslash\{\mathbf{0}_d\}} \frac{s_0\boldsymbol{\beta}^\top\mathbf{A}\boldsymbol{\beta}}{\|\boldsymbol{\beta}_{S_0}\|_1^2} + \inf_{\boldsymbol{\beta}'\in\mathbb{C}(S_0)\backslash\{\mathbf{0}_d\}} \frac{s_0\boldsymbol{\beta}'^\top\mathbf{B}\boldsymbol{\beta}'}{\|\boldsymbol{\beta}'_{S_0}\|_1^2} \\
&= \phi^2(\mathbf{A}) + \phi^2(\mathbf{B}) .
\end{aligned}$$
$\square$

**Lemma 21.** *Let* $\mathbf{x}$ *be a $d$-dimensional random vector and* $\boldsymbol{\Sigma} = \mathbb{E}\left[\mathbf{x}\mathbf{x}^\top\right] \in \mathbb{R}^{d\times d}$*. Assume that* $\|\mathbf{x}\|_\infty \le x_{\max}$ *almost surely. Then, for any* $\mathbf{v} \in \mathbb{C}(S_0) \backslash \{\mathbf{0}_d\}$*, it holds that*
$$0 \le \frac{s_0\mathbf{v}^\top\boldsymbol{\Sigma}\mathbf{v}}{\|\mathbf{v}_{S_0}\|_1^2} \le 16x_{\max}^2 s_0 .$$
*Consequently, it holds that* $0 \le \phi^2(\boldsymbol{\Sigma}) \le 16x_{\max}^2 s_0$ *and* $\phi^2(-\boldsymbol{\Sigma}) \ge -16x_{\max}^2 s_0$.

*Proof of Lemma 21.* From $\mathbf{v}^\top\left(\mathbf{x}\mathbf{x}^\top\right)\mathbf{v} = \left(\mathbf{x}^\top\mathbf{v}\right)^2 \ge 0$, it holds that
$$\begin{aligned}
\mathbf{v}^\top\boldsymbol{\Sigma}\mathbf{v} &= \mathbf{v}^\top\mathbb{E}\left[\mathbf{x}\mathbf{x}^\top\right]\mathbf{v} \\
&= \mathbb{E}\left[\mathbf{v}^\top\left(\mathbf{x}\mathbf{x}^\top\right)\mathbf{v}\right] \\
&\ge 0 ,
\end{aligned}$$

which proves $0 \leq \frac{s_0 \mathbf{v}^\top \mathbf{\Sigma} \mathbf{v}}{\|\mathbf{v}_{S_0}\|_1^2}$. The upper bound is proved as follows:

$$
\begin{aligned}
\mathbf{v}^\top \mathbf{\Sigma} \mathbf{v} &= \mathbb{E}\left[\mathbf{v}^\top \left(\mathbf{x}\mathbf{x}^\top\right) \mathbf{v}\right] \\
&= \mathbb{E}\left[\left(\mathbf{x}^\top \mathbf{v}\right)^2\right] \\
&\leq \mathbb{E}\left[\left(x_{\max} \|\mathbf{v}\|_1\right)^2\right] \\
&= x_{\max}^2 \|\mathbf{v}\|_1^2
\end{aligned}
\tag{51}
$$

where the inequality holds by Hölder's inequality and $\|\mathbf{x}\|_\infty \leq x_{\max}$. Since $\mathbf{v} \in \mathbb{C}(S_0)$, we have $\|\mathbf{v}\|_1 = \|\mathbf{v}_{S_0}\|_1 + \|\mathbf{v}_{S_0^c}\|_1 \leq 4\|\mathbf{v}_{S_0}\|_1$. Therefore, we have

$$
\begin{aligned}
\frac{s_0 \mathbf{v}^\top \mathbf{\Sigma} \mathbf{v}}{\|\mathbf{v}_{S_0}\|_1^2} &\leq \frac{s_0 x_{\max}^2 \|\mathbf{v}\|_1^2}{\|\mathbf{v}_{S_0}\|_1^2} \\
&\leq \frac{s_0 x_{\max}^2 \left(16\|\mathbf{v}_{S_0}\|_1^2\right)}{\|\mathbf{v}_{S_0}\|_1^2} \\
&= 16 x_{\max}^2 s_0,
\end{aligned}
$$

where the first inequality comes from inequality (51) and the second inequality holds by $\|\mathbf{v}\|_1 \leq 4\|\mathbf{v}_{S_0}\|_1$. □

**Lemma 22.** *Let $\{\mathbf{x}_t\}_{t=1}^\tau$ be a sequence of random vectors in $\mathbb{R}^d$ adapted to filtration $\{\mathcal{F}_t\}_{t=0}^\tau$ such that $\|\mathbf{x}_t\|_\infty \leq x_{\max}$ holds for all $t \geq 1$. Let $\hat{\mathbf{\Sigma}}_\tau = \frac{1}{\tau} \sum_{t=1}^\tau \mathbf{x}_t \mathbf{x}_t^\top$ and $\bar{\mathbf{\Sigma}}_\tau = \frac{1}{\tau} \sum_{t=1}^\tau \mathbb{E}\left[\mathbf{x}_t \mathbf{x}_t^\top \mid \mathcal{F}_{t-1}\right]$. If $\phi^2\left(\bar{\mathbf{\Sigma}}_\tau\right) \geq \phi_0^2$ for some $\phi_0 > 0$, then with probability at least $1 - 2d^2 \exp\left(-\frac{\tau \phi_0^4}{2048 x_{\max}^4 s_0^2}\right)$, $\phi^2(\hat{\mathbf{\Sigma}}_\tau) \geq \frac{\phi_0^2}{2}$ holds.*

*Proof of Lemma 22.* Let $\gamma_t^{ij} = (\mathbf{x}_t)_i \cdot (\mathbf{x}_t)_j - \mathbb{E}\left[(\mathbf{x}_t)_i \cdot (\mathbf{x}_t)_j \mid \mathcal{F}_{t-1}\right]$ for $1 \leq i, j \leq d$. Then, $\mathbb{E}\left[\gamma_t^{ij} \mid \mathcal{F}_{t-1}\right] = 0$ and $\left|\gamma_t^{ij}\right| \leq 2x_{\max}^2$. By the Azuma-Hoeffding inequality,

$$
\mathbb{P}\left(\left|\frac{1}{\tau} \sum_{t=1}^\tau \gamma_t^{ij}\right| \geq \varepsilon\right) \leq 2\exp\left(-\frac{\tau \varepsilon^2}{2x_{\max}^4}\right).
$$

By taking the union bound over $1 \leq i, j \leq d$, we have

$$
\mathbb{P}\left(\|\hat{\mathbf{\Sigma}}_\tau - \bar{\mathbf{\Sigma}}_\tau\|_\infty \geq \varepsilon\right) \leq 2d^2 \exp\left(-\frac{\tau \varepsilon^2}{2x_{\max}^4}\right).
$$

Alternatively, by taking $\varepsilon = \frac{\phi_0^2}{32 s_0}$, we obtain that with probability at least $1 - 2d^2 \exp\left(-\frac{\tau \phi_0^2}{2048 x_{\max}^4 s_0^2}\right)$,

$$
\|\hat{\mathbf{\Sigma}}_\tau - \bar{\mathbf{\Sigma}}_\tau\|_\infty \leq \frac{\phi_0^2}{32 s_0}.
$$

Then, by Lemma 30, we conclude that with probability at least $1 - 2d^2 \exp\left(-\frac{\tau \phi_0^2}{2048 x_{\max}^4 s_0^2}\right)$, $\phi^2(\hat{\mathbf{\Sigma}}_\tau) \geq \frac{\phi_0^2}{2}$ holds. □

**Lemma 23.** *Let $\{\mathbf{x}_t\}_{t=1}^\tau$ be a sequence of random vectors in $\mathbb{R}^d$ adapted to filtration $\{\mathcal{F}_t\}_{t=0}^\tau$ such that $\|\mathbf{x}_t\|_\infty \leq x_{\max}$ for all $t \geq 1$. Let $\hat{\mathbf{V}}_t := \sum_{i=1}^t \mathbf{x}_i \mathbf{x}_i^\top$ and $\bar{\mathbf{V}}_t := \sum_{i=1}^t \mathbb{E}\left[\mathbf{x}_i \mathbf{x}_i^\top \mid \mathcal{F}_{i-1}\right]$. Suppose that there exists a constant $\phi_0 > 0$ such that $\phi^2\left(\bar{\mathbf{V}}_t\right) \geq \phi_0^2 t$ for all $t \geq 1$. For any $\delta \in (0, 1]$, with probability at least $1 - \delta$, $\phi^2\left(\hat{\mathbf{V}}_t\right) \geq \frac{\phi_0^2 t}{2}$ holds for all $t \geq \frac{2048 x_{\max}^4 s_0^2}{\phi_0^4}\left(\log \frac{d^2}{\delta} + 2\log \frac{64 x_{\max}^2 s_0}{\phi_0^2}\right) + 1$.*

*Proof of Lemma 23.* By Lemma 22 with $\hat{\boldsymbol{\Sigma}}_t = \frac{1}{t}\hat{\mathbf{V}}_t$ and $\bar{\boldsymbol{\Sigma}}_t = \frac{1}{t}\overline{\mathbf{V}}_t$, $\phi^2\left(\frac{1}{t}\hat{\mathbf{V}}_t\right) \geq \frac{\phi_0^2}{2}$ holds with probability at least $1 - 2d^2\exp\left(-\frac{\phi_0^4 t}{2048 x_{\max}^4 s_0^2}\right)$. Let $t_0 = \left\lceil \frac{2048 x_{\max}^4 s_0^2}{\phi_0^4}\left(\log\frac{d^2}{\delta} + 2\log\frac{64 x_{\max}^2 s_0}{\phi_0^2}\right)\right\rceil$. By taking the union bound over $t \geq t_0 + 1$, we conclude that

$$\mathbb{P}\left(\exists t \geq t_0 + 1 : \phi^2\left(\hat{\mathbf{V}}_t\right) < \frac{\phi_0^2 t}{2}\right) \leq \sum_{t=t_0+1}^{\infty} \mathbb{P}\left(\phi^2\left(\hat{\mathbf{V}}_t\right) < \frac{\phi_0^2 t}{2}\right)$$

$$\leq \sum_{t=t_0+1}^{\infty} 2d^2 \exp\left(-\frac{\phi_0^4 t}{2048 x_{\max}^4 s_0^2}\right)$$

$$\leq 2d^2 \int_{t_0}^{\infty} \exp\left(-\frac{\phi_0^4 x}{2048 x_{\max}^4 s_0^2}\right) dx$$

$$= 2d^2\left(\frac{2048 x_{\max}^4 s_0^2}{\phi_0^4}\exp\left(-\frac{\phi_0^4 t_0}{2048 x_{\max}^4 s_0^2}\right)\right)$$

$$\leq \delta,$$

where the last inequality holds by $t_0 \geq \frac{2048 x_{\max}^4 s_0^2}{\phi_0^4}\left(\log\frac{d^2}{\delta} + 2\log\frac{64 x_{\max}^2 s_0}{\phi_0^2}\right)$. $\qquad\square$

## G.3 GUARANTEES OF GREEDY ACTION SELECTION

**Lemma 24.** *Suppose* $a_t = \operatorname{argmax}_{a \in \mathcal{A}} \mathbf{x}_{t,a}^\top \hat{\boldsymbol{\beta}}_{t-1}$ *is chosen greedily with respect to an estimator* $\hat{\boldsymbol{\beta}}_{t-1}$ *in round* $t$. *Then, the instantaneous regret in round* $t$ *is at most* $2x_{\max}\|\boldsymbol{\beta}^* - \hat{\boldsymbol{\beta}}_{t-1}\|_1$. *Consequently, if* $\Delta_t > 2x_{\max}\|\boldsymbol{\beta}^* - \hat{\boldsymbol{\beta}}_{t-1}\|_1$, *then* $a_t = a_t^*$.

*Proof of Lemma 24.* Let $a_t^* = \operatorname{argmax}_{a \in \mathcal{A}} \mathbf{x}_{t,a}^\top \boldsymbol{\beta}^*$. By the choice of $a_t$, the following inequality holds:

$$\mathbf{x}_{t,a_t}^\top \hat{\boldsymbol{\beta}}_{t-1} - \mathbf{x}_{t,a_t^*}^\top \hat{\boldsymbol{\beta}}_{t-1} \geq 0. \tag{52}$$

Then, the instantaneous regret is bounded as the following:

$$\operatorname{reg}_t = \mathbf{x}_{t,a_t^*}^\top \boldsymbol{\beta}^* - \mathbf{x}_{t,a_t}^\top \boldsymbol{\beta}^*$$

$$\leq \left(\mathbf{x}_{t,a_t^*}^\top \boldsymbol{\beta}^* - \mathbf{x}_{t,a_t}^\top \boldsymbol{\beta}^*\right) + \left(\mathbf{x}_{t,a_t}^\top \hat{\boldsymbol{\beta}}_{t-1} - \mathbf{x}_{t,a_t^*}^\top \hat{\boldsymbol{\beta}}_{t-1}\right)$$

$$= \mathbf{x}_{t,a_t^*}^\top\left(\boldsymbol{\beta}^* - \hat{\boldsymbol{\beta}}_{t-1}\right) + \mathbf{x}_{t,a_t}^\top\left(\hat{\boldsymbol{\beta}}_{t-1} - \boldsymbol{\beta}^*\right)$$

$$\leq \|\mathbf{x}_{t,a_t^*}\|_\infty\left\|\boldsymbol{\beta}^* - \hat{\boldsymbol{\beta}}_{t-1}\right\|_1 + \|\mathbf{x}_{t,a_t}\|_\infty\left\|\boldsymbol{\beta}^* - \hat{\boldsymbol{\beta}}_{t-1}\right\|_1$$

$$\leq 2x_{\max}\left\|\boldsymbol{\beta}^* - \hat{\boldsymbol{\beta}}_{t-1}\right\|_1, \tag{53}$$

where the first inequality holds by inequality (52) and the second inequality is due to Hölder's inequality. This result proves the first part of the lemma.

Suppose that $\Delta_t > 2x_{\max}\|\boldsymbol{\beta}^* - \hat{\boldsymbol{\beta}}_{t-1}\|_1$. Then, the instantaneous regret in round $t$ is either 0 or no less than $\Delta_t$, which implies that $\operatorname{reg}_t$ is either 0 or greater than $2x_{\max}\|\boldsymbol{\beta}^* - \hat{\boldsymbol{\beta}}_{t-1}\|_1$. By (53) we have $\operatorname{reg}_t \leq 2x_{\max}\|\boldsymbol{\beta}^* - \hat{\boldsymbol{\beta}}_{t-1}\|_1$. Therefore, the $\operatorname{reg}_t$ must be 0, which implies $a_t = a_t^*$. $\qquad\square$

## G.4 BEHAVIOR OF $\log\log n$

Let $b > 1$ be a constant and define $f(x) = \frac{2\log\log 2x + b}{x}$ for $x \geq 2$. The derivative of $f(x)$ is $f'(x) = \frac{\frac{2}{\log 2x} - 2\log\log 2x - b}{x^2}$. $f'(x)$ is decreasing in $x$ and $f'(2) < 0$, therefore $f(x)$ is decreasing for $x \geq 2$.

**Lemma 25.** *Suppose* $C \geq 2$, $b \geq 1$, *and* $n \geq Cb + 2C\log\left(2\log 2C + b\right)$. *Then,* $f(n) = \frac{2\log\log 2n + b}{n} \leq \frac{1}{C}$.

*Proof of Lemma 25.* Let $n_0 = Cb + 2C \log (2 \log 2C + b)$. Since $n_0 \geq Cb \geq 2$ and $f(x)$ is decreasing for $x \geq 2$, it is sufficient to show that $f(n_0) \leq \frac{1}{C}$. We rewrite $f(n_0) - \frac{1}{C}$ as the following:

$$
\begin{aligned}
f(n_0) - \frac{1}{C} &= \frac{2 \log \log 2n_0 + b}{n_0} - \frac{1}{C} \\
&= \frac{2C \log \log 2n_0 + Cb - n_0}{Cn_0} \\
&= \frac{2C \log \log 2n_0 - 2C \log (2 \log 2C + b)}{Cn_0} \\
&= \frac{2}{n_0} \Big( \log \log 2C \left( b + 2 \log (2 \log 2C + b) \right) - \log (2 \log 2C + b) \Big).
\end{aligned}
$$

Now, it is sufficient to prove $\log 2C(b + 2 \log (2 \log 2C + b)) \leq 2 \log 2C + b$. We prove it by applying $\log x \leq \frac{x}{e}$ for all $x > 0$ multiple times.

$$
\begin{aligned}
\log 2C \left( b + 2 \log (2 \log 2C + b) \right) &= \log 2C + \log \left( b + 2 \log(2 \log 2C + b) \right) \\
&\leq \log 2C + \log \left( b + \frac{2}{e} (2 \log 2C + b) \right) \\
&= \log 2C + \log \left( \frac{4}{e} \log 2C + \left( 1 + \frac{2}{e} \right) b \right) \\
&\leq \log 2C + \frac{4}{e^2} \log 2C + \frac{1 + \frac{2}{e}}{e} b \\
&\leq 2 \log 2C + b.
\end{aligned}
$$

$\square$

**Lemma 26.** *Let $f(x) = \frac{2 \log \log 2x + \log b}{x}$ for a constant $b \geq 1$ and $x \geq 2$. Suppose $8 \leq A < B$ are integers and $r \geq 0$ is a nonnegative real number. Then,*

$$
\sum_{n=A+1}^{B} f(n)^r \leq
\begin{cases}
\frac{1}{1-r} B^{1-r} (2 \log \log 2B + b)^r & r \in [0, 1) \\
(\log B) (2 \log \log 2B + b) & r = 1 \\
\frac{2r-1}{(r-1)^2} \cdot \frac{(2 \log \log 2A + b)^r}{A^{r-1}} & r \in (1, 2] \\
\frac{2}{r-1} \cdot \frac{(2 \log \log 2A + b)^r}{A^{r-1}} & r > 2
\end{cases}
$$

*holds.*

*Proof of Lemma 26.* Since $f(x)$ is decreasing for $x \geq 2$, we have

$$
\sum_{n=A+1}^{B} f(n)^r \leq \int_A^B f(x)^r \, dx.
$$

We bound $\int_A^B \left( \frac{2 \log \log 2x + b}{x} \right)^r dx$ for each case of $r$.
*Case 1: $r \in [0, 1)$*

$$
\begin{aligned}
\int_A^B \left( \frac{2 \log \log 2x + b}{x} \right)^r dx &\leq \int_A^B \left( \frac{2 \log \log 2B + b}{x} \right)^r dx \\
&= (2 \log \log 2B + b)^r \int_A^B x^{-r} \, dx \\
&= (2 \log \log 2B + b)^r \cdot \frac{1}{1-r} \left( B^{1-r} - A^{1-r} \right) \\
&\leq \frac{1}{1-r} B^{1-r} (2 \log \log 2B + b)^r.
\end{aligned}
$$

*Case 2: $r = 1$*

$$\int_A^B \frac{2 \log \log 2x + b}{x} \, dx \leq \int_A^B \frac{2 \log \log 2B + b}{x} \, dx$$

$$= (2 \log \log 2B + b) \int_A^B \frac{1}{x} \, dx$$

$$= (2 \log \log 2B + b) (\log B - \log A)$$

$$\leq (\log B) (2 \log \log 2B + b) \ .$$

*Case 3: $r \in (1, 2]$*
First, apply Jensen's inequality to $x^r$ with $p := \frac{2 \log \log 2A}{2 \log \log 2A + b}$ to obtain

$$(2 \log \log 2x + b)^r = \left( p \cdot \frac{2 \log \log 2x}{p} + (1 - p) \cdot \frac{b}{1 - p} \right)^r$$

$$\leq p \left( \frac{2 \log \log 2x}{p} \right)^r + (1 - p) \left( \frac{b}{1 - p} \right)^r$$

$$= p^{1-r} (2 \log \log 2x)^r + (1 - p)^{1-r} b^r \ .$$

Then, the integral can be split into

$$\int_A^B \left( \frac{2 \log \log 2x + b}{x} \right)^r \, dx \leq \underbrace{p^{1-r} \int_A^B \left( \frac{2 \log \log 2x}{x} \right)^r \, dx}_{I_1} + \underbrace{(1 - p)^{1-r} \int_A^B \left( \frac{b}{x} \right)^r \, dx}_{I_2} \ .$$

$I_2$ is bounded by

$$(1 - p)^{1-r} \int_A^B \left( \frac{b}{x} \right)^r \, dx = (1 - p)^{1-r} \cdot \frac{b^r}{r - 1} \left( \frac{1}{A^{r-1}} - \frac{1}{B^{r-1}} \right)$$

$$\leq \frac{(1 - p)^{1-r} b^r}{(r - 1) A^{r-1}}$$

$$= \frac{(1 - p) \left( \frac{b}{1-p} \right)^r}{(r - 1) A^{r-1}}$$

$$= \frac{(1 - p) (2 \log \log 2A + b)^r}{(r - 1) A^{r-1}} \ ,$$

where the last equality holds by the definition of $p$.
To bound $I_1$, use integration by parts with $u = (2 \log \log 2x)^r$ and $v' = \frac{1}{x^r}$ and get

$$\int_A^B \left( \frac{2 \log \log 2x}{x} \right)^r \, dx = \left[ -\frac{1}{r - 1} \frac{(2 \log \log 2x)^r}{x^{r-1}} \right]_A^B + \int_A^B \frac{r}{r - 1} \cdot \frac{(2 \log \log 2x)^{r-1} \frac{2}{x \log 2x}}{x^{r-1}} \, dx$$

$$\leq \frac{(2 \log \log 2A)^r}{(r - 1) A^{r-1}} + \frac{2r}{r - 1} \underbrace{\int_A^B \frac{(2 \log \log 2x)^{r-1}}{x^r \log 2x} \, dx}_{I_3} \ .$$

For $1 < r \leq 2$, it holds that $(2 \log \log 2x)^{r-1} \leq 2 \log \log 2x \leq \log 2x$. Then,

$$I_3 \leq \int_A^B \frac{1}{x^r} \, dx$$

$$= \frac{1}{r - 1} \left( \frac{1}{A^{r-1}} - \frac{1}{B^{r-1}} \right)$$

$$\leq \frac{1}{(r - 1) A^{r-1}} \ .$$

We have

$$
\begin{aligned}
I_1 &= p^{1-r} \int_A^B \left( \frac{2\log\log 2x}{x} \right)^r dx \\
&\le p^{1-r} \left( \frac{(2\log\log 2A)^r}{(r-1)A^{r-1}} + \frac{2r}{(r-1)^2 A^{r-1}} \right) \\
&= \frac{p \left( \frac{2\log\log 2A}{p} \right)^r}{(r-1)A^{r-1}} + \frac{p^{1-r} \cdot 2r}{(r-1)^2 A^{r-1}} \\
&= \frac{p \left( 2\log\log 2A + b \right)^r}{(r-1)A^{r-1}} + \frac{2rp \left( \frac{2\log\log 2A + b}{2\log\log 2A} \right)^r}{(r-1)^2 A^{r-1}} \\
&\le \frac{p \left( 2\log\log 2A + b \right)^r}{(r-1)A^{r-1}} + \frac{r \left( 2\log\log 2A + b \right)^r}{(r-1)^2 A^{r-1}} \, ,
\end{aligned}
$$

where the last inequality holds since $p \le 1$ and $2\log\log 2A \ge 2$ whenever $A \ge 8$. Finally, we obtain

$$
\begin{aligned}
\int_A^B \left( \frac{2\log\log 2x + b}{x} \right)^r dx &\le I_1 + I_2 \\
&\le \frac{p \left( 2\log\log 2A + b \right)^r}{(r-1)A^{r-1}} + \frac{2r \left( 2\log\log 2A + b \right)^r}{(r-1)^2 A^{r-1}} + \frac{(1-p) \left( 2\log\log 2A + b \right)^r}{(r-1)A^{r-1}} \\
&= \left( \frac{1}{r-1} + \frac{r}{(r-1)^2} \right) \frac{\left( 2\log\log 2A + b \right)^r}{A^{r-1}} \\
&= \frac{2r-1}{(r-1)^2} \cdot \frac{\left( 2\log\log 2A + b \right)^r}{A^{r-1}} \, .
\end{aligned}
$$

*Case 4: $r > 2$.*
Use integration by parts with $u = (2\log\log 2x + b)^r$ and $v' = \frac{1}{x^r}$ and get

$$
\begin{aligned}
\underbrace{\int_A^B \left( \frac{2\log\log 2x + b}{x} \right)^r dx}_{I_4} &= \left[ -\frac{1}{r-1} \cdot \frac{(2\log\log 2x + b)^r}{x^{r-1}} \right]_A^B + \int_A^B \frac{1}{r-1} \cdot \frac{2r (2\log\log 2x + b)^{r-1}}{x^r \log 2x} dx \\
&\le \frac{1}{r-1} \cdot \frac{(2\log\log 2A + b)^r}{A^{r-1}} + \frac{2r}{r-1} \int_A^B \frac{(2\log\log 2x + b)^{r-1}}{x^r \log 2x} dx \\
&\le \frac{1}{r-1} \cdot \frac{(2\log\log 2A + b)^r}{A^{r-1}} + 4 \underbrace{\int_A^B \frac{(2\log\log 2x + b)^{r-1}}{x^r \log 2x} dx}_{I_5} \, .
\end{aligned}
$$

For $x \ge A \ge 8$, it holds that $(2\log\log 2x + b)(\log 2x) \ge (2\log\log 16 + 1)(\log 16) \ge 8$. Then,

$$
\begin{aligned}
I_5 &\le \int_A^B \frac{(2\log\log 2x + b)(\log 2x)}{8} \frac{(2\log\log 2x + b)^{r-1}}{x^r \log 2x} dx \\
&= \frac{1}{8} \int_A^B \frac{(2\log\log 2x + b)^r}{x^r} dx \\
&= \frac{I_4}{8} \, .
\end{aligned}
$$

Therefore we have $I_4 \le \frac{1}{r-1} \cdot \frac{(2\log\log 2A + b)^r}{A^{r-1}} + \frac{I_4}{2}$, which implies $I_4 \le \frac{2}{r-1} \cdot \frac{(2\log\log 2A + b)^r}{A^{r-1}}$. $\quad\square$

### G.5 TIME-UNIFORM CONCENTRATION INEQUALITIES

The following lemma is a special case of Theorem 3 from Garivier (2013). For completeness, we provide the proof adapted to this lemma.

**Lemma 27** (Time-Uniform Azuma inequality). *Let $\{X_t\}_{t=1}^{\infty}$ be a real-valued martingale difference sequence adapted to a filtration $\{\mathcal{F}_t\}_{t=0}^{\infty}$. Assume that $\{X_t\}_{t=1}^{\infty}$ is conditionally $\sigma$-sub-Gaussian, i.e., $\mathbb{E}\left[e^{sX_t} \mid \mathcal{F}_{t-1}\right] \leq e^{\frac{s^2\sigma^2}{2}}$ for all $s \in \mathbb{R}$. Then, it holds that*

$$\mathbb{P}\left(\exists n \in \mathbb{N} : \left|\sum_{t=1}^{n} X_t\right| \geq 2^{\frac{3}{4}}\sigma\sqrt{n\log\frac{7(\log 2n)^2}{\delta}}\right) \leq \delta.$$

*Proof of Lemma 27.* By the union bound, it is sufficient to prove one side of the inequality, namely,

$$\mathbb{P}\left(\exists n \in \mathbb{N} : \sum_{t=1}^{n} X_t \geq 2^{\frac{3}{4}}\sigma\sqrt{n\log\frac{3.5(\log 2n)^2}{\delta}}\right) \leq \delta.$$

Let $t_j = 2^j$ for $j \geq 0$. Partition the set of natural numbers into $I_0, I_1, \ldots$, where $I_j = \{t_j, t_j + 1, \ldots, t_{j+1} - 1\}$. For a fixed positive real number $s_j$, whose values we assign later, define $D_t = \exp\left(s_j X_t - \frac{s_j^2\sigma^2}{2}\right)$. Then, by sub-Gaussianity of $X_t$, we have $\mathbb{E}[D_t \mid \mathcal{F}_{t-1}] \leq 1$. Define $M_n = D_1 D_2 \cdots D_n = \exp\left(s_j\sum_{t=1}^{n} X_t - \frac{s_j^2\sigma^2 n}{2}\right)$, where $M_0 = 1$. Then, $\mathbb{E}[M_n \mid \mathcal{F}_{n-1}] = \mathbb{E}[M_{n-1}D_n \mid \mathcal{F}_{n-1}] \leq M_{n-1}$, therefore $\{M_n\}_{n=0}^{\infty}$ is a super-martingale. By Ville's maximal inequality, we get

$$\mathbb{P}\left(\exists n \in I_j : M_n \geq \frac{1}{\delta}\right) \leq \delta.$$

Note that $M_n \geq \frac{1}{\delta}$ is equivalent to $\sum_{t=1}^{n} X_t \geq \frac{s_j\sigma^2 n}{2} + \frac{1}{s_j}\log\frac{1}{\delta}$. Take $s_j = \frac{1}{\sigma}\sqrt{\frac{\sqrt{2}}{t_j}\log\frac{1}{\delta}}$ and obtain

$$\mathbb{P}\left(\exists n \in I_j : \sum_{t=1}^{n} X_t \geq \sigma\left(\frac{n}{2}\sqrt{\frac{\sqrt{2}}{t_j}} + \sqrt{\frac{t_j}{\sqrt{2}}}\right)\sqrt{\log\frac{1}{\delta}}\right) \leq \delta.$$

For $n \in I_j$, $\frac{n}{2} < t_j \leq n$ holds, therefore $\frac{n}{2}\sqrt{\frac{\sqrt{2}}{t_j}} + \sqrt{\frac{t_j}{\sqrt{2}}} \leq \frac{n}{2}\sqrt{\frac{2\sqrt{2}}{n}} + \sqrt{\frac{n}{\sqrt{2}}} = 2^{\frac{3}{4}}\sqrt{n}$. Furthermore, replace $\delta$ with $\frac{6\delta}{\pi^2(j+1)^2}$ to obtain

$$\mathbb{P}\left(\exists n \in I_j : \sum_{t=1}^{n} X_t \geq 2^{\frac{3}{4}}\sigma\sqrt{n\log\frac{\pi^2(j+1)^2}{6\delta}}\right) \leq \frac{6\delta}{\pi^2(j+1)^2}.$$

From $\frac{\pi^2(j+1)^2}{6} = \frac{\pi^2(\log_2 2t_j)^2}{6} \leq \frac{\pi^2}{6(\log 2)^2}(\log 2t_j)^2 \leq \frac{7}{2}(\log 2n)^2$, we get

$$\mathbb{P}\left(\exists n \in I_j : \sum_{t=1}^{n} X_t \geq 2^{\frac{3}{4}}\sigma\sqrt{n\log\frac{7(\log 2n)^2}{2\delta}}\right) \leq \frac{6\delta}{\pi^2(j+1)^2}.$$

Take the union bound over $j \geq 0$, and by the fact $\sum_{j=0}^{\infty}\frac{1}{(j+1)^2} = \frac{\pi^2}{6}$, we get the desired result.

$$\mathbb{P}\left(\exists n \in \mathbb{N} : \sum_{t=1}^{n} X_t \geq 2^{\frac{3}{4}}\sigma\sqrt{n\log\frac{3.5(\log 2n)^2}{\delta}}\right) \leq \delta.$$

$\square$

Next lemma is a time-uniform version of Theorem 1 in Beygelzimer et al. (2011). We combine the proof of the theorem and a standard super-martingale analysis to obtain a time-uniform inequality.

**Lemma 28** (Time-uniform Freedman's inequality). *Let $\{X_t\}_{t=1}^{\infty}$ be a real-valued martingale difference sequence adapted to a filtration $\{\mathcal{F}_t\}_{t=0}^{\infty}$. Suppose there exists a constant $R > 0$ such that for all $t \geq 1$, $|X_t| \leq R$ holds almost surely. For any constant $\eta \in \left(0, \frac{1}{R}\right]$ and $\delta \in (0, 1]$, it holds that*

$$\mathbb{P}\left(\exists n \in \mathbb{N} : \sum_{t=1}^{n} X_t \geq \eta\sum_{t=1}^{n}\mathbb{E}\left[X_t^2 \mid \mathcal{F}_{t-1}\right] + \frac{1}{\eta}\log\frac{1}{\delta}\right) \leq \delta.$$

*Proof of Lemma 28.* We have $|\eta X_t| \leq 1$ almost surely for all $t \geq 1$. Since $1 + x \leq e^x$ for all $x \in \mathbb{R}$ and $e^x \leq 1 + x + x^2$ for all $x \in [-1, 1]$, it holds that

$$
\begin{aligned}
\mathbb{E}\left[e^{\eta X_t} \mid \mathcal{F}_{t-1}\right] &\leq \mathbb{E}\left[1 + \eta X_t + \eta^2 X_t^2 \mid \mathcal{F}_{t-1}\right] \\
&= 1 + \eta^2 \mathbb{E}\left[X_t^2 \mid \mathcal{F}_{t-1}\right] \\
&\leq e^{\eta^2 \mathbb{E}\left[X_t^2 \mid \mathcal{F}_{t-1}\right]}.
\end{aligned}
\tag{54}
$$

Define $D_t := \exp\left(\eta X_t - \eta^2 \mathbb{E}\left[X_t^2 \mid \mathcal{F}_{t-1}\right]\right)$. Eq. (54) implies $\mathbb{E}\left[D_t \mid \mathcal{F}_{t-1}\right] \leq 1$. Define $M_n := D_1 D_2 \cdots D_n = \exp\left(\eta \sum_{t=1}^n X_t - \eta^2 \sum_{t=1}^n \mathbb{E}\left[X_t^2 \mid \mathcal{F}_{t-1}\right]\right)$, where $M_0 = 1$. Then, $\mathbb{E}\left[M_n \mid \mathcal{F}_{n-1}\right] = \mathbb{E}\left[M_{n-1}D_n \mid \mathcal{F}_{n-1}\right] \leq M_{n-1}$, therefore $\{M_n\}_{n=0}^\infty$ is a super-martingale. By Ville's maximal inequality, we obtain

$$
\mathbb{P}\left(\exists n \in \mathbb{N}: M_n \geq \frac{1}{\delta}\right) \leq \frac{\mathbb{E}[M_0]}{1/\delta} = \delta.
$$

The proof is complete by noting that $M_n = \exp\left(\eta \sum_{t=1}^n X_t - \eta^2 \sum_{t=1}^n \mathbb{E}\left[X_t^2 \mid \mathcal{F}_{t-1}\right]\right) \geq \frac{1}{\delta}$ is equivalent to $\sum_{t=1}^n X_t \geq \eta \sum_{t=1}^n \mathbb{E}\left[X_t^2 \mid \mathcal{F}_{t-1}\right] + \frac{1}{\eta} \log \frac{1}{\delta}$. $\square$

Next lemma is a widely known application of Lemma 28.

**Lemma 29.** *Let $\{Y_t\}_{t=1}^\infty$ be a sequence of real-valued random variables adapted to a filtration $\{\mathcal{F}_t\}_{t=0}^\infty$. Suppose $0 \leq Y_t \leq 1$ holds almost surely for all $t \geq 1$. For any $\delta \in (0, 1]$, it holds that*

$$
\mathbb{P}\left(\exists n \in \mathbb{N}: \sum_{t=1}^n Y_t \geq \frac{5}{4} \sum_{t=1}^n \mathbb{E}\left[Y_t \mid \mathcal{F}_{t-1}\right] + 4 \log \frac{1}{\delta}\right) \leq \delta.
\tag{55}
$$

*Proof of Lemma 29.* Let $X_t = Y_t - \mathbb{E}\left[Y_t \mid \mathcal{F}_{t-1}\right]$. Then, $\{X_t\}_{t=1}^\infty$ is a martingale difference sequence adapted to $\{\mathcal{F}_t\}_{t=0}^\infty$ with $|X_t| \leq 1$ almost surely. Apply Lemma 28 with $\eta = \frac{1}{4}$ and obtain

$$
\mathbb{P}\left(\exists n \in \mathbb{N}: \sum_{t=1}^n X_t \geq \frac{1}{4} \sum_{t=1}^n \mathbb{E}\left[X_t^2 \mid \mathcal{F}_{t-1}\right] + 4 \log \frac{1}{\delta}\right) \leq \delta.
\tag{56}
$$

We have

$$
\begin{aligned}
\mathbb{E}\left[X_t^2 \mid \mathcal{F}_{t-1}\right] &= \mathbb{E}\left[(Y_t - \mathbb{E}\left[Y_t \mid \mathcal{F}_{t-1}\right])^2 \mid \mathcal{F}_{t-1}\right] \\
&\leq \mathbb{E}\left[Y_t^2 \mid \mathcal{F}_{t-1}\right] \\
&\leq \mathbb{E}\left[Y_t \mid \mathcal{F}_{t-1}\right],
\end{aligned}
$$

where the last inequality holds by $0 \leq Y_t \leq 1$. Then, Eq. (56) implies

$$
\mathbb{P}\left(\exists n \in \mathbb{N}: \sum_{t=1}^n Y_t - \sum_{t=1}^n \mathbb{E}\left[Y_t \mid \mathcal{F}_{t-1}\right] \geq \frac{1}{4} \sum_{t=1}^n \mathbb{E}\left[Y_t \mid \mathcal{F}_{t-1}\right] + 4 \log \frac{1}{\delta}\right) \leq \delta,
$$

which is equivalent to the desired result of Eq. (55). $\square$

## H  AUXILIARY LEMMAS

**Lemma 30** (Corollary 6.8 in (Bühlmann & Van De Geer, 2011)). *Let $\boldsymbol{\Sigma}_0, \boldsymbol{\Sigma}_1 \in \mathbb{R}^{d \times d}$. Suppose that the compatibility constant of $\boldsymbol{\Sigma}_0$ over the index set $S$ with cardinality $s = |S|$ is positive, i.e., $\phi^2(\boldsymbol{\Sigma}_0, S) > 0$. If $\|\boldsymbol{\Sigma}_0 - \boldsymbol{\Sigma}_1\|_\infty \leq \frac{\phi^2(\boldsymbol{\Sigma}_0, S)}{32 s_0}$, then $\phi^2(\boldsymbol{\Sigma}_1, S) \geq \phi^2(\boldsymbol{\Sigma}_0, S_0)/2$.*

**Lemma 31** (Transfer principle, Lemma 5.1 in (Oliveira, 2016)). *Suppose $\hat{\boldsymbol{\Sigma}}$ and $\bar{\boldsymbol{\Sigma}}$ are $d \times d$ matrices with non-negative diagonal entries. Assume $\eta \in (0, 1)$ and $m \in [d]$ are such that*

$$
\forall \mathbf{v} \in \mathbb{R}^d \text{with } \|\mathbf{v}\|_0 \leq m, \mathbf{v}^\top \hat{\boldsymbol{\Sigma}} \mathbf{v} \geq (1 - \eta)\mathbf{v}^\top \bar{\boldsymbol{\Sigma}} \mathbf{v}.
$$

*Assume $\mathbf{D}$ is a diagonal matrix whose elements are non-negative and satisfies $\mathbf{D}_{jj} \geq \hat{\boldsymbol{\Sigma}}_{jj} - (1 - \eta)\bar{\boldsymbol{\Sigma}}_{jj}$. Then,*

$$
\forall \mathbf{v} \in \mathbb{R}^d, \|\mathbf{v}\|_0 \leq m, \mathbf{v}^\top \hat{\boldsymbol{\Sigma}} \mathbf{v} \geq (1 - \eta)\mathbf{v}^\top \bar{\boldsymbol{\Sigma}} \mathbf{v} - \frac{\|\mathbf{D}\mathbf{v}\|_1^2}{m - 1}.
$$

# I    NUMERICAL EXPERIMENT DETAILS

Our numerical experiment in Section 4 measures the performance of various sparse linear bandit algorithms under two different distributions of context feature vectors. For both experiments, we set $d = 100$, $T = 2000$, and $\eta_t \sim \mathcal{N}(0, 0.25)$. For given $s_0$, we sample $S_0$ uniformly from all subsets of $[d]$ with size $s_0$, then sample $\boldsymbol{\beta}^*_{S_0}$ uniformly from a $s_0$-dimensional unit sphere. We tune the hyper-parameters of each algorithm to achieve their best performance.

**Experiment 1. (Figure 2a)** Following the experiments in Kim & Paik (2019); Oh et al. (2021); Chakraborty et al. (2023), for each $i \in [d]$, the $i$-th components of the $K$ feature vectors are sampled from $\mathcal{N}(\mathbf{0}_K, \mathbf{V})$, where $\mathbf{V}_{ii} = 1$ for $1 \leq i \leq K$ and $\mathbf{V}_{ij} = 0.7$ for $1 \leq i, j \leq K$ with $i \neq j$. In this way, the arms have high correlation across each other. Note that assumptions of Oh et al. (2021); Ariu et al. (2022); Li et al. (2021); Chakraborty et al. (2023) hold in this setting. By Theorem 3, `FS-WLasso` may take $M_0 = 0$. To distinguish our algorithm from `SA Lasso BANDIT`, we set $M_0 = 10$ and $w = 1$.

**Experiment 2. (Figure 2b)** We evaluate our algorithms for a context distribution that does not satisfy the strong assumptions employed in the previous Lasso bandit literature (Oh et al., 2021; Ariu et al., 2022; Li et al., 2021; Chakraborty et al., 2023). We sample $K - 1$ vectors for sub-optimal arms from $\mathcal{N}(\mathbf{0}_d, \mathbf{I}_d)$ and fix them for all rounds. For each $t \in [T]$, we sample the feature for the optimal arm from $\mathcal{N}(\mathbf{0}_d, \mathbf{I}_d)$. Then, we appropriately assign the expected rewards of the features by adjusting their $\boldsymbol{\beta}^*$-components. Specifically, for a sampled vector $\mathbf{x}$ and a desired value $c$, we set $\mathbf{x}' = \mathbf{x} + \frac{c - \mathbf{x}^\top \boldsymbol{\beta}^*}{\|\boldsymbol{\beta}^*\|_2^2} \boldsymbol{\beta}^*$ so that we have $\mathbf{x}'^\top \boldsymbol{\beta}^* = c$. We set the fixed sub-optimal arms to have expected rewards of $0.1, 0.2, \ldots, 0.9$, and sample the expected reward of the optimal arm from $\mathrm{Unif}(0.9, 1)$. To prevent the theoretical Gram matrix from becoming positive-definite or having a positive sparse eigenvalue, we sample five indices from $S_0^c$ in advance and fix their values at $5$ for all arms and rounds.

All experiments were held in a computing cluster with twenty Intel(R) Xeon(R) Silver 4210R CPUs and 187 GB of RAM.

