# OpenReview forum: "Lasso Bandit with Compatibility Condition on Optimal Arm"
_ICLR.cc/2025/Conference — ICLR 2025 Poster_

### Official Review · Reviewer_ynf4 · 2024-11-02

**Soundness:** 3
**Presentation:** 1
**Contribution:** 2
**Rating:** 5
**Confidence:** 4

**Summary:**

This paper proposes a weaker sufficient condition for high dimensional bandit algorithm. Under such weaker condition, the paper also proves regret bounds for bandit algorithms.

**Strengths:**

1. The authors provide a lot of intuitions for the algorithm design.
2. I think in general the weaker condition, and the algorithm induced, would be a good addition to the literature.

**Weaknesses:**

1. The paper needs to be restructured. If the key idea of this paper is the weaker condition proposed, then the main section and main results should be about this condition. Therefore, Appendix B should be the main part of the paper instead of just in the appendix.
Besides, the novelty claimed, such as the cyclic induction (Lemma 6?), should be moved to the main text as well. It's hard to read and understand the novelty for the current version.
2. Assumption 3 is claimed to be "strictly weaker" than the existing conditions. Then, some counterexamples should be provided to show that Assumption 3 holds but not any of the existing conditions.

I'd be willing to update my rating if the paper can be structured and easy to understand.

**Questions:**

N/A

---

> ### Author Response · Authors · 2024-11-16
>
> We sincerely thank you for your time and effort in reviewing our paper and for your feedback. We are eager to make our paper more accessible and presentable to the readers. We will do our utmost to improve the manuscript accordingly.
>
> - - -
> [W1] __Presentation__
>
> Please note that the main contribution of our paper is to propose a weaker (hence, more general) condition for achieving poly-logarithmic regret in the sparse linear contextual bandit problem.  To support this main contribution, two points must be demonstrated:
> 1) The newly proposed condition is weaker than the previous proposed conditions, and
> 2) There exists an algorithm that can achieve poly-logarithmic regret under the newly proposed condition.
>
> It is not sufficient to argue our main contribution by merely showing that our proposed assumption (Assumption 3) is weaker than those in previous lasso bandit studies. Therefore, we first introduced the new assumption, explained how it is weaker than prior assumptions, and then presented an algorithm and regret bound based on this assumption. To include the main contribution and the key observations within the limited page count, we have provided a detailed explanation of the technical aspects in the appendix.
>
> To further improve the clarity of our contribution, we will highlight the mildness of Assumption 3 by incorporating important results from Appendix B into the main paper.
> However, given that other reviewers have provided positive feedback on the current presentation,
> we kindly ask for your understanding that completely rearranging the content at this stage might risk affecting other reviewers’ evaluations of the paper's structure and presentation.
> Please note that this is not a matter of willingness. We are fully committed to addressing any feedback that can help improve our paper.
> If all reviewers agree, we would be more than happy to consider making the suggested changes. In the meantime, we are more than willing to address any further questions or engage in additional discussions about our results during the rebuttal period.
>
> [W2] __Counterexample__
>
> As explained in main contribution bullet points \& Figure 1, the case where the context feature vectors of sub-optimal arms are fixed and only the feature vector of the optimal arm has randomness serves as a counter-example.

---

> ### Author Response · Authors · 2024-11-25
>
> We sincerely appreciate the time and effort the reviewer has put to reviewing our paper. We hope we have adequately addressed the reviewer’s request and provided the necessary clarifications.
>
> As highlighted in the updated revised manuscript, we have incorporated the reviewer’s feedback to more clearly convey the main contribution by newly including a statement to Theorem 1 regarding the mildness of Assumption 3. Additionally, in the proof sketch (Section 3.3), we have included more detailed explanations to highlight the technical challenges unique to our setting compared to previous methodologies and the technical novelties we introduced to address them.
> We would be more than happy to incorporate further changes in the presentation if the reviewer deems it necessary, provided that such changes do not interfere with the overall structure and clarity of the paper.
>
> With our recently posted comment [("Our Key Contributions")](https://openreview.net/forum?id=f3jySJpEFT&noteId=FJzpaEgeba) in mind, we would like to know if the reviewer has any additional questions or comments that we can address. If so, we would be more than happy to discuss and respond. On the other hand, if our responses have sufficiently addressed the reviewer’s request, we sincerely hope that the reviewer will reconsider their assessment in light of the strengths and key contributions of our work.
> Thank you for your consideration.

---

### Official Review · Reviewer_WnKD · 2024-11-04

**Soundness:** 3
**Presentation:** 3
**Contribution:** 3
**Rating:** 6
**Confidence:** 4

**Summary:**

The author devised an algorithm called FS-WLasso, which is based on a greedy algorithm with the addition of forced exploration. According to the authors, this algorithm demonstrates polylogarithmic regret even under the weakest assumptions in the context of sparse linear bandits in a stochastic environment. They also demonstrated the algorithm’s practical effectiveness through simulations.

**Strengths:**

- Soundness: In theoretical research, relaxing assumptions is undoubtedly meaningful. This work is particularly valuable in the sense that the authors united all various assumptions in sparse contextual linear bandits, and allows researchers to focus on a single general condition.

- Clarity: The authors clearly illustrated the relationship between their results and previous work using diagrams, and, through an extensive literature review, they included comparisons in multi-parameter setups to highlight the versatility of their setting (Remark 1). Additionally, their meticulous preparation is evident in the various remarks and supplementary materials to address anticipated questions (even the fixed arm set case) and to emphasize their significance/novelty.

- They also introduced a novel proof technique using induction - like a domino, they've constructed an interesting proof scheme to maintain their compatibility condition.

**Weaknesses:**

- Their algorithm works on a more general 'environment', but to guarantee their performance the learning agent requires additional knowledge, such as the sparsity level $s_0$ or the gap $\Delta_0$. As they have mentioned, their result is not sparsity-agnostic. Even though they've mentioned in Appendix D, it is still true that they rely on the sparsity parameter $s_0$ and many other parameters, and I personally think this parameter is somewhat less general info than the noise level $\sigma$ or the norm bound of the context vector $x_{max}$.

**Questions:**

- Why the exploration in Algorithm 1, line 5 should be a uniform one? I think the most intuitive exploration to guarantee the minimum eigenvalue is $a = argmax_{x\in x_t} \|\|x\|\|_{V_t^{-1}}$.

- I don't understand why the fact that $M_0$ depends on multiple variables implies that it is a tunable parameter. Isn't it a worse situation, that a core parameter depends on multiple parameters (even $\Delta_*$ and $\phi_*$)? To compare it with Oh et al., 2021, it feels like 'an algorithm which is easy to run but should satisfy some strong assumptions on the environment' vs 'an algorithm which works on a weaker assumption but requires more information about the environment'. Some readers might think of this as an incremental improvement.

---

> ### Author Response · Authors · 2024-11-16
>
> Thank you for taking the time to review our paper and for your thoughtful and valuable feedback. We deeply appreciate your recognition of our work and the constructive comments you have provided. Below, we address each of your comments and questions in detail:
> - - -
> [W1] __Less general parameter__
>
> We are more than happy to address your comment regarding the reliance on the sparsity parameter $s_0$ and other parameters.
>
> It is critical to note that existing works also incorporate additional assumptions or conditions, even with sparsity awareness. For example, Li et al. (2021) incorporate additional diversity conditions despite knowing sparsity, while Bastani \& Bayati (2020) and Wang et al. (2018) impose specific optimality criteria for context distributions under sparsity awareness. In this context, we believe that our approach, which alleviates stringent diversity assumptions on context distributions and introduces the weakest known conditions to achieve poly-logarithmic regret, provides new insights and significant value to the bandit community.
>
> Additionally, with all due respect, we are not entirely certain that sparsity $s_0$ is inherently less general information than the noise level $ \sigma$ or even the norm bound $x_{\text{max}}$ in advance. For instance, practitioners in some applications might argue that they have prior knowledge of an upper bound on the number of relevant features (i.e., an upper bound on $s_0$; please note that our algorithm can operate on an upper bound on $s_0$ instead of exact $s_0$ if such information is available) but lack precise information about the noise level. While we are not claiming that $s_0$ is more general information, we do observe that rigorous discussions on whether one parameter is inherently more general than another have not been extensively addressed in the prior literature and depends on applications.
> Hence, at least in theoretical studies, we believe that these parameters should be treated without bias toward their relative generality.
> Thank you for providing us the opportunity to clarify this point.
>
> [Q1] __Uniform sampling__
>
> Thank you for the insightful question.
> Since we do not use the notation $\mathbf{V}\_t$ in our manuscript, we believe that the exploration method the reviewer mentioned is $a\_t = \text{argmax}\_{k \in [K]} || \mathbf{x}\_{t,k} ||\_{\mathbf{V}^{-1}\_t}$ where $\mathbf{V}\_t = \sum\_{i=1}^{t-1} \mathbf{x}\_{i, a\_i} \mathbf{x}\_{i, a\_i}^\top$.
> First of all, we would like to remind you that what we aim to guarantee through the forced sampling stage is a positive compatibility constant for the empirical Gram matrix (Eq 21), NOT a positive minimum eigenvalue.
> Therefore, any exploration strategy other than uniform sampling that can ensure positive compatibility constant of the empirical Gram matrix could potentially be plugged into our algorithm.
> What we show is that the simplest exploration strategy possible is sufficient, and of course it is not the case that it "must" be a uniform sampling.
>
> Now, we explain why the method proposed by the reviewer is not suitable for ensuring this guarantee.
>
> First, we do not assume positive definiteness for the expected Gram matrix on the averaged arm $\boldsymbol{\Sigma}$. Thus, even if sampling is conducted as the proposed method, the minimum eigenvalue of the empirical Gram matrix could still be zero in the worst case.
>
> Second, the proposed method is not statistically efficient in the sparse linear bandit problem setting.
> In other words, even if sampling is conducted according to the proposed method, the number of samples required to ensure a positive minimum eigenvalue of the empirical Gram matrix will depend on $d$. This would result in regret bound that depends on a polynomial term of $d$. Such dependence becomes even more catastrophic, especially in high-dimensional setting ($d \gg T$).
>
> Finally, the proposed method is computationally inefficient. Computing $\mathbf{V}_t^{-1}$ requires a computational complexity of $\mathcal{O}(d^2)$, whereas uniform random sampling among $K$ $d$-dimensional vectors requires a computational complexity of $\mathcal{O}(1)$.

---

> ### Author Response · Authors · 2024-11-16
>
> [Q2] __Dependence of $M_0$__
>
> - __Tunable parameter:__ What we convey in Remark 5 and Appendix D is that whether an algorithmic parameter relies on one unknown variable or many unknown variables, the practical use of the algorithm does not change since that ONE parameter would have to be tuned anyway. It is true that $M_0$ theoretically depends on multiple problem-dependent variables, but it does not make the situation worse just because the number of related variables is more than one.
> Furthermore, even if a parameter is fully specified by theory, it is often tuned in practice for better empirical performance.
> __We believe the reviewer recognizes that such tunable parameters are not unique to our work but are also present in almost all Lasso bandit and parametric bandit algorithms__ (Abbasi-Yadkori et al., 2011; Li et al., 2017; Kim & Paik, 2019; Li et al., 2021; Oh et al., 2021; Wang et al., 2018; Bastani & Bayati, 2020; Ariu et al., 2022;  Chakraborty et al., 2023;).
> Additionally, we would like to highlight that our algorithm does not appear to be sensitive to the choice of $M_0$ in numerical experiments, as shown in Figure 4.
>
> - __Comparison to Oh et al. (2021):__ We appreciate your comparison of our work to Oh et al. (2021) and acknowledge some of the points raised. However, we respectfully disagree with the possible oversimplification of this comparison, as we believe it requires a more nuanced approach. Furthermore, we strongly contend that our results represent a significant advancement rather than an incremental improvement for the following reasons.
>
>     Regarding Oh et al. (2021), we emphasize that the diversity assumptions on the context distribution are only known to be sufficient conditions for the success of the _greedy action selection algorithm_. As the reviewer also knows, greedy algorithms do not require any exploration strategies such as forced sampling or UCB-type exploration. Therefore, if the diversity assumptions allow a greedy algorithm to succeed, the algorithm does not need to be specified with parameters required for exploration, such as $s_0$. In other words, greedy algorithms do not require any hyperparameters to tune their level of exploration.
>
>     On the other hand, there has been no rigorous discussion on whether such diversity assumptions are necessary for the success of the greedy algorithm or even for other explorative algorithms using Lasso. Even after Oh et al. (2021), the subsequent works propsing explorative Lasso bandit algorithms (e.g., UCB: Li et al., 2021; TS: Chakraborty et al., 2023) that rely on diversity assumptions have continued to be proposed.
>
>     Thus, we strongly believe that demonstrating an explorative Lasso bandit algorithm capable of achieving poly-logarithmic regret _without diversity assumptions_ is NOT an incremental improvement. These contributions are made possible not only through the introduction of new, more relaxed conditions and enhanced regret analysis but also by presenting a practical, easy-to-implement algorithm enabled by our new theoretical results. We assert with confidence that our technical contributions are distinct and significant compared to other works in the Lasso bandit literature. If further clarification is needed, we would be happy to provide additional details.

---

> > ### Comment · Reviewer_WnKD · 2024-11-27
> >
> > Thanks for your sincere clarification.
> >
> > - About $M_0$, yes it is true that many practitioners tune parameters when they use bandit algorithms (and I understand the authors sell their paper that only one hyperparameter $M_0$ needs a tuning). Still, I believe the main value of this paper lies in the theoretical improvement, and the fact that the learner knows $\phi_*$ and $\Delta_*$ are quite strong assumptions. Do other sparse linear bandit works (with greedy approach + lasso as you did) also need this assumption in advance? I just wish to clarify this one.
> >
> > - Also, for the uniform sampling, I wished to get some information on mathematical understandings about $\arg \max_{a \in \mathcal{A}} \|\|a\|\|_{V_t^{-1}}$ based sampling if you have thought about it, like what kind of minimum eigenvalue (or compatibility constant) we can expect. However, it is still true that uniform sampling is computationally much simpler and easier to analyze. (I also know that the main objective of your initial search is for compatibility constant as you use Lasso) So I agree with your direction.

---

> ### Author Response · Authors · 2024-11-25
>
> We truly appreciate the reviewer's valuable feedback and comments. We hope we have addressed feedback/questions and provided the needed clarification in our responses. With our recently posted comment [("Our Key Contributions")](https://openreview.net/forum?id=f3jySJpEFT&noteId=FJzpaEgeba) in mind, we would like to know if there are any additional questions or comments that we can address. We would be more than happy to address them if there are any.
> If our responses have sufficiently addressed the reviewer's comments, we kindly and respectfully ask the reviewer to reconsider their assessment in light of the strengths they have highlighted, as well as the key contributions outlined in our recently posted comment. We believe these points further underscore the significance and impact of our work. Thank you again for your time and consideration.

---

> ### Author Response · Authors · 2024-11-30
>
> Thank you so much for your response to our comments, and we are pleased to have the opportunity to address your questions.
> - - -
>
> > Do other sparse linear bandit works (with greedy approach + lasso as you did) also need this assumption in advance?
>
> In other Lasso bandit literature, the algorithmic parameters also depend on assumption parameters to guarantee the theoretical regret bounds.
> We introduce a few examples here.
> For fair comparison, we focus on works that achieve $\mathcal{O}(\text{poly} \log dT)$ regret under 1-margin condition.
> Since each paper uses different notations, we explain them using the notations adopted in each respective paper.
>
> - Li et al. (2021): initial regularization parameter $\lambda_0 = \mathcal{O}(\sigma x_{\text{max}})$, initial diameter $\tau_0 = \mathcal{O}(s_0 \sigma x_{\text{max}} \phi_0^{-2})$,
> where $\phi_0^2$ is the compatibility constant related to the expected Gram matrix on the averaged arm.
>
> - Ariu et al. (2022): initial regularization parameter $\lambda_0 = \mathcal{O}(\sigma s_A \sqrt{c})$, where $s_A$ is the maximum feature norm and the constant $c$ depends on $\nu, C_b, s_0, \phi_0^2, s_A, \theta_{\text{min}}$ (Please refer to the proof of Theorem 5.2 in their work).
>
> - Chakraborty et al. (2023) : $\sigma$ and $x_{\text{max}}$ are used for posterior computation.
>
> - Wang et al. (2018), Bastani & Bayati (2020):
>
>     initial regularization parameters $\lambda_1 = \mathcal{O}( (\phi_0^2 p_* h) / (s_0 x_{\text{max}})), \lambda_{2, 0} = \mathcal{O}( \sigma x_{\text{max}} p_*^{-1/2} )$,  sampling period parameter $q = \mathcal{O}(\sigma^2 x_{\text{max}}^4  \log d / (\phi_0^4 p_*^2 h^2))$.
> Note that the parameters $h$ and $p_*$ in Bastani & Bayati (2020) play the role of $\Delta_*$ in our work.
>
> We would also like to emphasize that the reason why some previous Lasso bandit works (e.g., Oh et al. 2021) could remain independent of other instance-dependent constants is that the stronger diversity assumptions on the context distribution ensured automatic exploration of the feature space. This allowed algorithms without explicit exploration to avoid identifying the specific phase at which the empirical Gram matrix's compatibility condition is satisfied.
> As shown in Theorem 3, under an additional diversity assumption on the context distribution—comparable in strength at worst to those in the aforementioned studies—our algorithm also does not require pinpointing such a phase. Consequently, our algorithmic parameters depend only on $\sigma$ and $x_{\text{max}}$, which are parameters required by all previous Lasso bandit algorithms.
> We hope this answers your question. If you have any remaining questions, please feel free to let us know.
>
> - - -
> > for the uniform sampling
>
> We are very glad that our previous answers addressed your question well!
>
> - - -
>
> Thank you again for your questions.
> If you believe that our responses have sufficiently addressed your questions, we kindly ask the reviewer to reflect an updated assessment of our work, considering our contributions.

---

### Official Review · Reviewer_Ltvm · 2024-11-04

**Soundness:** 3
**Presentation:** 3
**Contribution:** 3
**Rating:** 8
**Confidence:** 3

**Summary:**

The paper contributes to the high-dimensional bandits literature by relaxing the compatibility conditions required for analysis. Specifically, it reduces the need for compatibility conditions on all arms to only the optimal arm, thereby enhancing the generality of the results.

**Strengths:**

By weakening the compatibility condition from all arms to only the optimal arm and relaxing all other diversification requirement in other high-dimensional bandits papers, this paper makes a solid contribution that broadens the applicability of high-dimensional bandit algorithms. Figure 1 and Table 1 clearly sketched the comparison against existing literature.

**Weaknesses:**

- While the paper criticizes other studies for unverifiable conditions, it does not clearly explain how the compatibility condition on the optimal arm (Assumption 3) is more verifiable in practice. This lack of clarity weakens the argument against existing methods.

- The suggestion to treat the forced sampling iteration number  $M_0$  as a tuning parameter could also be applied to other methods where assumptions are stronger and cannot be easily verified as well. This weakens the significance of the advantage in assuming compatibility for only the optimal arm and does not effectively address the issue of unverifiable assumptions.

- The proof sketch in Section 3.3 lacks sufficient emphasis on the techniques used to overcome the challenges posed by the weakened compatibility assumption. Without a clear explanation of how the standard compatibility condition is avoided, readers may find it difficult to understand the paper’s methodological contributions.

**Questions:**

See weakenesses.

---

> ### Author Response · Authors · 2024-11-16
>
> Thank you for taking the time to review our paper and for your thoughtful and valuable feedback. We deeply appreciate your recognition of our work and the constructive comments you have provided. Below, we address each of your comments and questions in detail:
> - - -
> [W1] __Verifiability__
>
> We thank the reviewer for providing us with the opportunity to clarify this point.
> We would like to respectfully clarify that our primary goal is not to propose a verifiable condition. It is well-known in both offline and online high-dimensional statistics---although perhaps not explicitly documented---that compatibility conditions of all kinds in the literature, including ours (and similar regularity conditions such as restricted eigenvalue conditions), are generally not verifiable in polynomial time.
> Given this inherent unverifiability, the __strength of our work lies in proposing a weaker (and thus more general) condition__ under which theoretical guarantees still hold. By doing so, we contribute to advancing the field by demonstrating that an algorithm relying on such weaker assumptions can still achieve favorable regret guarantees.
>
> In practical terms, because it may not be feasible to verify whether any algorithm's theoretical guarantees apply to a specific instance, the fact that __our paper establishes that whenever existing algorithms are efficient, our algorithm is also efficient__ becomes an important contribution. More importantly, __our algorithm broadens the scope of problem instances where efficiency can be achieved.__ Specifically, we demonstrate that _our algorithm remains efficient even in regimes where existing algorithms fail,_ making it strictly more robust and practical. We believe this clarification highlights the significance of our contribution and underscores the meaningful advance we bring to the Lasso bandit literature.
>
> [W2] __Tuning parameter__
>
> We are glad to address this comment. Treating the algorithmic parameters in previous Lasso bandit algorithms as tunable does not eliminate the dependence on stronger context distribution assumptions. Specifically, even with the introduction of tunable parameters, one cannot eliminate the stronger diversity assumptions required in prior works (e.g., Oh et al., 2021). Additionally, existing algorithms already incorporate tunable parameters (e.g., Li et al., 2021; Chakraborty et al., 2023), yet still rely on such stronger diversity assumptions.
>
> As mentioned in the main paper, stronger diversity assumptions on the context distribution allow the reward estimation error to decrease even when data is obtained through a greedy policy or, in some cases, any policy.
> However, when the context distributions do not satisfy such diversity assumptions, the algorithms proposed under these assumptions can critically compromise regret performance.
> This is because, without the diversity assumptions, a greedy policy or other explorative policies from the previous literature no longer ensure the compatibility condition on the empirical Gram matrix. _What is even more problematic is that there are no resources for adjustment via tuning algorithmic parameters in such cases._
>
> In contrast, our proposed algorithm, which relies on the compatibility condition on the optimal arm, _achieves a poly-logarithmic regret bound without requiring additional diversity assumptions on the context distribution_.
> We believe this advantage is significant, as it provides a new perspective that was not previously explored in the literature.

---

> ### Author Response · Authors · 2024-11-16
>
> [W3] __Proof sketch emphasizing techniques overcoming challenges__
>
> Thank you for your suggestion. We are more than happy to elaborate on the techniques used to address the challenges posed by the weakened compatibility assumption. Due to space limitations, the proof sketch in Section 3.3 could not fully detail the techniques overcoming these key challenges, but we will ensure this is addressed in the revision.
>
> First, we assume that by the "standard compatibility condition," you are referring to the compatibility condition on the averaged arm used in previous Lasso bandit literature (Kim & Paik, 2019; Oh et al., 2021; Ariu et al., 2022), and by the "weakened compatibility condition," you are referring to our proposed compatibility condition on the optimal arm. For clarification on the relationship between these two conditions, please see Remark 4.
>
> Now, let us elaborate on the technical challenges posed by our compatibility condition on the optimal arm, which required a novel analytical approach.
> Under Assumption 3, if the optimal arm has been selected sufficiently many times up to time $t$, it ensures the compatibility constant of the empirical Gram matrix is $\Omega(\phi^2_* t)$. Then, the Lasso estimation error at $t$ can be controlled via the oracle inequality for the weighted squared Lasso estimator (Lemma 19). A well-estimated estimator, in turn, leads to the selection of the optimal arm at the next time step. This observation highlights the cyclic structure between the estimation error and the selection of optimal arms.
>
> We analyze the cyclic structure of such good events using a novel mathematical induction argument (Proposition 1 in Appendix.E.1). In the proof of Proposition 1, Lemma 7 and Lemma 8 constitute the mathematical induction argument by demonstrating that a small number of sub-optimal selections at one time step implies a well-estimated estimator and a small number of sub-optimal selections at the next time step. Finally, we bound the cumulative regret by combining the bounds for the estimation error and the number of sub-optimal selections.
>
> On the other hand, we emphasize that this type of proof technique has not been applied in previous Lasso bandit literature. Prior works do not address this cyclic structure; instead, they rely on diversity assumptions on the context distribution, which ensure that samples obtained by the agent’s policy (greedy or otherwise) automatically explore the feature space, resulting in a positive compatibility constant for the empirical Gram matrix.
>
> Additionally, there is another challenge in analyzing the cyclic structure. Due to the stochastic nature of the problem, the algorithm suffers a small probability of failing to propagate the good event at every round. To bound the probability of failure to be less than $\delta$, which is challenging because it is accumulated across every time step of the induction, we carefully construct high-probability events that enables the mathematical induction, which are analyzed in Appendix.E.4.
>
> Again, we would be glad to include this discussion in the revision. We assert with confidence that our technical contributions are more distinct and significant than the other Lasso bandit literature. We believe that this new analytical techniques will significantly influence future research in this field.
> We will include a detailed discussion of these techniques and their implications in the revised manuscript to better communicate our methodological contributions. Thank you for highlighting this opportunity for improvement.

---

> > ### Comment · Reviewer_Ltvm · 2024-11-26
> >
> > Thank you very much for your response. I went through the other responses and reviews and would like to maintain my positive rating of the paper.

---

> > > ### Author Response · Authors · 2024-11-27
> > >
> > > We again sincerely appreciate your supportive feedback and recognition of the contributions of our work!

---

### Author Response · Authors · 2024-11-25
**Our Key Contributions**

We sincerely thank you for taking the time to review our paper and for recognizing our work. We hope our responses have adequately addressed your questions/feedback. If you have any further questions or comments, we would be more than happy to address them. In the sparse linear bandit problem, various assumptions and methods have been introduced to achieve sharp regret, but many of these assumptions are needed solely for technical purposes. A rigorous discussion on the relative strength of one assumption over another has not been provided before. **This paper offers the first rigorous discussion on the relative strength of assumptions, introduces a strictly relaxed sufficient condition needed to achieve sharp regret, presents a practical algorithm, and analyzes the proposed algorithms using a novel mathematical induction-based technique.** We hold our work in high regard and are immensely proud of the contributions we've made through our research. The depth of our analysis, the rigor of our methodology, and the potential impact of our findings are aspects of our work that we believe stand on their own merits. We strongly believe that our submission well surpasses the threshold of sufficient contribution. In the meantime, we would like to highlight some of the principal points.

**Weakest known condition for achieving poly-logarithmic regret in Lasso bandits:** In this paper, we primarily focus on alleviating strong stochastic assumptions imposed on context distribution in the previous Lasso bandit literature. Existing Lasso bandit literature necessitate stronger assumptions on the context distribution (e.g., relaxed symmetry \& balanced covariance or anti-concentration), which importantly, are non-verifiable in practical scenarios. However, **we show that the compatibility condition on the optimal arm is sufficient to achieve poly-logarithmic regret under the margin condition**, and **demonstrate that our assumption is strictly weaker than those used in other Lasso bandit literature** (Please refer to Figure 1 \& Appendix B). We genuinely believe that this paper stands on its own merit by establishing the weakest known condition for achieving poly-logarithmic regret in Lasso bandits, paving the way for future advancements in this area of research!

**Practically Applicable Algorithm:** Previous forced-sampling-based algorithm (Bastani \& Bayati, 2020) maintains two estimators and operates in two steps when choosing actions, whereas our algorithm only needs to select actions greedily based on the Lasso estimator, making our algorithm simple to implement. In addition, compared to the ESTC algorithm (Hao et al., 2020b), which is restricted to using only the samples gathered during the exploration period, our algorithm is designed to leverage data from both the exploration period (forced-sampling stage) and the adaptive action selection period (greedy selection stage) for estimation purpose. **Therefore, our algorithm demonstrates superior statistical performance, achieving lower regret (and thus higher reward) by fully utilizing all accessible data.** Also, our algorithm does NOT require strong context diversity assumption, which, importantly, are non-verifiable in practical scenarios. Even in cases where the context features of all arms except for the optimal arm are fixed (thus, assumptions such as anti-concentration are not valid), our proposed algorithms outperform the existing algorithms.

**Novel Analysis Technique:** Our proof technique significantly differs from the regret analysis in previous Lasso bandits. We observe that under Assumption 3, if the optimal arm has been selected sufficiently many times up to time
$t$, the estimation error at $t$ can be controlled, and a well-estimated estimator leads to the selection of the optimal arm at the next time step. This observation reveals the cyclic structure regarding the estimation error and the selection of the optimal arms. Previous Lasso bandit papers do not capture this cyclic structure but instead rely on more convenient assumptions or automatic exploration ensured by diverse context distributions. **We analyze the cyclic structure of such good events using a novel mathematical induction argument (Proposition 1 in Appendix.E.1). We assert with confidence that our technical contributions are more distinct and significant than the other Lasso bandit literature, and we believe that this new analytical techniques will significantly influence future research in this field!**

In conclusion, we remain confident in the substantial value our work brings to the table, not only through the **introduction of new, more relaxed conditions and enhanced regret analysis** but also by **presenting a practical, straightforward-to-implement algorithm which is made possible by our new theoretical results.** This work is indeed a complete package, offering theoretical depth, novel insights, and practical applicability.

---

### Meta-Review · Area_Chair_rUXH · 2024-12-22

**Metareview:**

This paper examines the stochastic sparse linear bandit problem and proposes an algorithm that achieves $O(\mathrm{polylog}(T))$ regret under relaxed assumptions compared to prior work. The algorithm is analyzed theoretically, and its effectiveness is validated through numerical experiments. This work can be highly valued for presenting solid improvements on a problem of significant interest within the bandit community.

However, the paper has some weaknesses. It lacks sufficient discussion of specific applications or realistic scenarios that become feasible due to the relaxed assumptions. Additionally, reviewers raised concerns about the requirement for prior knowledge of sparsity parameters and the inability to pre-verify conditions corresponding to the relaxed assumptions. Specifically, while the authors highlight that the conditions in prior work are unverifiable, they do not address the verifiability of their own conditions, which is unfair and warrants additional discussion. Furthermore, concerns about the clarity and impact of the technical contributions were also noted.

Despite these weaknesses, discussions with the authors, including their rebuttal addressing these points, led to an overall positive assessment from the reviewers. Based on the premise that the above weaknesses will be addressed in the final version, I support the acceptance of this paper.

**Additional Comments On Reviewer Discussion:**

The reviewers expressed concerns regarding the requirement for prior knowledge of the sparsity parameters and the inability to pre-verify the conditions corresponding to the relaxed assumptions. Regarding the latter, while the authors highlight the unverifiability of conditions in prior work, they do not address the verifiability of their own conditions, which is not fair. This point should be clarified and added to the discussion. Additionally, concerns were raised about the clarity and impact of the technical contributions.

However, following the authors' rebuttal and subsequent discussions that addressed these weaknesses to some extent, the reviewers have provided an overall positive evaluation. On the premise that these weaknesses will be addressed in the final version, I support the acceptance of this paper.

---

### Decision · Program_Chairs · 2025-01-22

Accept (Poster)